# TabICLv2: A Better, Faster, Scalable, and Open Tabular Foundation Model

**Jingang Qu** [* 1]  **David Holzmüller** [* 1]  **Gaël Varoquaux** [1 2]  **Marine Le Morvan** [1]

## Abstract

Tabular foundation models, such as TabPFNv2 and TabICL, have recently dethroned gradient-boosted trees at the top of predictive benchmarks, demonstrating the value of in-context learning for tabular data. We introduce TabICLv2, a new state-of-the-art foundation model for regression and classification built on three pillars: (1) a novel synthetic data generation engine designed for high pretraining diversity; (2) various architectural innovations, including a new scalable softmax in attention improving generalization to larger datasets without prohibitive long-sequence pretraining; and (3) optimized pretraining protocols, notably replacing AdamW with the Muon optimizer. On the TabArena and TALENT benchmarks, TabICLv2 without any tuning surpasses the performance of the current state of the art, RealTabPFN-2.5 (hyperparameter-tuned, ensembled, and fine-tuned on real data). With only moderate pretraining compute, TabICLv2 generalizes effectively to million-scale datasets under 50GB GPU memory while being markedly faster than RealTabPFN-2.5. We provide extensive ablation studies to quantify these contributions and foster open research by releasing code for inference, pretraining, and synthetic data generation at https://github.com/soda-inria/tabicl.

## 1. Introduction

Tabular data, whether stored in spreadsheets or databases, is ubiquitous across applications ranging from healthcare to credit card fraud detection (Borisov et al., 2022; Jesus et al., 2022; Grinsztajn et al., 2025). While supervised learning on tabular data has long been dominated by gradient-boosted

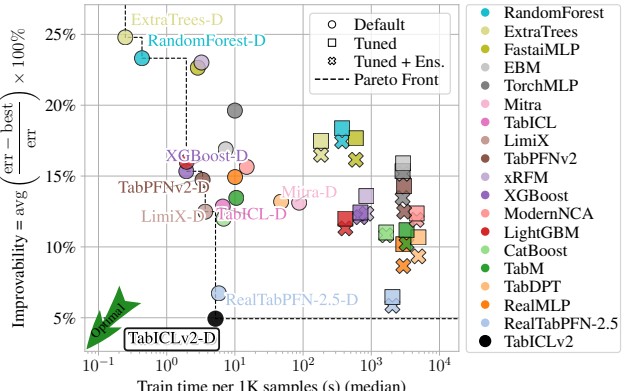

*Figure 1.* **Improvability vs. train time on TabArena (Erickson et al., 2025).** Improvability (lower is better) measures the relative error gap to the best method, averaged across datasets. Train time is training + inference in 8-fold cross-validation. For foundation models, it is dominated by forward passes that perform in-context learning. *Default* uses default hyperparameters; *Tuned* selects the best of 200 random hyperparameter configurations on validation; *Tuned + Ens.* applies post-hoc weighted ensemble of all configurations. The runtime of TabICLv2 is measured on an H100 GPU, while others are from TabArena. Results for inapplicable model-dataset pairs are imputed with default RandomForest.

decision trees (Grinsztajn et al., 2022), both pretrained and trained-from-scratch deep learning models have recently been able to match or even surpass their accuracy on tables with up to 100K samples (Erickson et al., 2025; Ye et al., 2024). In particular, starting from TabPFN (Hollmann et al., 2023), tabular foundation models (TFMs) have received a lot of attention thanks to their ability to perform training and inference in a single forward pass of a Transformer-based architecture. The development of better TFMs also benefits downstream adaptations, such as causal inference, generative modeling, joint predictive distributions, and simulation-based inference (Ma et al., 2026; Robertson et al., 2025; Balazadeh et al., 2025; Hollmann et al., 2025; Hassan et al., 2026; Vetter et al., 2025). To foster this research, there is a pressing need for fully open-source TFMs that rival closed-source ones to democratize access to top-tier performance and demystify the recipe behind top-performing TFMs.

**Contributions** We introduce TabICLv2, a state-of-the-art tabular foundation model, as shown in Figure 1. Our contributions include architectural innovations (Section 3),

*Equal contribution [1]SODA Team, INRIA Saclay, Palaiseau, France [2]Probabl, France. Correspondence to: Jingang Qu <jingang.qu@inria.fr>, David Holzmüller <david.holzmuller@inria.fr>, Marine Le Morvan <marine.le-morvan@inria.fr>.

*Proceedings of the 43rd International Conference on Machine Learning*, Seoul, South Korea. PMLR 306, 2026. Copyright 2026 by the author(s).

pretraining improvements (Section 4), a novel synthetic data generator (Section 5), extensive evaluations (Section 6), and an ablation study (Section 7).

## 2. Related Work

### 2.1. Tabular foundation models

Tabular foundation models (TFMs) based on Prior-data fitted networks (PFNs, Müller et al. 2022) emerged as a paradigm shift in tabular learning. Given a training set and test input, a TFM $q_\theta$ with parameters $\theta$ directly predicts a distribution $q_\theta(y_{\text{test}} \mid x_{\text{test}}, \mathcal{D}_{\text{train}})$ with a forward pass on input $(x_{\text{test}}, \mathcal{D}_{\text{train}})$. For a single dataset, it hence performs in-context learning (ICL) without gradient updates. For pretraining, given a prior $p(\mathcal{D})$ from which datasets $\mathcal{D}$ (train+test) can be sampled, TFMs are trained to minimize

$$\mathcal{L}(\theta) = \mathbb{E}_{\mathcal{D} \sim p(\cdot)} \big[ -\log q_\theta(y_{\text{test}} \mid x_{\text{test}}, \mathcal{D}_{\text{train}}) \big] .$$

**Architectural perspectives.** TabPFN (Hollmann et al., 2023) treats each row as a token and performs ICL over rows. TabPFNv2 (Hollmann et al., 2025) moves to a cell-based design with alternating row and column attentions, where each cell receives a separate representation. However, this incurs $O(n^2 m + nm^2)$ complexity for a table with $n$ rows and $m$ columns. TabPFN-2.5 (Grinsztajn et al., 2025) extends TabPFNv2 with deeper networks. TabICL (Qu et al., 2025) reduces the computational complexity to $O(n^2 + nm^2)$ via a two-stage design: a lightweight column-then-row attention first constructs fixed-dimensional row embeddings, after which ICL is performed over these embeddings. Recent work continues to innovate on these foundations, including Mitra (Zhang et al., 2026), LimiX (Zhang et al., 2025), and Orion-MSP (Bouadi et al., 2025).

**Synthetic prior datasets.** Synthetic priors are central to PFN-style TFMs. TabPFN uses structural causal models (SCMs). TabICL and TabForestPFN (Breejen et al., 2024) extend these by mixing tree-based priors to inject tree inductive biases. Mitra studies prior design principles and proposes mixed priors (SCM + tree ensembles) to better control decision boundaries. TabPFNv2 enriches priors with more sophisticated DAG construction and computational mappings. LimiX introduces hierarchical SCMs with controllable difficulty. Drift-Resilient TabPFN (Helli et al., 2024) tackles temporal distribution shifts with a two-level generative prior that modulates SCM parameters over time. However, TabDPT (Ma et al., 2025) shows that large-scale pretraining on real datasets can be competitive, and Real-TabPFN (Garg et al., 2025) indicates that continued pretraining on real datasets can improve TabPFNv2.

**Fine-tuning, retrieval, and distillation.** Beyond pretraining, adaptation strategies for TFMs include (a) fine-tuning

to shift the learned prior toward a target distribution (Feuer et al., 2024; Liu & Ye, 2025; Garg et al., 2025; Kolberg et al., 2025; Rubachev et al., 2025), (b) retrieval-based context selection to enhance compute-constrained scalability (Thomas et al., 2024; Xu et al., 2025; Zhang et al., 2025; Sergazinov & Yin, 2025), and (c) distilling TFMs into compact MLPs or trees (Bonet et al., 2024; Mueller et al., 2024; Grinsztajn et al., 2025).

**LLM-based tabular models.** In parallel to table-native TFMs, large language models (LLMs) have been adapted to tabular data via table serialization and continued pretraining (Hegselmann et al., 2023; Gardner et al., 2024; Dong et al., 2025), which are promising but underperform TFMs when sufficient training data is available.

### 2.2. Attention struggles with long-context generalization

**Attention fading.** Attention is central to PFN-style TFMs. But standard attention, based on softmax, suffers from *attention fading* (Veličković et al., 2025; Nakanishi, 2025): the softmax denominator increases as context length $n$ grows, causing attention distributions to flatten and preventing sharp focus on relevant tokens. This limits length generalization, as models trained on shorter sequences cannot maintain discriminative attention patterns when applied to longer ones.

**Temperature scaling.** To address attention fading, some work focuses on softmax alternatives (Peters et al., 2019; Ramapuram et al., 2025), which, however, require specialized implementations incompatible with the softmax ecosystem, e.g., FlashAttention (Dao et al., 2022). We thus focus on a less invasive solution: *temperature scaling*. Standard attention (Vaswani et al., 2017) already incorporates a fixed scaling temperature factor $1/\sqrt{d}$ to prevent dot-product magnitudes from growing with dimension, but this does not address length-dependent fading. YaRN (Peng et al., 2024) scales temperature with context length for RoPE (Su et al., 2024) extension, but the scaling is fixed and requires positional encoding. Recent work proposes dynamic alternatives. Scalable Softmax (SSMax, Nakanishi 2025) scales attention logits by $s \log n$, where $s$ is a learnable per-head parameter. Concurrent theoretical analysis (Chen et al., 2025b) establishes that $\log n$ scaling is necessary to maintain attention sharpness as context length $n$ grows. Adaptive-Scalable Entmax (ASEntmax, Vasylenko et al. 2026) extends this with content-aware scaling $\delta + \beta(\log n)^\gamma$, where $\delta$ is a length-independent constant offset, and $\beta = \text{softplus}(\text{MLP}(X))$ and $\gamma = \tanh(\text{MLP}(X))$ are input-dependent. Selective Attention (Zhang et al., 2024) takes a different approach by introducing query-dependent temperature $\tau(q)$ via lightweight MLPs, decoupling the scaling from context length entirely.

# 3. Architecture

The architecture of TabICLv2 is illustrated in Figure 2. Following TabICL, TabICLv2 chains column-wise embedding, row-wise interaction, and dataset-wise ICL, thus preserving the efficiency of TabICL with a runtime complexity of $O(n^2 + nm^2)$ for tables with $n$ rows and $m$ columns. In addition, we introduce several improvements that significantly enhance performance without increasing model size (see the ablation study in Section 7), placing it around 28M parameters. In the following, we mark these improvements with ▶. We focus below on architectural innovations. For model configuration details (e.g., number of layers), refer to Appendix A.4. We provide a short self-contained implementation of the TabICLv2 architecture for educational and experimental purposes, inspired by nanoTabPFN (Pfefferle et al., 2025), at

https://github.com/soda-inria/nanotabicl.

▶ **Repeated feature grouping.** TabICL embeds each feature independently, which can lead to representation collapse when features share similar distributions. TabPFNv2 and TabPFN-2.5 mitigate this collapse by grouping multiple columns into single tokens, which also reduces the number of effective features to improve efficiency, but this reduction may lose fine-grained feature information. We propose *repeated feature grouping*, which places each feature into multiple groups via circular shifts while preserving the number of effective features. Specifically, for a table with $m$ columns, we create $m$ groups where the $j$-th group contains columns at positions $(j, j+1, j+3) \bmod m$. Each group is encoded by a shared linear layer $\text{Lin} : \mathbb{R}^3 \to \mathbb{R}^d$:

$$E_1[i,j] = \text{Lin}\big(x_{i,j},\ x_{i,(j+1) \bmod m},\ x_{i,(j+3) \bmod m}\big).$$

The shift pattern $(0, 1, 3)$ ensures that for tables with at least 7 columns, no pair of columns appears together in more than one group. We show in Appendix A.1 that this pattern generalizes to arbitrary group sizes, although we did not observe consistent improvements from larger groups.

▶ **Target-aware embedding.** We find it beneficial to inject target information early. After repeated feature grouping produces input data representation $E_1 \in \mathbb{R}^{n \times m \times d}$, we add target embeddings to each training token:

$$E_2[i,j] = E_1[i,j] + \text{Embed}_{\text{TAE}}(y_i), \quad i \in \mathcal{D}_{\text{train}},$$

where $\text{Embed}_{\text{TAE}}$ is a linear layer for regression or a learnable lookup table for classification. Unlike TabPFNv2 appending the target as an additional column, we directly add target embeddings to all features. This also helps mitigate representation collapse because even when two features share similar distributions, their association with target values often differs across samples. Unlike TabICL, which

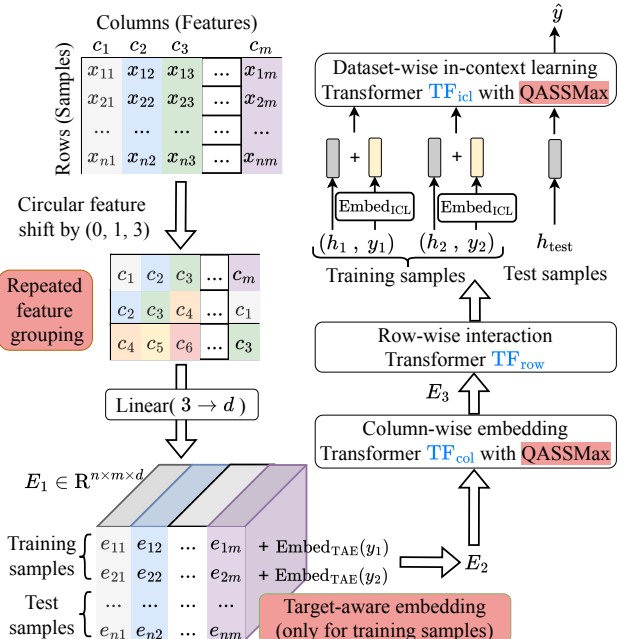

*Figure 2.* **Architecture of TabICLv2.** Given an input $X \in \mathbb{R}^{n \times m}$, *repeated feature grouping* encodes columns into multiple groups via circular shifts to break feature symmetries, and *target-aware embedding* injects target information from the beginning. $\text{TF}_{\text{col}}$ embeds each feature through a set transformer, $\text{TF}_{\text{row}}$ aggregates features into row representations $h$, and $\text{TF}_{\text{icl}}$ performs in-context learning to predict test targets $\hat{y}$. QASSMax, our query-aware scalable softmax, is applied in the part of $\text{TF}_{\text{col}}$ where inducing points aggregate input information and $\text{TF}_{\text{icl}}$ to mitigate attention fading and improve long-context generalization.

compresses each row in a few tokens before seeing the targets, our early target injection allows TabICLv2 to determine the relevance of features before compression.

**Compression then ICL.** TabICLv2 processes $E_2$ in three stages: (1) *column-wise embedding* applies a set transformer $\text{TF}_{\text{col}}$ (Lee et al., 2019) to each column; (2) *row-wise interaction* uses a transformer $\text{TF}_{\text{row}}$ with [CLS] tokens to collapse feature embeddings per row into a single vector; (3) *dataset-wise ICL* combines row embeddings with target embeddings and uses a transformer $\text{TF}_{\text{icl}}$ where test samples attend to training samples for prediction. See Appendix A.2 for details. ▶ Compared to TabICL, our key innovation here is applying a novel scalable softmax to $\text{TF}_{\text{icl}}$ and to the part of $\text{TF}_{\text{col}}$ where inducing points aggregate input information.

▶ **Query-aware scalable softmax.** To improve generalization to larger datasets, we extend Scalable Softmax (SSMax, Nakanishi 2025), a temperature scaling method that sharpens attention distributions by rescaling queries before computing logits. Let $q_h = (q_{hi})$ be a query vector at head $h$ with head dimension indexed by $i$, and let $n$ be the size of the training set. SSMax rescales queries with a learnable

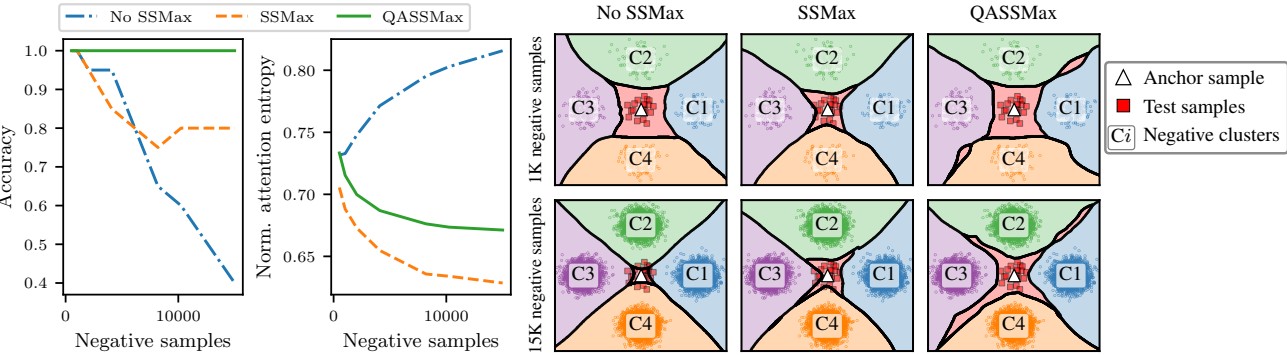

*(a)* Accuracy and attention entropy *vs.* negative samples          *(b)* Decision boundaries for anchor and negative clusters

*Figure 3.* **SSMax variants mitigate attention fading in a synthetic 2D classification task.** We create a dataset consisting of four negative clusters (C1–C4) and one anchor cluster containing a single anchor sample (triangle) in the training set. We increase negative samples while evaluating 20 fixed test samples (red squares) nearest to the anchor. We evaluate three stage 1 checkpoints trained for 280K steps with QASSMax, SSMax, and without SSMax. **(a)** Attention entropy is divided by $\log n$ to ensure values in $(0, 1)$ and averaged across all heads and layers in $\text{TF}_{\text{icl}}$, measuring how uniformly test samples attend to training ones. Without SSMax, accuracy drops and entropy rises as negative samples increase, which is a hallmark of attention fading where the model fails to focus on the relevant anchor. QASSMax maintains 100% accuracy with consistently low entropy. **(b)** shows decision boundaries at 1K and 15K negative samples. The region of the anchor cluster shrinks for all variants as negative samples increase. No SSMax collapses at 15K, while QASSMax preserves a stable boundary containing all test samples.

per-head scalar $s_h$:

$$\tilde{q}_{hi} = q_{hi} \cdot s_h \log n .$$

We propose query-aware scalable softmax (QASSMax), which rescales each query element as:

$$\tilde{q}_{hi} = q_{hi} \cdot \underbrace{\text{MLP}_{\text{base}}(\log n)_{hi}}_{\text{base scaling}} \cdot \underbrace{\left(1 + \tanh(\text{MLP}_{\text{gate}}(q_h)_i)\right)}_{\text{query-aware gating}}$$

where for $H$ attention heads, $\text{MLP}_{\text{base}} : \mathbb{R} \to \mathbb{R}^{H \times d_{\text{head}}}$ and $\text{MLP}_{\text{gate}} : \mathbb{R}^{d_{\text{head}}} \to \mathbb{R}^{d_{\text{head}}}$ are two-layer MLPs with 64 hidden neurons and GELU activation.

We design QASSMax based on the following rationale: (a) The $\log n$ factor is critical as it counteracts the linear growth of the softmax denominator with respect to $n$ (Nakanishi, 2025; Chen et al., 2025b); (b) ASEntmax (Vasylenko et al., 2026) uses learnable $\delta + \beta(\log n)^{\gamma}$, inspiring us to generalize to $\text{MLP}_{\text{base}}(\log n)$; (c) Element-wise scaling increases expressiveness beyond per-head scalars; (d) Selective Attention (Zhang et al., 2024) introduces query-awareness in temperature scaling, which motivates us to use the bounded query-aware gating $\in (0, 2)$ that modulates the base scaling without dominating the $\log n$ trend. In addition, our gating design shares similar insights with Gated Attention (Qiu et al., 2025), which applies gating to attention outputs and finds query-dependent, element-wise gating most effective.

QASSMax applied to $\text{TF}_{\text{col}}$ and $\text{TF}_{\text{icl}}$ yields substantial performance improvements, as shown in the ablation study (Section 7). To study its effect on attention fading, we design a toy needle-in-haystack classification task (Figure 3):

the model must focus on a single anchor sample (the needle) among increasing negative samples (the haystack). Without scalable softmax, attention entropy rises and accuracy drops. However, QASSMax maintains low entropy and 100% accuracy even with 15K negatives, outperforming SSMax which largely degrades at extreme scales.

**Many-class classification.** Like many TFMs, TabICLv2 is pretrained with up to 10 classes. We use hierarchical classification (Qu et al., 2025) at the ICL stage for more classes. However, target-aware embedding introduces labels before hierarchical partitioning. ► To address this, we propose *mixed-radix ensembling*: for $C > 10$ classes, we compute balanced bases $[k_0, \ldots, k_{D-1}]$ with each $k_i \le 10$ and $\prod_i k_i \ge C$, then decompose each label $y$ into $D$ digits $y^{(i)} \in \{0, \ldots, k_i - 1\}$ via mixed-radix representation. Each digit defines a coarser grouping of the original classes. We run $\text{TF}_{\text{col}}$ once per digit and average the outputs:

$$O_{\text{avg}} = \frac{1}{D} \sum_{i=0}^{D-1} \text{TF}_{\text{col}}(E_1 + \text{Embed}_{\text{TAE}}(y^{(i)})) .$$

Combined with hierarchical classification in $\text{TF}_{\text{icl}}$, this enables TabICLv2 to handle an arbitrary number of classes. See Appendix A.3 for details.

► **Quantile predictions for regression.** Existing TFMs adopt different strategies for regression: TabPFNv2 and TabPFN-2.5 model the full predictive distribution by discretizing the target space into bins and applying cross-entropy loss, while Mitra and TabDPT predict point estimates using MSE loss. In addition, like most TFMs except LimiX, we train separate models for classification and re-

gression.

TabICLv2 instead predicts 999 quantiles at probability levels $\alpha \in \{0.001, 0.002, \ldots, 0.999\}$, trained with pinball loss summed across all quantiles. In preliminary experiments using RMSE evaluation, we found that quantile regression outperforms MSE and the bin-based approach of TabPFNv2.

At inference, for point estimation we simply average the predicted quantiles, which proves both fast and effective. For probabilistic predictions, we construct a full distribution from the quantiles by enforcing monotonicity via sorting (the default) or isotonic regression (Barlow & Brunk, 1972; Busing, 2022), extrapolating tails with parametric exponential models, and deriving closed-form PDF, CDF, and moments. See Appendix I for details.

## 4. Pretraining and Inference

### 4.1. Pretraining setup

We significantly improve the pretraining setup compared to TabICL, the reference TFM with open pretraining.

**Three pretraining stages.** We retain the three-stage structure of TabICL that progressively expands the size of pretraining datasets, with up to 100 features throughout all stages. However, following TabPFNv2, we reduce the batch size to 64, allowing more steps with fewer datasets ($\approx$35M) than TabICL ($\approx$83M) and TabPFNv2 ($\approx$130M). The three stages are:

- **Stage 1**: 500K steps on datasets with 1,024 samples, 30–90% for training, max learning rate 8e-4.

- **Stage 2**: 40K steps on datasets with 400–10,240 samples (log-uniform), 80% for training, max learning rate 1e-4.

- **Stage 3**: 10K steps on datasets with 400–60K samples (log-uniform), 80% for training, max learning rate 2e-5.

In Appendix B.1, we show that stages 2 and 3 yield progressive performance improvements, especially on large datasets.

**Optimizer.** We use the Muon optimizer (Jordan et al., 2024b) based on the implementation of Schaipp (2025) instead of AdamW (resp. Adam) used by TabICL (resp. TabPFNv2). Following Moonlight (Liu et al., 2025), this implementation multiplies the learning rate for each parameter $W \in \mathbb{R}^{n \times m}$ by $0.2 \cdot \sqrt{\max\{n, m\}}$. Muon orthogonalizes gradient updates via Newton-Schulz iteration, which corresponds to steepest descent under the spectral norm (Bernstein & Newhouse, 2024), and has been observed to speed up LLM training (Jordan et al., 2024a; Liu et al., 2025). We find higher learning rates preferable for Muon. As a result, we use a max learning rate of 8e-4 for stage 1

compared to 1e-4 for AdamW in TabICL. We adopt cautious weight decay (Chen et al., 2025a) with parameter 0.01, which applies decay only when the update and parameter have the same sign, avoiding interference with beneficial gradient directions. We also increase gradient clipping from 1 to 10 for stages 1 and 2, sample different train/test sizes per micro-batch, and use a cosine learning rate schedule across all stages.

**Pretraining cost.** On H100 GPUs with 80GB memory, stage 1 takes around 20 GPU-days, stage 2 around 2.5 GPU-days, and stage 3 around 2 GPU-days, totaling 24.5 GPU-days per model. Given that one H100-hour is roughly equivalent to 2 A100-hours, our pretraining cost is lower than TabICL (60 A100-days).

### 4.2. Inference optimizations

We implement *disk offloading* (Appendix H.2), reducing requirements to under 24 GB CPU and 50 GB GPU to process a table with 1M samples and 500 features within 450 seconds (Figure H.2). Combined with QASSMax for long-context generalization, TabICLv2 can natively handle million-scale tables without retrieval and distillation. In addition, we reduce redundant computation by selectively computing $Q/K/V$ projections (Appendix H.1).

## 5. Synthetic Data Prior

Our pretraining data is *entirely synthetic*, following the approach pioneered in TabPFN (Hollmann et al., 2023). The data-generating mechanism is termed *prior*, as it implicitly defines a Bayesian prior over datasets. For TabICLv2, we design a new prior that retains the structural causal model framework used in Hollmann et al. (2023), incorporates innovations brought by TabICL and TabPFNv2 priors, and adds many novel design options and sampling mechanisms (see Appendix E.1). Unlike architectural and pretraining choices, the new prior is developed mostly without experimental feedback, since fine-grained ablations are impractical and prone to overfitting validation datasets. Instead, the prior development is guided by general design principles (Wilson & Izmailov, 2020), maximizing dataset diversity (e.g., variable dependencies and categorical cardinalities) while encoding useful inductive biases and preserving computational efficiency. This new prior is key to the final performance: pretraining TabICLv2 with the TabICL prior yields substantially lower performance (Figure 10, gray). An ablation using the TabPFNv2 prior is not possible, as it is not open-source. We provide a high-level prior description below and defer details to Appendices E and F.

**High-level structure.** Figure 4 summarizes the TabICLv2 prior. We first sample global dataset properties, such as

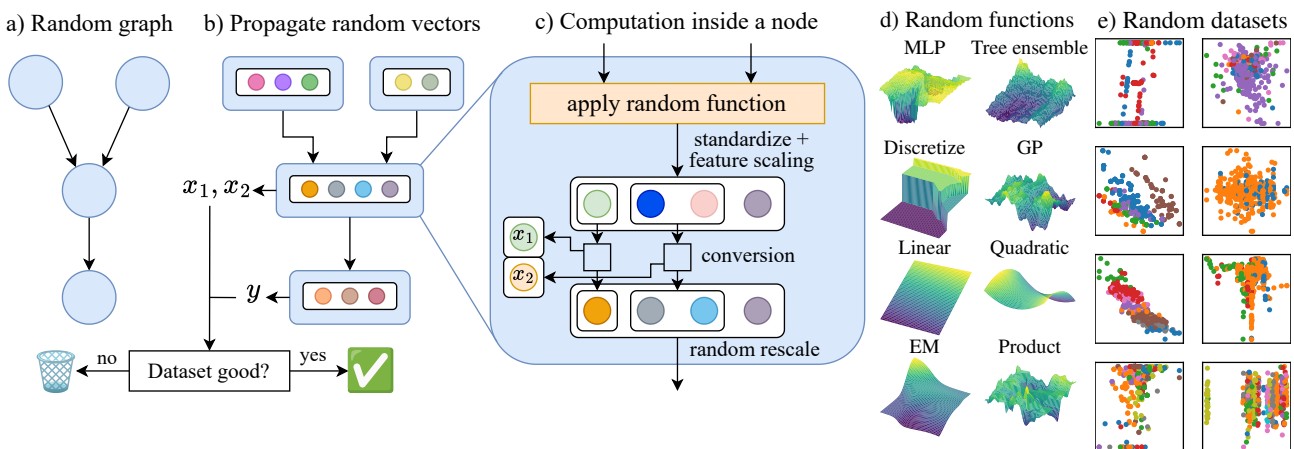

*Figure 4.* **High-level structure of the synthetic dataset generation prior.** Random vectors (one per sample) are propagated through a randomly generated graph where each node computes a random function of its parents. Columns of the final dataset are extracted from randomly assigned nodes. The resulting datasets can be rejected based on different filtering criteria. (d) List of the 8 random functions applied: (MLP) Multilayer perceptrons, (Tree Ensemble) Ensembles of symmetric trees inspired by CatBoost (Prokhorenkova et al., 2018), (Discretize) Discretization to nearest neighbors among a random set; (GP) Multivariate Gaussian process functions; (Linear) Linear functions; (Quadratic) Multivariate quadratic functions; (EM) functions with plateaus inspired by the cluster assignment in the EM algorithm; (Product) products of other random functions. (e) Examples of generated 2D classification datasets (cf. Figure F.1).

the number of numerical and categorical features, and the dataset size. We then sample a directed acyclic graph and random functions defining parent–child relationships, yielding a causal data-generating model. To obtain a dataset with $n$ samples, a matrix $X \in \mathbb{R}^{n \times d_i}$ of $n$ random vectors is sampled at each root node $i$ and propagated through the graph. Each dataset feature is extracted from a randomly assigned node. Only a subset of a node's dimensions is used to generate each feature, leaving other dimensions unobserved and thereby introducing noise into the dataset. Unlike prior work, we do not add Gaussian noise at the node level. For numerical features (e.g., $x_1$), feature values are extracted from a single node dimension. For categorical features (e.g., $x_2$), multiple node dimensions are extracted and discretized, either via nearest-neighbor assignment or by applying a softmax to obtain a categorical distribution.

**New sampling mechanisms.** Since the **random graph** sampling mechanism used by Hollmann et al. (2025) can only generate tree-structured graphs, we introduce a "random Cauchy graph" mechanism, which models different global and local node connectivities and is described in Appendix E.4.

The relation between a child node and its parents is generated using several steps depicted in Figure 4(c). The key step consists in sampling diverse **random functions** to apply to the parent data. We use eight types of random functions, listed in Figure 4(d). The first three are adapted from TabPFNv2 while the other five are new. These functions are chosen to cover different levels of smoothness (which we prove for Gaussian Process functions) and different types of inductive biases (e.g., plateaus or axis-alignment). To

handle the case of more than one parent node, we randomly select between two options: concatenate all parent matrices and apply a single random function, or apply random functions to every parent matrix and aggregate the results using sum, product, max, or logsumexp. Even within each function type, we diversify the generated functions using new or extended building blocks including multiple random matrix types (for MLP, linear, quadratic functions, etc), random weight vectors (for singular values, feature importances, etc.), and random activations (for MLPs, random matrices), see Appendix E.

After applying the random function, we standardize $X$ and randomly rescale its columns to emulate different feature importances. The random converters extract feature values but can also modify node values, applying a warping function to scalars or discretization mechanisms to sub-vectors (see Appendix E.6). Finally, the node data $X$ is multiplied by a random scalar emulating a "node importance".

**Postprocessing.** We apply some postprocessing similar to TabICL (Qu et al., 2025), including discarding problematic columns and datasets, permuting columns and class labels, and preprocessing features and targets.

**Data filtering.** Inspired by Dong et al. (2025) and Zhang et al. (2025), we filter out datasets on which a simple Extra-Trees model cannot improve on a constant baseline according to a bootstrap test. In addition, we directly filter graphs in which nodes associated with $x$ do not have common ancestors with the node associated with $y$, which implies that $y$ is independent of $x$. In pretraining stage 1, roughly 35% classification and 25% regression datasets are filtered.

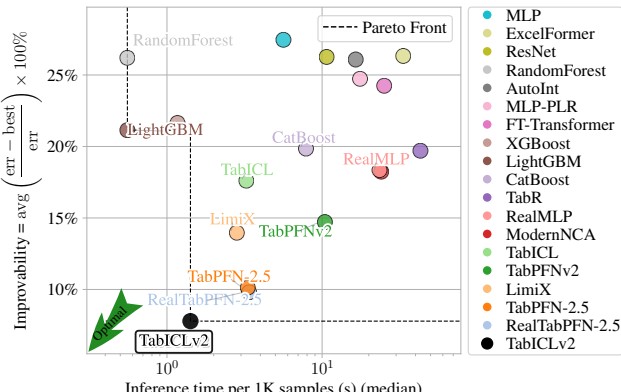

Figure 5. **Improvability vs. inference time on TALENT (Ye et al., 2024).** The runtime of TabICLv2 is measured on an H100 GPU, while other runtimes are taken from TALENT.

Figure 10 shows that filtering improves the convergence of pretraining.

**Sampling correlated scalars.** We often sample scalars ("hyperparameters") from the same distribution multiple times, e.g., the number of categories within a column. We introduce a mechanism to sample them in a correlated fashion based on sampling shared parameters of a Beta distribution. For example, we observe that in some real datasets, many categorical columns have the same cardinality. Through correlated sampling, our prior is more likely to replicate this phenomenon.

## 6. Experiments

**Benchmarks.** We use the TabArena (Erickson et al., 2025) and TALENT (Ye et al., 2024) benchmarks. TabICLv2 ensembles predictions using random column/class shuffles and different preprocessors, as in TabICL. TabArena contains 51 datasets (38 classification with ≤10 classes, 13 regression), evaluated via repeated cross-validation with ROC AUC for binary, log-loss for multiclass, and RMSE for regression. We use 8 estimators for TabICLv2 to match RealTabPFN-2.5 in TabArena. TALENT contains 300 datasets (120 binary, 80 multiclass, 100 regression) with 64%/16%/20% train/validation/test splits. Hyperparameters are selected on the validation set using accuracy for classification and RMSE for regression. We use 32 estimators for both TabICLv2 and TabPFN-2.5/RealTabPFN-2.5 in TALENT.

We primarily report improvability, which measures the average relative error gap to the best method on each dataset, allowing the best method to vary across datasets. Improvability reflects the magnitude of performance differences compared to rank-based metrics. We provide other metrics in Appendices J and K.

In addition, following TabArena and TALENT, we use the

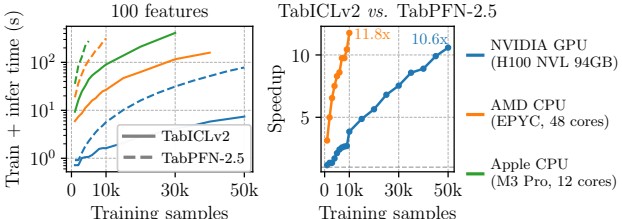

Figure 6. **Runtime comparison between TabICLv2 and TabPFN-2.5 with respect to the number of training samples and hardware.** Both use 8 estimators. We use classification with 500 test samples.

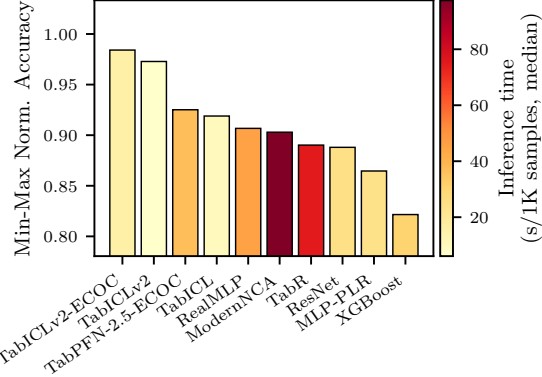

Figure 7. **Normalized accuracy across 12 datasets with more than 10 classes on the TALENT benchmark.**

TabICLv1.1 checkpoint for TabICL (Qu et al., 2025), which is TabICL post-trained on an earlier version of our prior.

**TabICLv2 is state-of-the-art on both benchmarks.** As shown in Figures 1 and 5, TabICLv2 dominates the Pareto fronts of improvability versus runtime. Without any tuning, TabICLv2 surpasses RealTabPFN-2.5 (tuned + ensembled), the current state-of-the-art that is not fully open-source. A statistical test confirms that the win-rate of TabICLv2 vs. RealTabPFN-2.5 on TALENT is greater than 50% (Appendix L.2). TabICLv2 also substantially outperforms heavily tuned traditional methods, such as CatBoost and XGBoost, despite requiring orders of magnitude less training time.

**TabICLv2 is consistently faster than TabPFN-2.5.** As shown in Figure 6, for 100 features, TabICLv2 is faster than TabPFN-2.5 across all hardware, with speedups increasing at larger scales: 10.6× on an H100 GPU at 50K samples. The efficiency gap is even more pronounced on CPU, reaching 11.8× at just 10K samples.

**TabICLv2 excels on many-class classification.** TabICLv2 with both the error-correcting output codes (ECOC) wrapper from TabPFNv2 and our native mixed-radix ensembling substantially outperforms all baselines on TALENT

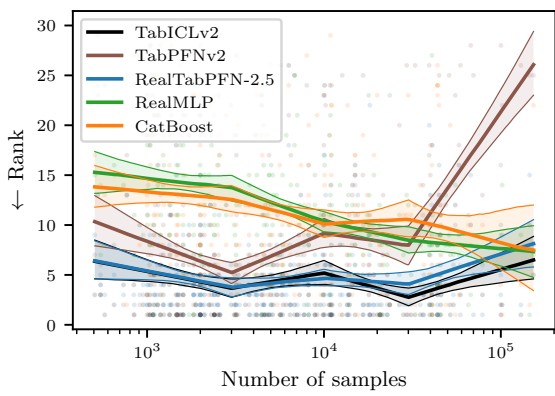

*Figure 8.* **Model rankings as a function of sample size on TAL-ENT.** The lines show the bootstrap median and 10% / 90% bootstrap confidence intervals of a piecewise linear fit (Qu et al., 2025).

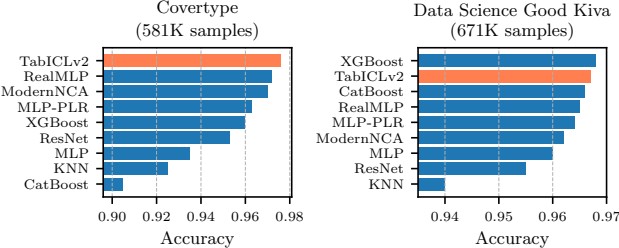

*Figure 9.* **Accuracy on two huge classification datasets from the TALENT extension.** TabICLv2 still performs strongly. TabPFN-2.5 resulted in out-of-memory errors.

datasets with $> 10$ classes (Figure 7). The ECOC wrapper is slightly better but $3\times$ slower than our native handling.

**TabICLv2 scales to large datasets.** As shown in Figure 8, TabICLv2 maintains top rankings across all dataset sizes from $10^3$ to $10^5$, outperforming RealTabPFN-2.5 on larger datasets ($>$20K). On even larger datasets (600K) from the TALENT extension, TabICLv2 still performs strongly (Figure 9). These results show that TabICLv2 further extends the frontier of TFMs for natively handling large-scale data.

**Other experiments.** We provide more experimental results in Appendix L. Our default ensemble size of 8 estimators unlocks most of the benefits of ensembling. In few-shot settings with 16 training samples, which are not included in TabICLv2's pretraining, TabICLv2 comes second to RealTabPFN-2.5, while outperforming (untuned) classical baselines as well as LLMs Claude Opus 4.6 and Qwen3.5-112B. Unlike TabICL(v1), TabICLv2 does not benefit from TabPFNv2's random forest extension for larger datasets (Appendix L.5), showing that it efficiently processes larger datasets. TabICLv2 and RealTabPFN-2.5 are also quite robust to adding label noise compared to tree-based models.

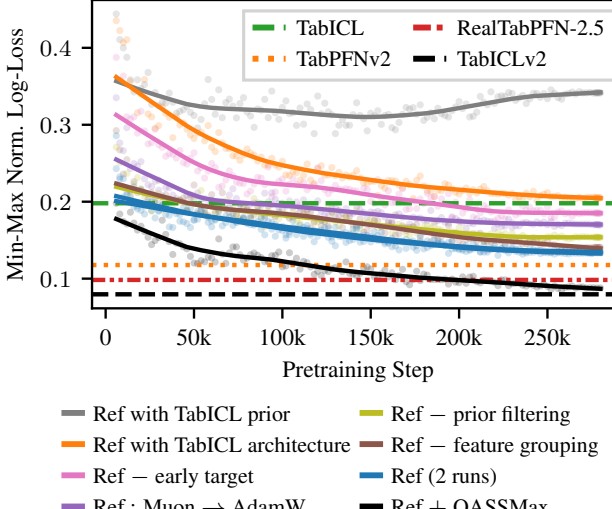

*Figure 10.* **Ablating different components of TabICLv2.** Non-solid horizontal lines denote performance of official checkpoints; solid lines denote ablated models pretrained for 280K steps. Each ablation modifies one component of the reference model (blue) by adding ($+$), removing ($-$), or replacing ($\rightarrow$) it. The reference model corresponds to TabICLv2 without QASSMax and with 4 instead of 8 heads for $TF_{col}$ and $TF_{icl}$. Performance metrics are computed on the 60 validation datasets used for TabPFNv2 development (Hollmann et al., 2025, supplementary Table 5). For each dataset, we use up to 2,048 training samples (fewer when the dataset is smaller) and two train/test splits. The AdamW ablation uses regular weight decay for AdamW and a learning rate of 1e-4 following TabICL. Scatterplots display per-step validation performance and reveal decreasing noise as the learning rate decays.

## 7. Ablation Study

We conduct an ablation study to assess the impact of architectural, prior, and pretraining choices (Figure 10). See Appendix C for more ablation results. The performance gap between the reference TabICLv2 checkpoint (dotted black) and its ablation (solid black) is explained by pretraining length: the ablation is trained for 280K steps, whereas the official checkpoint is pretrained for more steps (500K steps for stage 1 plus stages 2 & 3) and uses 8 instead of 4 attention heads in $TF_{col}$ and $TF_{icl}$. Interestingly, TabICLv2 matches RealTabPFN-2.5 in log-loss after $\approx$ 200K steps, and in $<$ 100K steps in terms of normalized accuracy.

First, we observe a strong interaction between architecture and prior. Pretraining TabICLv2 with the TabICL prior fails (gray line): the performance remains below TabICL and the validation loss degrades in the second half of the pretraining. This suggests that the TabICLv2 architecture requires higher prior diversity to generalize, perhaps similar to how Ma et al. (2025) observed that scaling laws can break down with weak synthetic data generators. Additionally, pretraining the TabICL architecture with the TabICLv2 prior (orange) only matches TabICL, indicating that the TabICL architecture

has limited ability to exploit increased prior diversity.

Across metrics (Figure C.1) including normalized accuracy, Elo, and log-loss, the ordering of ablations is consistent. The prior yields the largest effect. Three components provide comparable, significant gains ($\approx$100 Elo, 64% win rate): early target inclusion, Muon instead of AdamW, and QASSMax. Repeated feature grouping and prior filtering yield smaller gains.

## 8. Limitations

TabICLv2 shares common limitations with related models: it does not natively leverage semantic information from column names or textual features, shown to be valuable (Spinaci et al., 2025), but its scalability to large numbers of features suggests it should remain reasonably fast when combined with text embedding models. Additionally, despite improved scalability, datasets with millions of samples remain challenging. Many extensions, such as multi-output regression or handling distribution shifts (Helli et al., 2024) are also left to future work. Due to the lack of established benchmarks, the distributional regression capabilities of TabICLv2 are not evaluated beyond toy datasets (Appendix I.9). Adding missing indicators (Le Morvan & Varoquaux, 2025) or introducing missingness during pretraining may improve the handling of missing values, which are currently imputed by the mean, but remain unexplored. Finally, hyperparameter tuning or fine-tuning (Rubachev et al., 2025) could further improve the performance at the cost of increased runtime, but is not explored here.

## 9. Conclusion

TabICLv2 represents a large step forward in tabular foundation models (TFMs), as it achieves state-of-the-art performance and redefines the native scalability of TFMs. Through its open-source prior, pretraining, and inference code, we democratize access to state-of-the-art TFMs. With its moderate pretraining and inference cost, TabICLv2 provides an excellent basis for future adaptations, including foundation models for time-series, causality, and relational databases. In addition, we prioritize out-of-the-box performance and principled innovations over fine-tuning on real data (Garg et al., 2025) or scaling up with deeper (Grinsztajn et al., 2025) or wider (Ma et al., 2025; Zhang et al., 2026) architectures. We hope that TabICLv2 motivates continued open innovation towards smaller, faster, better models.

## Contribution Statement

JQ and DH contributed equally to this work (co-first authors). JQ developed the regression framework and multi-class extensions, enabled scaling to large $n$ through opti-

mized implementation and the design of QASSMax, conducted the majority of experiments and benchmark evaluations, and prepared production-ready code. DH conceived and implemented the new generative prior and other core architectural and pretraining enhancements, conducted small-scale experiments, and implemented nanotabicl. MLM and JQ managed larger-scale pretraining runs and the systematic evaluation of model variants. All authors participated in the weekly technical steering, experimental design, and iterative refinement of the manuscript.

## Acknowledgements

We thank Fabian Schaipp for encouraging us to try his Muon implementation. We thank Tizian Wenzel and Ingo Steinwart for helpful discussions on theory. We are grateful to the authors of the TALENT benchmark for providing information about the TALENT extension. We also thank the LimiX team, especially Xingxuan Zhang and Peng Cui, for their generous support in providing computational resources. We thank Ingo Steinwart for providing computational resources funded by Deutsche Forschungsgemeinschaft (DFG, German Research Foundation) under Germany's Excellence Strategy - EXC 2075 – 390740016.

This work was performed using HPC resources from GENCI–IDRIS (Grant 2024-AD011014864R1, 2024-AD011016033, 2025-AD011016033R1). MLM and JQ acknowledge support from INRIA AEX NAP, and from ANR via the grant StatQA (ANR-25-CE23-5646). GV acknowledges support from ANR via grant TaFoMo (ANR-25-CE23-1822). This work is partly supported by Hi! PARIS and ANR/France 2030 program (ANR-23-IACL-0005).

## Impact Statement

This paper presents work whose goal is to advance the field of Machine Learning. There are many potential societal consequences of our work, none of which we feel must be specifically highlighted here.

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

# Appendices

## Contents

# A. More Architecture Details About TabICLv2

## A.1. Repeated feature grouping

The generalization of our repeated feature grouping pattern from groups of three columns to groups of $k$ columns among $m$ columns is to define group $i$ (for $i \geq 0$) as

$$\left(x_{i+2^0-1 \bmod m}, x_{i+2^1-1 \bmod m}, \ldots, x_{i+2^{k-1}-1 \bmod m}\right).$$

Lemma A.1 shows that whenever $m \geq 2^k$, no pair of columns occurs together in two groups. In principle, larger $k$ yields a more expressive architecture, as the linear layer can always learn to ignore some of the columns. In practice, we use $k = 3$ since larger values did not seem beneficial in initial experiments.

**Lemma A.1** (Intersections of feature groups). *For $k \geq 0, m \geq 1, i \in \{0, \ldots, m-1\}$, define the set*

$$I_{i,k,m} := \{i + 2^l - 1 \bmod m \mid 0 \leq l \leq k-1\}.$$

*Then, if $m \geq 2^k$, we have $|I_{i,k,m} \cap I_{j,k,m}| \leq 1$ for all $0 \leq i < j \leq m-1$.*

*Proof.* Suppose that $|I_{i,k,m} \cap I_{j,k,m}| \geq 2$. But then, by shift invariance, for all $n \in \mathbb{Z}$, $|I_{i+n \bmod m,k,m} \cap I_{j+n \bmod m,k,m}| \geq 2$. Hence, by a suitable shift, we can assume without loss of generality that $i = 0$ and $j \leq m/2 \leq m - 2^{k-1}$. Hence, the modulo in the definitions of $I_{i,k,m}$ and $I_{j,k,m}$ does nothing and we can find $a, b, c, d \in \{0, \ldots, k-1\}$ with $a \neq b$ and

$$i + 2^a - 1 = j + 2^c - 1,$$
$$i + 2^b - 1 = j + 2^d - 1.$$

which in particular implies

$$(i + 2^b - 1) - (i + 2^a - 1) = (j + 2^c - 1) - (j + 2^d - 1),$$

yielding $2^b + 2^d = 2^a + 2^c$. But this means that $\{b, d\} = \{a, c\}$. From $i < j$ we know that $a > c$ and $b > d$, implying $a = b$ and $c = d$, contradicting our previous assumption. $\square$

Note that the assumption $m \geq 2^k$ can likely be relaxed to $m \geq 2^k - 1$.

## A.2. Compression then ICL

**Column-wise embedding.** Column-wise embedding processes each column through a set transformer $\text{TF}_{\text{col}}$ (Lee et al., 2019). Its core is *induced self-attention* with $k$ inducing vectors that proceeds in two stages: the first stage aggregates input information into inducing vectors and the second broadcasts back to the input, reducing complexity from $O(n^2)$ to $O(nk)$. We make three improvements: (a) we directly use the outputs of $\text{TF}_{\text{col}}$ as feature embeddings, while TabICL applies an additional transformation; (b) we apply our query-aware scalable softmax (QASSMax) to the first stage of induced self-attention; and (c) $\text{TF}_{\text{col}}$ is essentially an in-context learner operating within each column thanks to target-aware embedding.

**Row-wise interaction.** Following TabICL (Qu et al., 2025), we prepend four learnable [CLS] tokens to each row and process them through a transformer $\text{TF}_{\text{row}}$ with RoPE (Su et al., 2024). The outputs of the [CLS] tokens are concatenated to form a $4d$-dimensional row embeddings, effectively collapsing the feature dimension.

**Dataset-wise in-context learning.** Training row embeddings are combined with target embeddings. A transformer $\text{TF}_{\text{icl}}$ processes all embeddings, where test samples attend only to training samples. A two-layer MLP converts the outputs into target predictions. We apply QASSMax to $\text{TF}_{\text{icl}}$ to improve its long-context generalization.

## A.3. Many-class classification

Like many tabular foundation models, TabICLv2 is pretrained on classification tasks with up to 10 classes. For tasks with more classes, TabICL (Qu et al., 2025) proposed hierarchical classification during the ICL stage, which recursively partitions classes into subproblems with at most 10 classes each. However, our target-aware embedding introduces label information

before hierarchical partitioning occurs. Since our label encoder supports at most 10 classes, we need a mechanism to handle many-class scenarios at this early stage. We propose *mixed-radix ensembling*, which generates multiple simplified views of the original labels, each containing at most 10 classes. The key idea is to decompose the class label using a mixed-radix number system, where each digit corresponds to a different view of the classification problem.

**Computing balanced bases.** For a task with $C > 10$ classes, we first compute a sequence of balanced bases $[k_0, k_1, \ldots, k_{D-1}]$ satisfying two constraints:

1. Each base is bounded: $k_i \leq 10$ for all $i$

2. The product covers all classes: $\prod_{i=0}^{D-1} k_i \geq C$

We select bases to be as balanced as possible (i.e., $k_i \approx k_j$) to ensure each view captures roughly equal discriminative information. The number of views $D$ is minimized subject to these constraints.

**Mixed-radix label encoding.** Given the bases $[k_0, \ldots, k_{D-1}]$, each class label $y \in \{0, 1, \ldots, C - 1\}$ is re-encoded into $D$ views using mixed-radix decomposition:

$$y^{(i)} = \left\lfloor y \,/\, \prod_{j>i} k_j \right\rfloor \bmod k_i, \quad i = 0, \ldots, D-1 \tag{A.1}$$

This is analogous to representing a number in a mixed-radix positional system, where each "digit" $y^{(i)}$ takes values in $\{0, 1, \ldots, k_i - 1\}$. Consider a 16-class problem ($C = 16$) with bases $[k_0, k_1] = [4, 4]$:

**View 0**: $y^{(0)} = \lfloor y/4 \rfloor$ partitions classes into consecutive blocks:
  - Classes $\{0, 1, 2, 3\} \to 0$
  - Classes $\{4, 5, 6, 7\} \to 1$
  - Classes $\{8, 9, 10, 11\} \to 2$
  - Classes $\{12, 13, 14, 15\} \to 3$

**View 1**: $y^{(1)} = y \bmod 4$ partitions classes by remainder:
  - Classes $\{0, 4, 8, 12\} \to 0$
  - Classes $\{1, 5, 9, 13\} \to 1$
  - Classes $\{2, 6, 10, 14\} \to 2$
  - Classes $\{3, 7, 11, 15\} \to 3$

Each view creates a different grouping of the original classes, and no single view can distinguish all 16 classes. However, the combination of both views uniquely identifies each class: the pair $(y^{(0)}, y^{(1)})$ forms a bijection with the original label $y$.

**Ensemble aggregation.** For each view $i$, we run the column-wise transformer $\text{TF}_{\text{col}}$ with the corresponding re-encoded labels:

$$O^{(i)} = \text{TF}_{\text{col}}(E_1 + \text{Embed}_{\text{TAE}}(y^{(i)})) \tag{A.2}$$

where $E_1$ denotes the embeddings before label injection and $\text{Embed}_{\text{TAE}}$ is the target-aware embedding layer. The final output is the average across all views:

$$O_{\text{avg}} = \frac{1}{D} \sum_{i=0}^{D-1} O^{(i)} = \frac{1}{D} \sum_{i=0}^{D-1} \text{TF}_{\text{col}}(E_1 + \text{Embed}_{\text{TAE}}(y^{(i)})) \tag{A.3}$$

**Relationship to error-correcting output codes.** Our approach is inspired by error-correcting output codes (ECOC) (Dietterich & Bakiri, 1994), which decomposes multi-class classification into multiple binary problems. However, unlike ECOC which uses binary codes and trains separate classifiers, our method: (1) uses $k$-ary codes with $k \leq 10$ to match our pretrained label encoder capacity, (2) operates at the embedding level rather than the prediction level and averages embeddings rather than combining binary predictions.

**Combined with hierarchical classification.** Hierarchical classification operates at the ICL stage, while mixed-radix ensembling handles many-class scenarios at the column-wise embedding stage. Together, they enable TabICLv2 to handle classification tasks with an arbitrary number of classes without retraining.

### A.4. Model configuration

TabICLv2 adopts an architecture similar to TabICL, consisting of column-wise embedding through a Set Transformer $TF_{col}$, row-wise interaction through a Transformer encoder $TF_{row}$, and dataset-wise in-context learning through a Transformer encoder $TF_{icl}$. We train separate checkpoints for classification and regression tasks. Table A.1 summarizes the key architectural differences between the two models.

**Classification model.** $TF_{col}$ consists of three induced self-attention blocks with 128 inducing vectors, model dimension $d = 128$, and 8 attention heads. The target-aware embedding $\text{Embed}_{TAE}$ is a learnable lookup table providing class embeddings for 10 classes.

$TF_{row}$ is a 3-layer Transformer encoder with model dimension $d = 128$ and 8 attention heads. It uses 4 learnable [CLS] tokens to aggregate feature-wise information into a single row representation.

$TF_{icl}$ is a 12-layer Transformer encoder with model dimension $d = 512$ and 8 attention heads. The ICL-stage target embedding $\text{Embed}_{ICL}$ is also a learnable lookup table for 10 classes.

All Transformer attention blocks use pre-norm layer normalization with learnable weights and biases and GELU activations. The feedforward modules use a dimension expansion factor of 2.

The final prediction head is a 2-layer MLP with hidden dimension 1024 and output dimension 10.

For QASSMax, $\text{MLP}_{base} : \mathbb{R} \to \mathbb{R}^{H \times d_{head}}$ and $\text{MLP}_{gate} : \mathbb{R}^{d_{head}} \to \mathbb{R}^{d_{head}}$ are both 2-layer MLPs with hidden dimension 64 and GELU activation. The last layer of $\text{MLP}_{gate}$ is initialized to zero, ensuring the initial modulation is identity.

**Regression model.** Adapting TabICLv2 for regression requires minimal architectural changes: the target-aware embedding $\text{Embed}_{TAE}$ and ICL-stage target embedding $\text{Embed}_{ICL}$ use linear layers to embed continuous targets instead of class lookup tables, and the final MLP outputs 999 quantile predictions instead of class probabilities.

Compared to the classification model, the regression model uses bias-free layer normalizations with learnable weights only.

*Table A.1.* Model configuration for classification and regression.

| Component | Classification | Regression |
|---|---|---|
| *TF$_{col}$ (Column-wise embedding)* | | |
| Layers | 3 | 3 |
| Inducing vectors | 128 | 128 |
| Model dimension | 128 | 128 |
| Attention heads | 8 | 8 |
| *TF$_{row}$ (Row-wise interaction)* | | |
| Layers | 3 | 3 |
| Model dimension | 128 | 128 |
| Attention heads | 8 | 8 |
| `[CLS]` tokens | 4 | 4 |
| *TF$_{icl}$ (In-context learning)* | | |
| Layers | 12 | 12 |
| Model dimension | 512 | 512 |
| Attention heads | 8 | 8 |
| *Target embedding* | | |
| Embed$_{TAE}$ | Lookup (10 classes) | Linear |
| Embed$_{ICL}$ | Lookup (10 classes) | Linear |
| *Prediction head* | | |
| Hidden dimension | 1024 | 1024 |
| Output dimension | 10 | 999 |
| *Other settings* | | |
| LayerNorm bias | Yes | No |
| FFN expansion | 2× | 2× |
| Activation | GELU | GELU |

# B. More Pretraining Details About TabICLv2

## B.1. Three pretraining stages

Following TabICL (Qu et al., 2025), we adopt a three-stage pretraining curriculum that progressively increases the sample size of synthetic datasets with batch size of 64 as follows:

- **Stage 1**: 500K steps with 1,024 samples per dataset.
- **Stage 2**: 40K steps with 400–10,240 samples (log-uniform).
- **Stage 3**: 10K steps with 400–60,000 samples (log-uniform).

Figure B.1 shows the performance of TabICLv2 after each stage on the TALENT benchmark. Each stage yields consistent improvements: on all datasets (Figure B.1a), average rank improves from 9.94 (Stage 1) to 5.69 (Stage 2) to 5.41 (Stage 3). The gains are most pronounced on large datasets with more than 10K samples (Figure B.1c): Stage 1 achieves only rank 14.91, comparable to XGBoost (14.60), but Stage 2 dramatically improves to 5.50 and Stage 3 further reaches 4.71, substantially outperforming all baselines including RealTabPFN-2.5 (6.35). This demonstrates that exposure to larger synthetic datasets during pretraining is crucial for generalization to real-world large-scale datasets.

## B.2. Speed and memory optimization

Automatic mixed precision is always used. For stage 3, we enable gradient checkpointing when the sample size exceeds 20K to avoid the out-of-memory error. For stages 2 and 3, we use FlashAttention-3 (Shah et al., 2024), which provides an average 1.3× speedup over FlashAttention-2 (Dao, 2024) on large-scale pretraining.

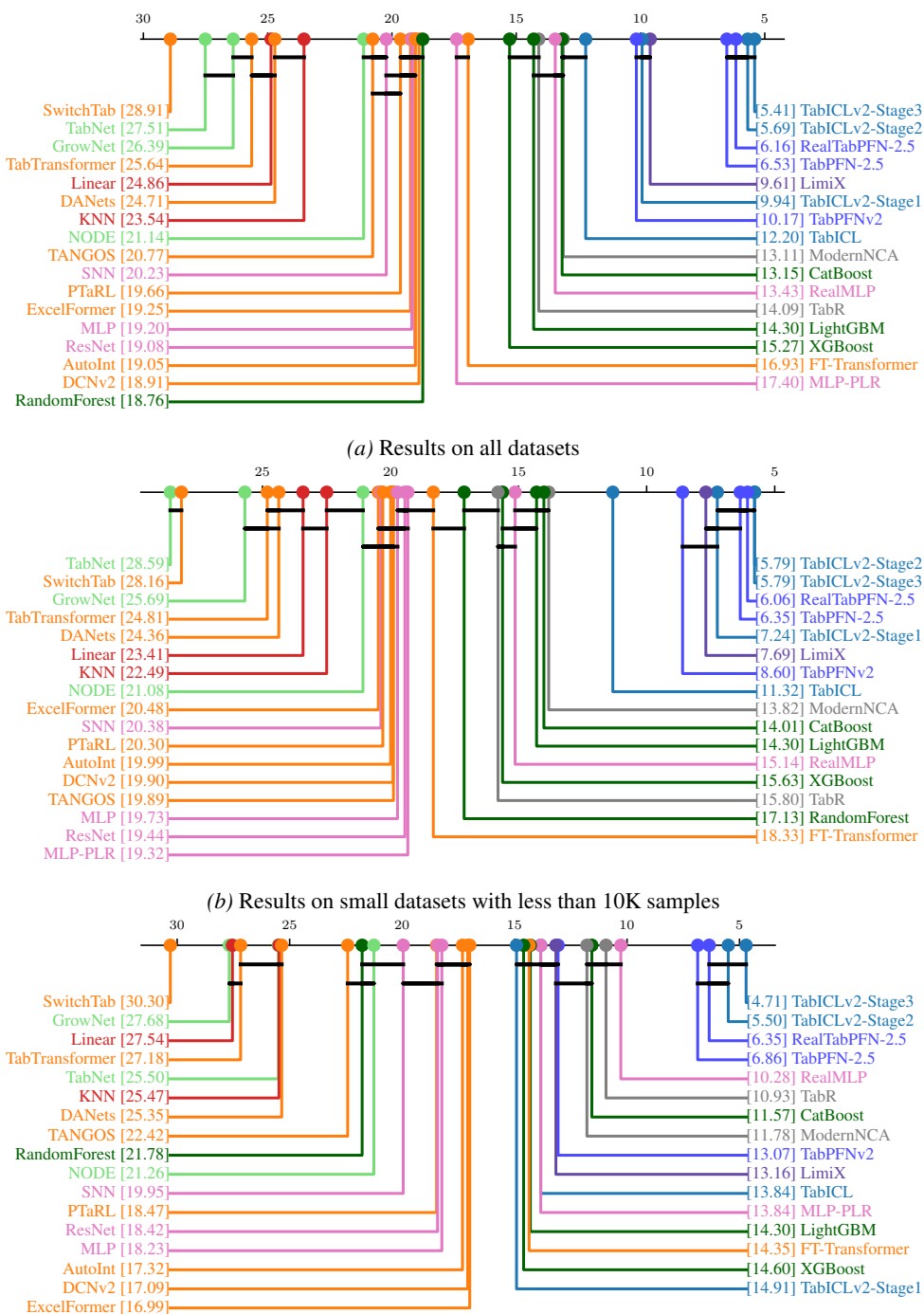

*(a)* Results on all datasets

*(b)* Results on small datasets with less than 10K samples

*(c)* Results on large datasets with more than 10K samples

*Figure B.1.* **Critical difference diagram on the TALENT benchmark for TabICLv2 pretrained on three stages.**

# C. Additional Ablation Results

In the main text (Section 7), we present ablation results based on log-loss. Here, we provide additional results using normalized accuracy (Figure C.1a) and Elo (Figure C.1b). Across all three metrics, the ordering of ablations remains consistent, leading to the same conclusions about the contribution of each component.

We additionally perform two more ablation experiments:

**Increasing model depth.** We ablate the effect of increasing model depth (light red line in Figure C.1): 4 layers for $TF_{col}$ and $TF_{row}$ (instead of 3) and 18 layers for $TF_{icl}$ (instead of 12). Based on log-loss, this deeper model shows no clear improvement over the reference model. However, normalized accuracy and Elo suggest a slight improvement toward the end of pretraining. This marginal gain is likely due to insufficient pretraining for the larger model to fully converge. Nonetheless, since our goal is to achieve state-of-the-art performance through principled innovations rather than simply scaling up model size, we did not pursue this direction further.

**Adding noise to the prior.** Following TabPFNv2 (Hollmann et al., 2025), which adds Gaussian noise at each edge of the causal graph during synthetic data generation to introduce uncertainty, we experimented with incorporating similar noise into our prior (green solid line in Figure C.1). However, this modification has negligible impact on performance across all metrics.

# D. Other Things We Tried

Here, we want to describe some other things that we tried but did not end up using, mostly in smaller-scale experiments and without careful analysis. We hope that it can serve as anecdotal evidence to other model developers. Generally, the results of pretraining runs are somewhat noisy, so these observations have to be taken with at least one grain of salt. Reducing the pretraining noise could itself be a useful contribution of future research.

**Pretraining.** Contrary to Ma et al. (2025), we did not see a benefit from using schedule-free AdamW (Defazio et al., 2024) over regular AdamW with a cosine schedule. We found some benefit from AdEMAMix (Pagliardini et al., 2025) in small-scale runs compared to AdamW, but it performed worse than Muon, and a combination of both did not seem to help. For AdamW, decreasing $\beta_2$ showed improvements at least for shorter runs. The comparisons between cautious weight decay, weight decay, and no weight decay were not very clear; we went with cautious weight decay due to its inclusion in the NanoGPT speedrun (Jordan et al., 2024a). Since Ma et al. (2025) used label smoothing but this can hurt the performance on metrics like log-loss, we tried a label smoothing schedule that decays to zero at the end of training, but it resulted in equal performance.

**Architecture: embeddings.** We did not see benefits from using MLPs instead of linear layers for embedding $x$. Adding $\log(n)$ together with $x$ was not helpful in small runs either. Surprisingly, using regular column-wise attention instead of ISAB performed worse in some runs. Mixing the layers of the column- and row-attention stages did not seem beneficial, and it can be a disadvantage since it requires more transposes and yields a less optimized CPU- and disk-offloading. Even with such mixing, it did seem worse to have a separate column for $y$ instead of adding the embedding of $y$ to the embeddings of the columns $x_i$.

**Architecture: row interaction.** It seems that the full row-wise attention (attention across columns, within a row) is important, replacing it with induced self-attention performed considerably worse. We experimented a bit with random feature identifiers in the version used by LimiX (Zhang et al., 2025), but it was unclear if they are beneficial.

**Architecture: normalizations.** We did not see improvements from different placements of normalization layers (though the experiments used more shallow nets). Additionally, bias-free or parameter-free layernorms seemed to perform similar to full layernorms while being faster, but we were not very confident in whether these measurements are good enough.

**Architecture: other.** TabPFNv2-type architectures seemed to perform well, and we have no clear conclusion in which situations they are better or worse than the TabICLv2 architecture. Due to the higher runtime complexity and per-step time of the TabPFNv2 architecture, we discarded it. Experiments with residual connections did not show much difference in the

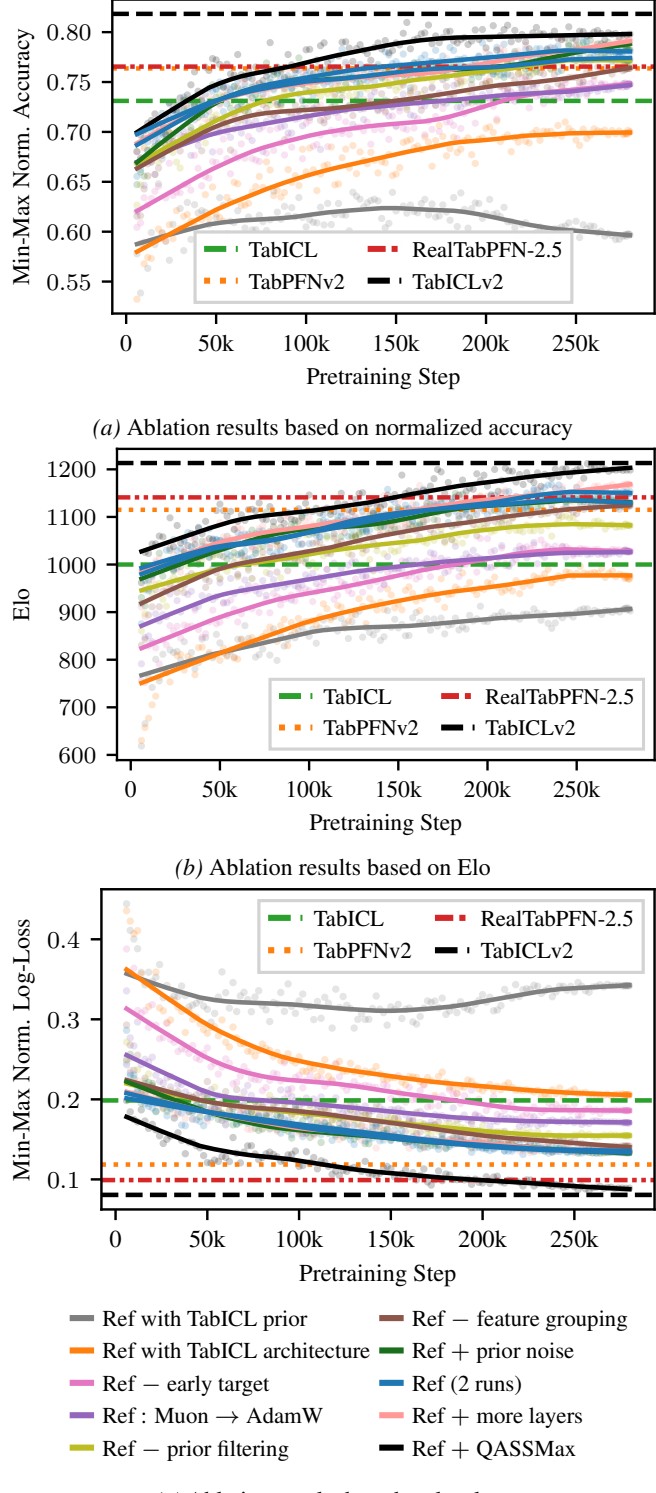

*(a)* Ablation results based on normalized accuracy

*(b)* Ablation results based on Elo

*(c)* Ablation results based on log-loss

*Figure C.1.* **Ablation results using different metrics.**

results. Increasing the number of attention heads from $4$ to $8$ in $\text{TF}_{\text{col}}$ and $\text{TF}_{\text{icl}}$ seemed slightly beneficial in small-scale runs with a smaller model, but not necessarily in large-scale runs.

**Other things.**  For regression (judged by MSE), we found incremental spline quantile functions (ISQF, Park et al., 2022) to work similarly well as regular quantile regression, but discarded them due to their computational (and conceptual) overhead. While LimiX (Zhang et al., 2025) included convolution-based functions in their prior, we did not see a measurable benefit from including these functions into our prior, at least when fine-tuning TabICL on our prior. TabICL uses a masking mechanism to deal with micro-batches in which some datasets have fewer columns, where the rest of the columns are filled up with zeros. We did not implement this and the pretraining still worked well.

# E. Details on the Prior

In the following, we describe left-out details from the prior description in Section 5, with a focus on a more implementation-related description. To keep the prior modular, we decompose it into sampling methods for different objects: random datasets, random functions, random points, random matrices, and so on. These will be described in the following subsections (which may sometimes refer to components introduced by later subsections). Some of the components are visualized in Appendix F.

## E.1. Differences to previous priors

The closest prior to ours is probably the TabPFNv2 prior (Hollmann et al., 2025), though not all details about it are known. However, our prior still differs from it in many ways by introducing new mechanisms:

- We introduce a new correlated scalar sampling mechanism (Appendix E.2).

- We introduce a new random graph sampling mechanism (Appendix E.4).

- We introduce additional computations at each node that create random node and feature importances (Appendix E.5).

- We make the extraction of numerical and categorical features more precise through the introduction of random converters (Appendix E.6) and provide more variants for categorical converters.

- We explicitly introduce multiple ways to apply random functions in the case of multiple parent nodes (Appendix E.7; it is unclear if this case can occur in TabPFNv2).

- We introduce new random function types and diversify existing ones (Appendix E.8). In particular, for tree-based functions we do not only use single trees as in TabPFNv2. Instead, we use an ensemble of CatBoost-style symmetric trees which facilitate efficient computations. Our GP functions are extended and multivariate versions of the random GP activations from TabICL and come with a detailed theoretical analysis (Appendix G). We also introduce random linear, quadratic, clustering-based, and product functions.

- We add more random activations (Appendix E.9).

- While TabPFNv2 uses random Gaussian matrices, we introduce four additional random matrix types (Appendix E.10).

- We introduce random weights that are useful in multiple places, be it for sampling correlated categoricals, feature importances, or singular values (Appendix E.11).

- We introduce more mechanisms to sample random points, by applying a random function to different kinds of base distributions (Appendix E.12).

- We introduce a filtering mechanism similar to Dong et al. (2025) in Appendix E.14.

- In addition, for scalar random variables like the number of categories etc., we generally use different choices of distributions, either to fit our framework, or because they are not known for TabPFNv2.

### E.2. Sampling correlated scalars

When sampling scalar numerical values within the prior, we want some of them to be correlated, since we expect some properties to be correlated in real datasets. For example, correlated quantities could include the cardinalities of different categorical columns. To know which values should be correlated, we assign names to them such as "categorical_cardinality". All values with that name are sampled from the same distribution, whose parameters are themselves sampled once for every name.

**Mechanism:** For each variable name, we sample $t \sim \text{Uniform}[0, 1]$ and $s \sim \text{LogUniform}[0.1, 10000]$ and set $\alpha := st, \beta := s(1 - t)$. For each time a variable should be sampled for that name, a base variable is sampled as $u \sim \text{Beta}(\alpha, \beta)$. Then, it is affinely transformed to a desired range, the exponential is taken for log-type distributions, and it is rounded down for integer-valued random variables. We denote uniform-like real-valued and integer-valued random variables with bounds $a, b$ by $\text{Num}(a, b), \text{Int}(a, b)$ and log-uniform-like versions as $\text{LogNum}(a, b), \text{LogInt}(a, b)$.

For categorical values (like the random function type used at each node), we intended to implement correlated sampling, but it was not used due to a bug.

### E.3. Random dataset

We first sample some general characteristics of the dataset. For classification, we sample the number of classes from $\text{UniformInt}(2, 10)$. The ratio of categorical columns is sampled from $\text{Uniform}(-0.5, 1.2)$ and subsequently clipped to $[0, 1]$. The categorical cardinalities are sampled from $\text{LogInt}(2, M)$, where the maximum cardinality is sampled once as $M \sim \text{LogInt}(2, 9)$, and a uniform random fraction of them is sampled through the correlated sampling mechanism. The total number of columns is configurable but is sampled from $\text{UniformInt}(2, 100)$ in our training runs.

We then sample a random graph with $\text{LogInt}(2, 32)$ nodes. To assign the different target columns to nodes in the randomly sampled graph, for each column type (either input columns for $x$, or the target column $y$), we sample the number of eligible nodes uniformly, then sample a subset of nodes of that size, and then assign each column to a random node in that subset. The graph sampling and node assignment is potentially repeated until the graph filtering mechanism accepts it. The nodes are then traversed in topological order and their corresponding random node functions are called with the data from all parent nodes as input. Nodes that are not needed for the final dataset computation are pruned. Finally, the obtained dataset is shuffled randomly and split into train and test parts.

### E.4. Random graph

For node indices $i < j$, an edge is placed with probability

$$p_{ij} = \text{sigmoid}(A + B_i + C_j),$$

where $A, B_i, C_j$ are independent standard Cauchy random variables. $A$ controls the general level of connectivity, while $B_i$ and $C_j$ control the individual outgoing and incoming connectivities of the nodes, respectively. Compared to independent Bernoulli probabilities, this model yields more diverse connectivity patterns. Cauchy random variables have heavy tails and therefore yield higher probabilities of "exceptions to the rule".

### E.5. Random node function

We first obtain a matrix $X \in \mathbb{R}^{n \times d_i}$ from a random multi-function applied to the parent node data, or from the random points mechanism if there are no parents. Here, $d_i = \sum_j d_{ij} + \text{LogInt}(1, 32)$, where the $d_j$ are the dimensions required by the random converters for extracting the dataset columns from the node. A random converter can extract a column and also modify (e.g., discretize) the corresponding portion of the node data. We then standardize the columns of $X$. Then, we sample a weights vector $w \in \mathbb{R}^{d_i}$ and multiply each column of $X$ by the corresponding weight. Afterwards, we divide $X$ by the average (over samples) $L^2$ norm of the vectors. The motivation to use the $L^2$ norm instead of the RMS norm is to keep the vector norms small in high dimensions, such that high-dimensional functions do not become too difficult to learn. Now, we apply the random converters for the assigned columns to their respective $d_{ij}$ dimensions, updating the respective part of $X$ with their output. Finally, in the "random rescale" step, we multiply $X$ by a scalar $\text{LogNum}(0.1, 10)$.

### E.6. Random converter

We introduce converters

$$X', v = \text{converter}(X), \quad X, X' \in \mathbb{R}^{n \times d}, v \in \mathbb{R}^n \,,$$

which extract a column $v$ for the generated dataset while also potentially modifying the node data $X$ to $X'$.

**Numerical converters:** We set $v = X$, $d = 1$, and choose $X' = f(X)$, where $f$ is sampled as the identity or a warping function based on a Kumaraswamy distribution (Kumaraswamy, 1980) after min-max scaling, following Hollmann et al. (2025). For the Kumaraswamy warping, we min-max scale values $x$ to the range $[0, 1]$ and then compute $1 - (1 - x^a)^b$ with $a, b \sim \text{LogNum}(0.2, 5)$. The Kumaraswamy warping is unintentionally applied to compute $x'$ instead of $v$.

**Categorical converters:** Let $c \in \mathbb{N}$ be the number of desired categories. We use two main approaches:

- **Neighbor-based**: We choose a random subset of $c$ points from the data. As in the RandomDiscretizationFunction, we then map each point $x$ to its closest point in the subset as measured by the $L^p$ distance, $p = \text{LogNum}(0.5, 4)$. The index of the closest center is the class index. For neighbor-based approaches, we sample the desired dimension $d$ of $x$ as $c$ with probability $1/2$ and $\text{Int}(1, c - 1)$ otherwise.

- **Softmax-based**: We sample the category from

$$\text{softmax}(a\tilde{x} + b) \,,$$

  where $\tilde{x}$ is a standardized version of the input $x \in \mathbb{R}^c$, $a \sim \text{LogNum}(0.1, 10)$, and $b = \log(w + 10^{-4})$ with $w \in \mathbb{R}^c$ being a random weight vector. The variation in $a$ can create different levels of separation between categories, and the variation in $b$ can create different levels of imbalance. For softmax-based approaches, we always need the dimension of $x$ to be $d = c$.

We further distinguish different approaches to compute the transformed node vector $x'$:

- Output the input $x$ (neighbor- or softmax-based)

- Output the category index $i$, repeated to get a $d$-dimensional vector (neighbor- or softmax-based).

- Output the closest center (neighbor-based) or a random function applied to the closest center (neighbor-based).

- Sample random points $\{z_1, \ldots, z_c\}$, then output $z_i$ where $i$ is the category index (softmax-based).

In total, we obtain seven combinations of categorical converters, of which we sample one randomly.

### E.7. Random multi-function

If there is only a single input node, we use a random function (see below). Otherwise, if there are $n_{\text{in}}$ input nodes, we proceed as follows: With probability $1/2$, we concatenate the tensors of all input nodes along the features dimension and apply a random function to it. Else, we apply separate random functions to each input node, obtaining $n_{\text{in}}$ tensors of dimension $n \times d$ that are aggregated along the $n_{\text{in}}$ axis using one of the following four element-wise aggregations: sum, product, max, or logsumexp.

### E.8. Random functions

**RandomNNFunction** A random NN with $\text{LogInt}(1, 3)$ linear layers, hidden width $\text{LogInt}(1, 127)$ (drawn once per NN), and a 50% chance each of including an activation at the beginning or end of the network. The linear layers use RandomLinearFunction (no bias, but there is a bias in the activation function).

**RandomTreeFunction** Generates $\text{LogInt}(1, 128)$ trees, of depth $\text{Int}(1, 7)$. Each tree is symmetric (= oblivious), meaning that it uses the same splitting criterion for all nodes on the same layer. The split dimension is chosen randomly with probability proportionally to the standard deviation of data in that dimension, to respect the feature importances that were randomly sampled on the input nodes. The split points are random samples from the arriving data in the respective dimension. The leaf values are standard normal random values. Then, each tree is evaluated for the data and the corresponding leaf values are averaged.

**RandomDiscretizationFunction**   Chooses a subset of samples from $X$ as centers, with the number of samples being $\text{LogInt}(2, 255)$. It then maps each point $x$ to its closest center as measured by the $L^p$ distance, $p = \text{LogNum}(0.5, 4)$, and applies a random linear function to the result.

**RandomGPFunction**   This function computes $f(Mx) = (f_1(Mx), \ldots, f_d(Mx))$, where each component $f_i$ is sampled from a Gaussian process with a random kernel $k$ shared between all $i$. Here, $M_{ij} = \alpha w_i A_{ij}$ with random weights vector $w$, random scale $\alpha \sim \text{LogNum}(0.5, 10)$, and a random Gaussian matrix $A$. This choice is inspired by the random GP activations in TabICL as well as the success of tuning over different kernels with different bandwidths (scales) and learnable linear input transformations in xRFM (Beaglehole et al., 2026).

The first question is how to design the random kernels $k$. In order to use a random Fourier features approximation (Rahimi & Recht, 2007), we design $k$ directly in the Fourier domain. Suppose that $g : \mathbb{R}^d \to \mathbb{R}_{\geq 0}$ is integrable and even ($g(x) = g(-x)$ for all $x$). Then $k(x, x') = \breve{g}(x - x')$ is a real-valued kernel on $\mathbb{R}^d$ (see e.g. Lemma G.2), where $\breve{g}(x) = \int e^{i\langle x, \omega \rangle} g(\omega) \, d\omega$ is the inverse Fourier transform of $g$. In Appendix G, we show that the tail behavior of $g$ is directly related to the smoothness of functions sampled from the GP: If there exist constants $c, C, r_0 > 0$ such that

$$c\|\omega\|^{-q} \leq g(\omega) \leq C\|\omega\|^{-q} \qquad \text{whenever } \|\omega\| \geq r_0, \tag{E.1}$$

then the sample paths (sampled functions) from the GP essentially have smoothness $(q - d)/2$, at least when $q > 2d$. If $g$ is a probability density function (integrates to 1), we can approximate $k$ using random Fourier features (Rahimi & Recht, 2007):

$$k(x, x') \approx \phi(x)^\top \phi(x'), \quad \phi(x) = \sqrt{2/p} \cos(Wx + b) \in \mathbb{R}^p \, ,$$

where the rows of $W$ are independently drawn from $g$ and $b_i \sim \text{Unif}[0, 2\pi]$ i.i.d. The dimension $p$ can be chosen arbitrarily large to improve the approximation quality; we follow Qu et al. (2025) and set $p = 256$.

As argued in Qu et al. (2025), for a standard normal random vector $z \in \mathbb{R}^p$, $z^\top \phi(x)$ follows a Gaussian process with kernel $\phi(x)^\top \phi(x')$, which therefore approximates the Gaussian process with kernel $k$. For a general multi-output setting, we therefore sample $Z \in \mathbb{R}^{d_{\text{out}} \times p}$ with i.i.d. standard normal entries and compute (omitting the factor $\sqrt{2}$ which only rescales the output)

$$f(Mx) = p^{-1/2} Z \phi(Mx) = p^{-1/2} Z \cos(WMx + b) = p^{-1/2} Z \cos((W \text{diag}(w) A)x + b) \, ,$$

with $w, A$ from the beginning of the explanation.

It remains to find a probability density function $g$ for a given $q$ from which we can efficiently sample. Since we can choose a rotation-invariant distribution, we can just sample it as $\omega = rz/\|z\|$, where $z \in \mathbb{R}^d$ is a standard normal vector and $r$ controls the radial component of $\omega$. For densities with tail $\Theta(\|\omega\|^{-q})$, integration in spherical coordinates yields that the tail of the density of $r = \|\omega\|$ must behave like $\Theta(r^{d-1} r^{-q}) = \Theta(r^{d-1-q})$.

We construct a family of 1D distributions with power-law tails that are easy to sample from: For $a > 1$, we define a CDF of $H_a(r) = 1 - (1 + r)^{1-a}$ (for $r \geq 0$). The associated PDF is $h_a(r) = H_a'(r) = (a - 1)(1 + r)^{-a}$. We can then sample from $H$ using inverse CDF sampling: For $u \sim \text{Unif}[0, 1]$,

$$H_a^{-1}(u) = (1 - u)^{\frac{1}{1-a}} - 1$$

is distributed according to $H_a$. We sample $a \sim \text{LogNum}(2, 20)$, corresponding to $q = a + d - 1$, and sample $r$ from $H_a$.

With 50% probability, we choose **another way** to sample the kernel: Inspired by the choice of non-rotationally invariant axis-aligned kernels in xRFM (Beaglehole et al., 2026), we alternatively sample each entry of $\omega$ independently from $H_a$. This yields a product distribution $g(\omega) = g_1(\omega_1) \cdot \ldots \cdot g_d(\omega_d)$ for $\omega$, whose inverse Fourier transform $\breve{g}$ yields a kernel that is a product of one-dimensional kernels, as for the case "$p = q$" in xRFM (Beaglehole et al., 2026). We do not explicitly prove a result about the path smoothness for product kernels, but a similar argument to Theorem G.1 in conjunction with Theorem 4.2 of Steinwart (2024) should yield that for $a > 2$, the paths are contained in Sobolev spaces of dominating mixed smoothness of order $s$ whenever $s < (a - 1)/2$. As in the other case, we sample $a \sim \text{LogNum}(2, 20)$. Since the product kernel constructed in this way is axis-aligned, we do not apply the random matrix $M$ in the construction above to preserve axis-alignment.

**RandomLinearFunction**   Samples a random matrix (Appendix E.10) and multiplies each vector $x$ by this matrix.

**RandomQuadraticFunction**    Computes

$$f(x)_i = \sum_{j,k} M_{ijk} x_j x_k \ ,$$

where each $M_i$ is a random matrix, jointly sampled from the same random matrix type. To avoid quadratic complexity in the dimension of $x$, we first subsample $x$ to at most 20 dimensions if it has more than 20 dimensions. We include linear and constant terms by appending 1 to the vector $x$.

**RandomEMAssignmentFunction**    This function type is inspired by the computation of the probability $p_i$ that the input $x$ is from the cluster $i$ in the EM algorithm. However, we add some things like different $L^p$-norms and random powers $q$ without making sure that this corresponds to any real "cluster assignment" computation, simply for further increasing the diversity of computed functions.

First, a number $m = \text{LogInt}(2, \max\{16, 2d_{\text{out}}\})$ of "Gaussians" is sampled. Then, centers $\mu_1, \dots, \mu_m$ are chosen using random input vectors plus standard normal noise. Standard deviations $\sigma_1, \dots, \sigma_m$ are chosen independently as $\exp(0.1 \cdot \text{Normal}(0, 1))$. Then, logits are computed as

$$l_i(x) = -\frac{1}{2} \log(2\pi\sigma_i^2) - (\|x - \mu_i\|_p / \sigma_i)^q \ ,$$

where $p = \text{LogNum}(1, 4)$, $q = \text{LogNum}(1, 2)$. (The Gaussian case would be using $p = 2$, $q = 1$, and using $d/2$ instead of $1/2$ in the equation.) Finally, the output is computed as

$$L(\text{softmax}(l(x))) \ ,$$

where $L$ is a random linear function.

**RandomProductFunction**    Computes $f(x)g(x)$, where $f, g$ are two random functions (not product, NN, or EM, to optimize speed).

### E.9. Random activations

Following TabICL and TabPFNv2, we further expand the set of available activation functions. By activation functions, we mean functions $f : \mathbb{R}^d \to \mathbb{R}^d$ that preserve the input dimension, even if they are not element-wise.

We use them as follows inside a NN, expanding upon the standardization + random rescaling from TabICL:

- We first standardize (subtract the mean and divide by the standard deviation) along the batch dimension.

- We rescale randomly, as $x \leftarrow a(x - b)$, where $a = \text{LogNum}(1, 10)$ and $b$ is a random sample from the standardized data. We choose $b$ this way to avoid getting only zeros for activations like ReLU that are zero in a large portion of the space.

- Now, we apply the activation.

- Finally, we standardize again.

For the activation, with probability $2/3$ we pick one of the following fixed activations, otherwise one of the parametric activations below that.

**Fixed activations.**    As fixed activations, we use Tanh, LeakyReLU, Elu, Identity, SELU, SiLU, ReLU, softplus, ReLU6, HardTanh, signum, Heaviside, $\exp(-x^2)$, exp, $\mathbb{1}_{[0,1]}$, sin, square, abs, softmax, one-hot argmax, argsort, logsigmoid, $\log(\max(|x|, 10^{-6}))$, rank, sigmoid, round, modulo 1.

**Parametric activations.**    We introduce the following activations with random parameters:

- $\text{ReLU}(x)^q$ with $q \sim \text{LogNum}(0.1, 10)$.

- $\text{sign}(x)|x|^q$ with $q \sim \text{LogNum}(0.1, 10)$.

- $(|x| + 10^{-3})^{-q}$ with $q \sim \text{LogNum}(0.1, 10)$.

- $x^m$ with $m \sim \text{Int}(2, 5)$.

The latter two were only used for the random activation matrix.

### E.10. Random matrix

We randomly sample a matrix from one of the following five types:

RandomGaussianMatrix: Consists of i.i.d. standard normal entries.

RandomWeightsMatrix: To sample a matrix of shape $m \times k$, we compute $M_{ij} = W_{ij} \odot G_{ij}$, where $G$ is a random Gaussian matrix and $W_{i,\cdot}$ are random weight vectors (which are in general correlated through the correlated sampling mechanism). Afterwards, the rows of $M$ are normalized (divided by their norm).

RandomSingularValuesMatrix: To sample a matrix of shape $m \times k$, we compute $U \text{diag}(w) V^\top$, where $w \in \mathbb{R}^{\min\{m,k\}}$ is a random weights vector and $U, V$ are random Gaussian matrices of suitable shape. While sampling orthogonal $U, V$ would mean that we explicitly sample the singular value decomposition, using Gaussian $U, V$ is faster and still produces a rotation-invariant distribution (since for Gaussian $U$ and arbitrary orthogonal matrix $R$, the distribution of $U$ is the same as the distribution of $UR$ or $RU$).

RandomKernelMatrix: To sample a matrix of shape $m \times k$, we sample $k + m$ random covariance points $x_1, \ldots, x_{k+m} \in \mathbb{R}^3$ and a scaling factor $\gamma \sim \text{LogNum}(0.1, 10)$, then create the Laplace kernel matrix $K_{ij} = \exp(-\gamma \|x_i - x_{m+j}\|)$ and multiply each entry by an independent random sign (a number in $\{-1, 1\}$).

RandomActivationMatrix: After sampling a matrix from one of the other types, we apply a random activation to the flattened matrix, then add Gaussian noise with standard deviation $10^{-3}$. Unlike for the NN, we omit the standardization and random rescaling in the activations, since the batch size is 1, preventing standardization.

**Postprocessing.** After creating a matrix using one of the described types, we add 1e-6 times a random Gaussian matrix and normalize each row of the resulting matrix. This prevents all-zero rows that could arise from some activation functions in the RandomActivationMatrix.

### E.11. Random weights

To emulate random feature importances, singular value decays, or (unnormalized) probability distributions, we introduce a dedicated way to sample random positive vectors $w \in \mathbb{R}^d_{>0}$. We first generate

$$w_m = m^{-q} \cdot \exp(\text{Normal}(0, \sigma^2)), \quad 1 \le m \le d$$

with $q \sim \text{LogNum}(0.1/\log(d+1), 6)$ and $\sigma \sim \text{LogNum}(10^{-4}, 10)$. The lower and upper bounds for $q$ are chosen such that we can sample vectors where no weights are close to zero as well as vectors where almost all weights are close to zero. Finally, we normalize and shuffle $w$.

### E.12. Random points

Generating a matrix $X \in \mathbb{R}^{n \times d}$ of random points is in principle the same problem as generating a random dataset with numerical columns, but here we only use cheaper mechanisms (and avoid infinite recursions). First, we sample either standard normal points, uniform on $[-1, 1]^d$, uniform on the unit ball, or random covariance points. Then, we apply a random function to these points. The random covariance points are sampled as follows: For a given dimension $d$, sample $x \in \mathbb{R}^d$ from a standard normal or uniform on $[-1, 1]^d$ distribution, then compute $A(w \odot x)$, where $\odot$ is the elementwise product, $w \in \mathbb{R}^d$ are random weights, and $A$ is a random Gaussian matrix.

### E.13. Postprocessing

Following TabICL, columns with a single value are removed. Datasets are discarded if all columns were removed or less than 2 classes are present or the train and test splits cannot be fixed to contain the same classes. We ordinal-encode categoricals.

For all columns $x_i$ and in the regression case also $y$, we remove outliers, then standard-scale. We permute the column order and the class indices (but not the categorical indices).

### E.14. Filtering

The ExtraTrees-based filtering works as follows: We convert classification problems to regression using one-hot encoding, to unify the two cases. To obtain very fast filtering, we then fit an ExtraTreesRegressor (Geurts et al., 2006) from scikit-learn (Pedregosa et al., 2011) with `n_estimators=25`, `bootstrap=True`, and `max_depth=6` on the full dataset. We then test whether the out-of-bag predictions can achieve a lower MSE than the mean label by checking if this is the case on at least 95% out of 200 bootstrap subsamples. If it is not the case, the dataset is rejected.

## F. Plots for the Prior

Figure F.1 shows random datasets from the prior. Random function types are shown in Figures F.2, F.3, F.4, F.7, F.5, F.6, F.8, and F.9. Random graphs are shown in Figure F.10. Figure F.11 shows random points. Random matrices are visualized in Figures F.12, F.13, F.14, F.15, F.16.

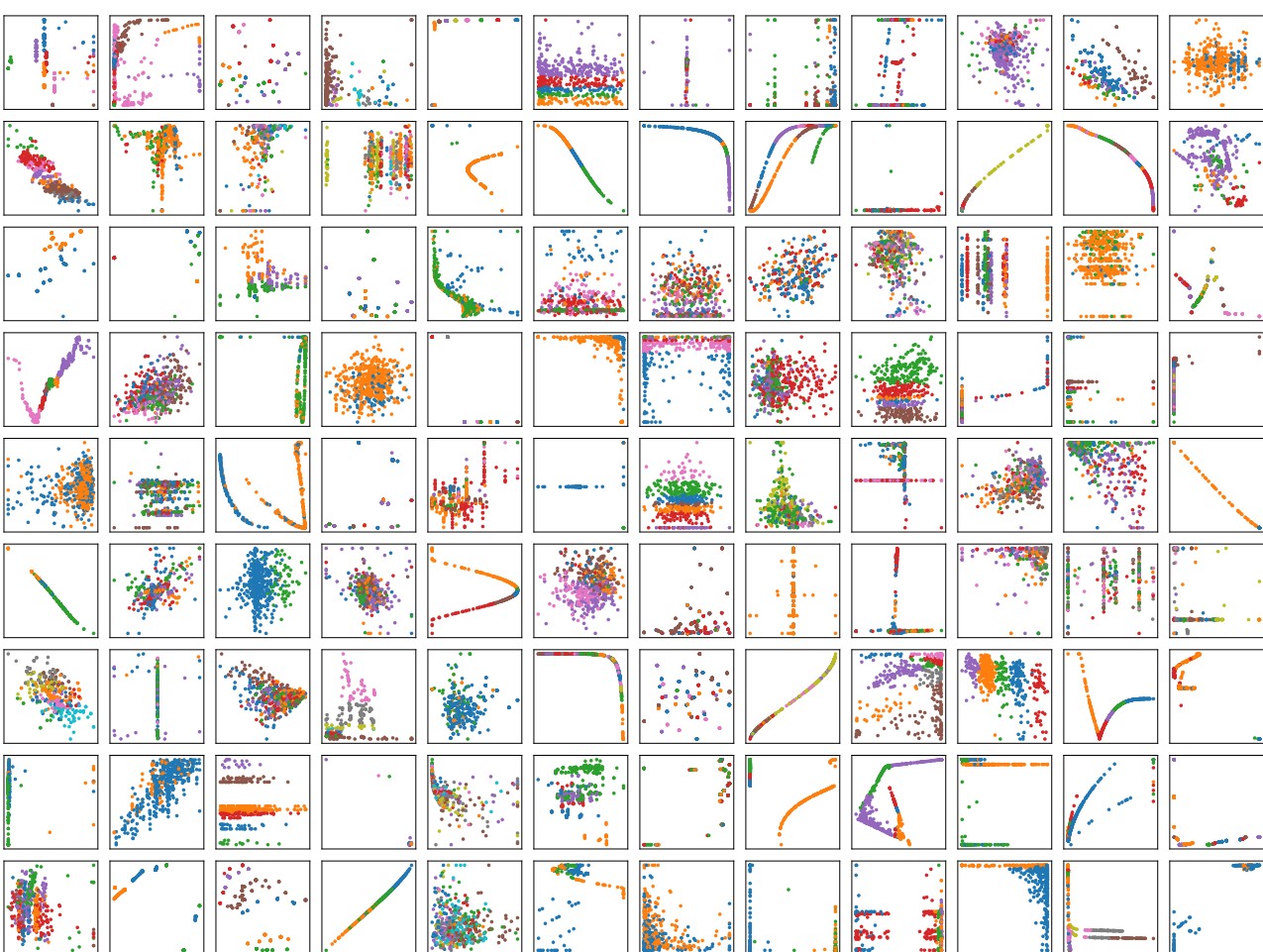

*Figure F.1.* **Random classification datasets from the prior.** Datasets contain 500 samples and two columns in $x$. The color shows the class label. Only datasets containing at least 10 unique values on each axis are shown (for visualization purposes, since otherwise the points can overlap a lot).

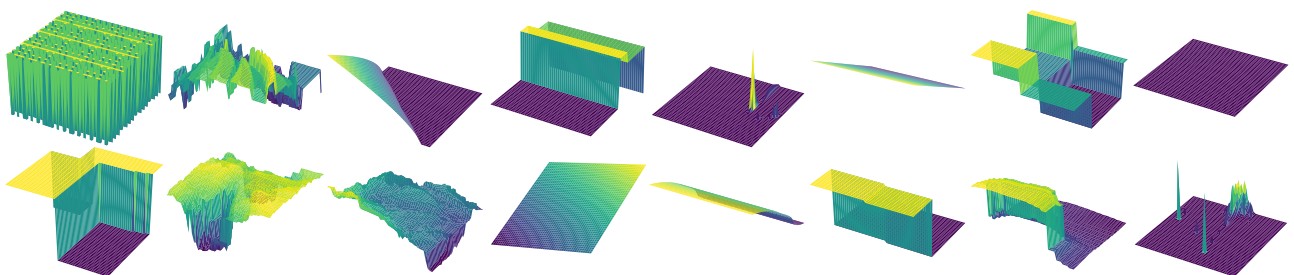

*Figure F.2.* **Samples of RandomNNFunction.** We use inputs from $[-3, 3]^2$ and one-dimensional output.

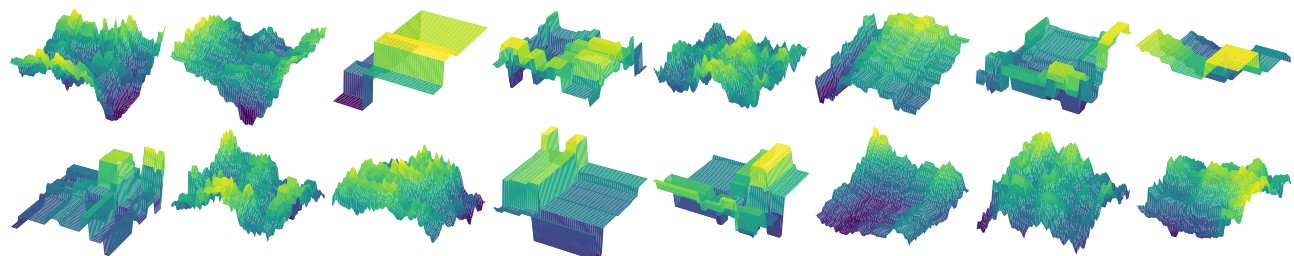

*Figure F.3.* **Samples of RandomTreeFunction.** We use inputs from $[-3, 3]^2$ and one-dimensional output.

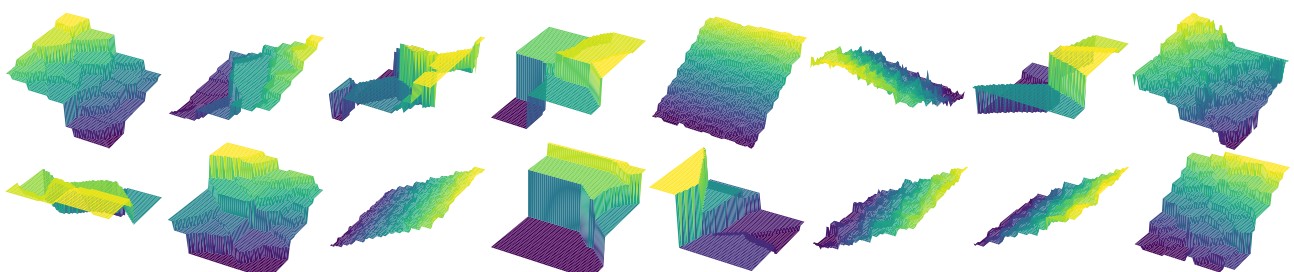

*Figure F.4.* **Samples of RandomDiscretizationFunction.** We use inputs from $[-3, 3]^2$ and one-dimensional output.

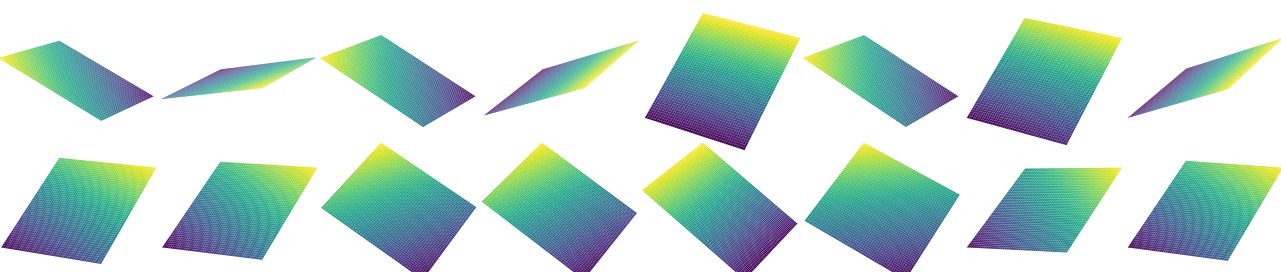

*Figure F.5.* **Samples of RandomLinearFunction.** We use inputs from $[-3, 3]^2$ and one-dimensional output.

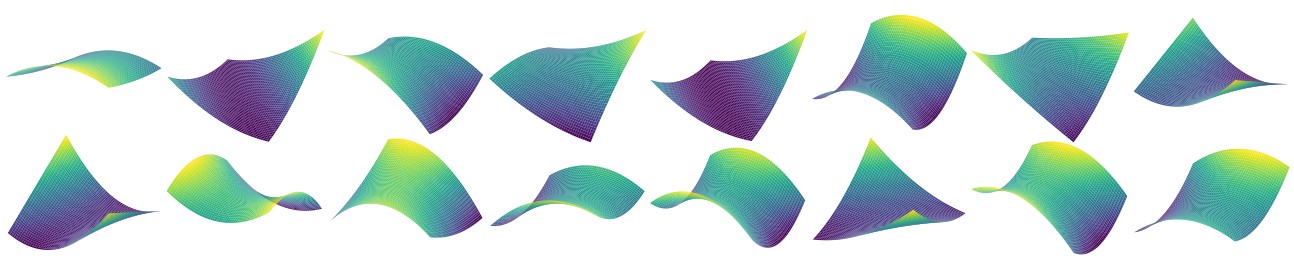

*Figure F.6.* **Samples of RandomQuadraticFunction.** We use inputs from $[-3, 3]^2$ and one-dimensional output.

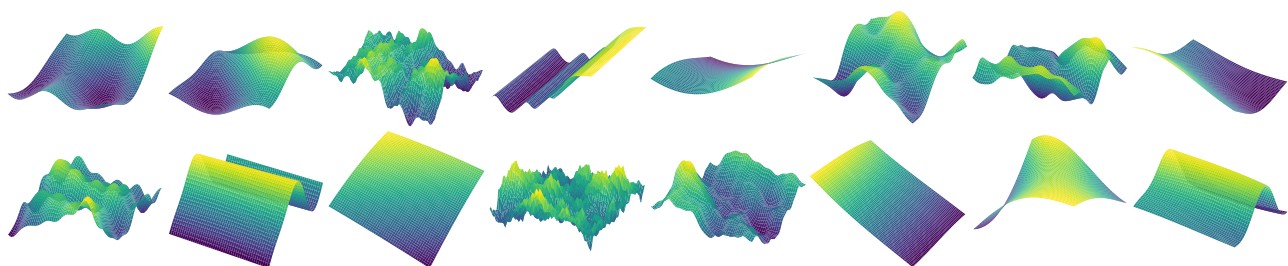

*Figure F.7.* **Samples of RandomGPFunction.** We use inputs from $[-3, 3]^2$ and one-dimensional output.

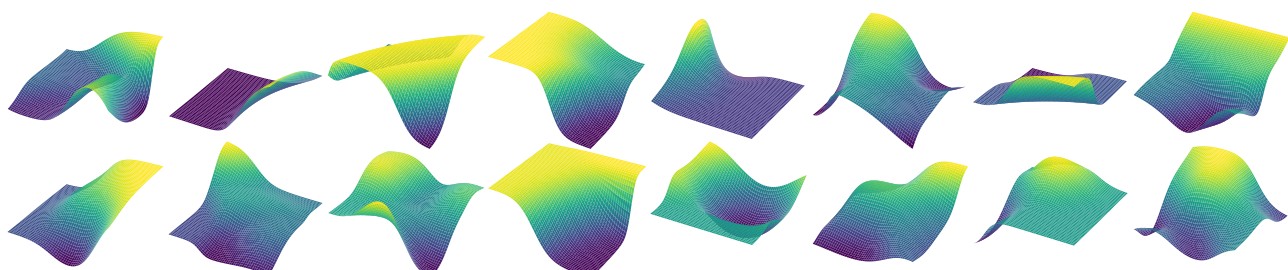

*Figure F.8.* **Samples of RandomEMAssignmentFunction.** We use inputs from $[-3, 3]^2$ and one-dimensional output.

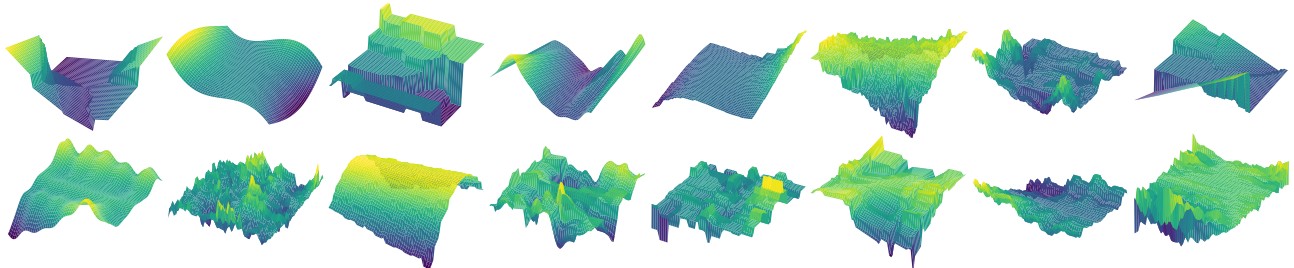

*Figure F.9.* **Samples of RandomProductFunction.** We use inputs from $[-3, 3]^2$ and one-dimensional output.

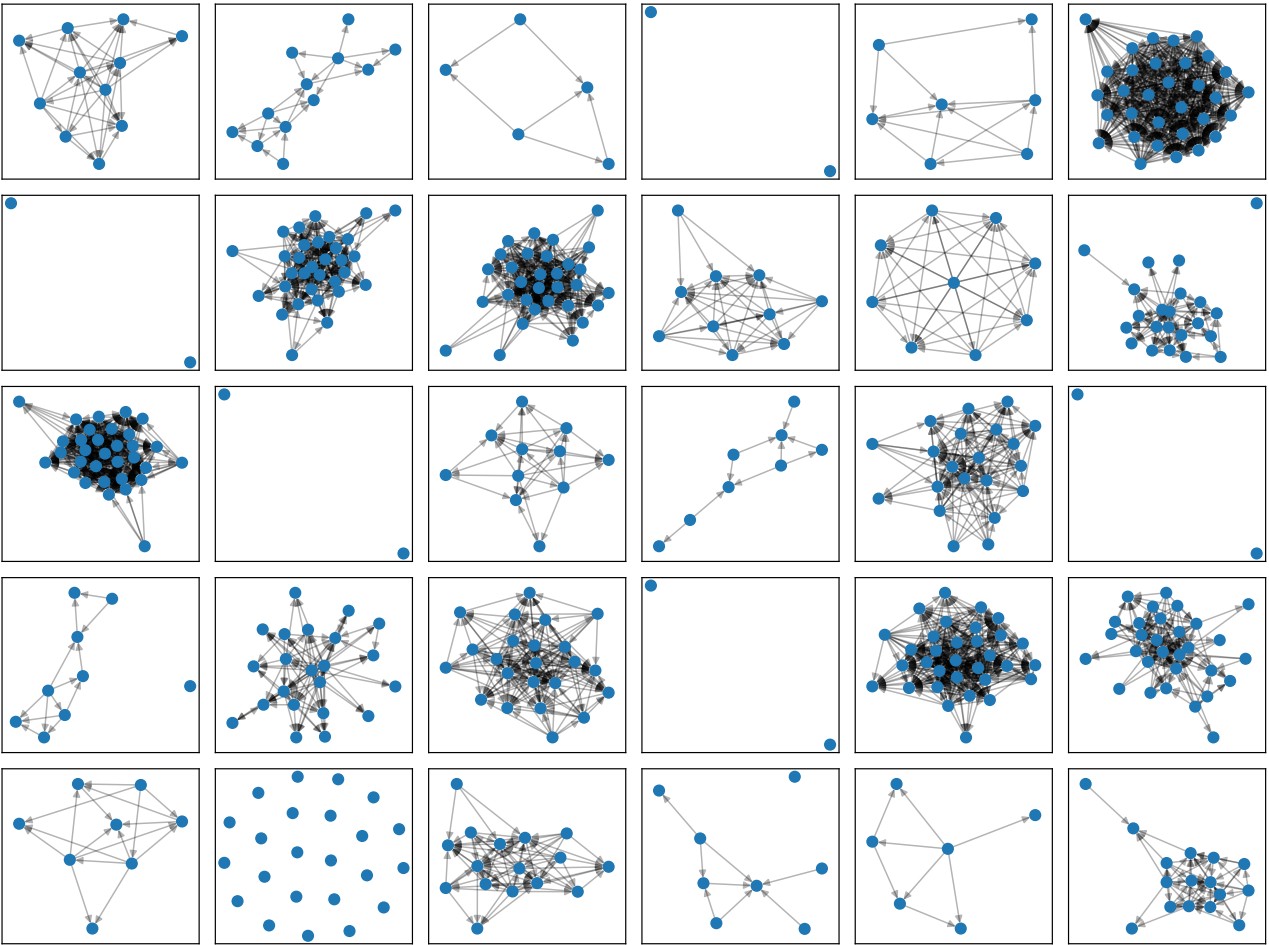

*Figure F.10.* **Randomly sampled graphs.** Graphs are not filtered (this would require knowing the assignment of columns to nodes).

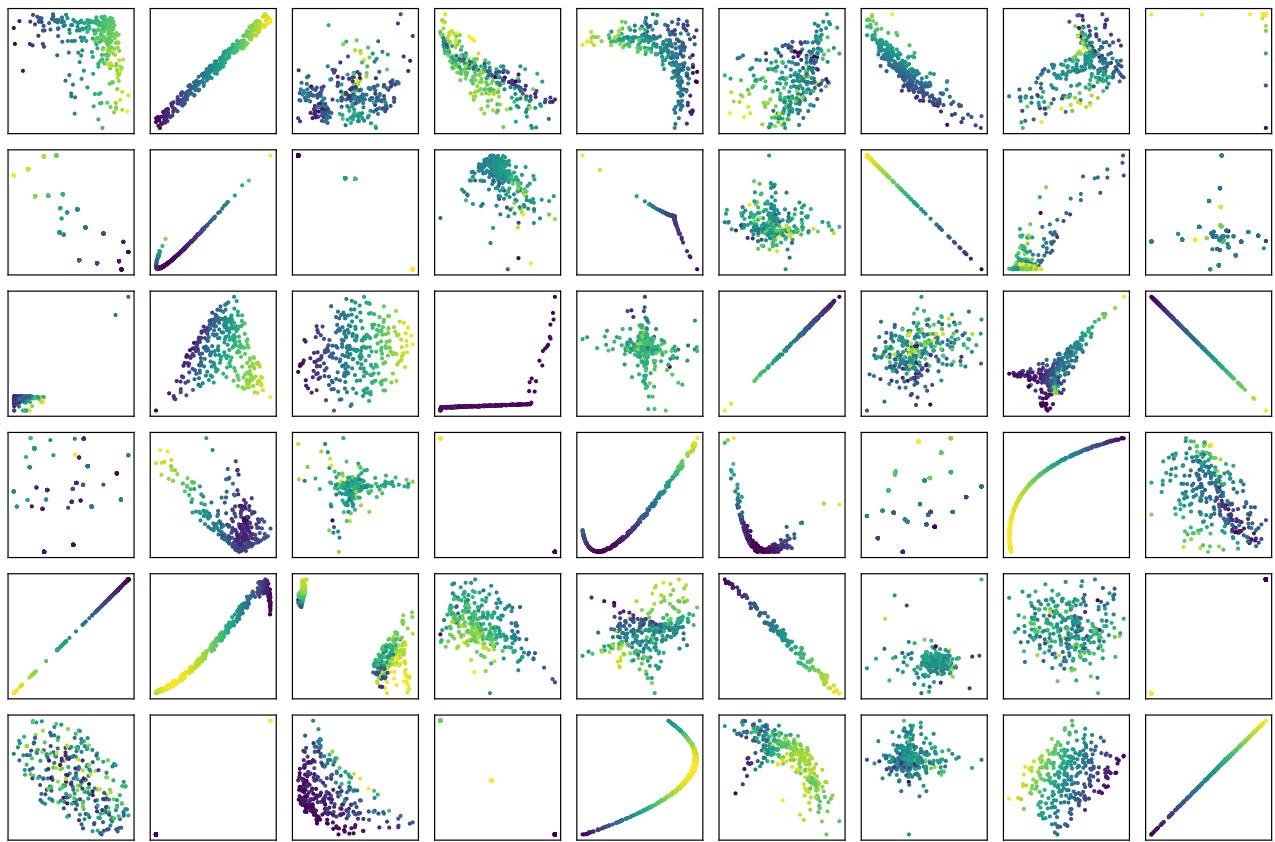

*Figure F.11.* **Samples of RandomPoints from the prior.** We sample 300 three-dimensional points and encode the third dimension through the color.

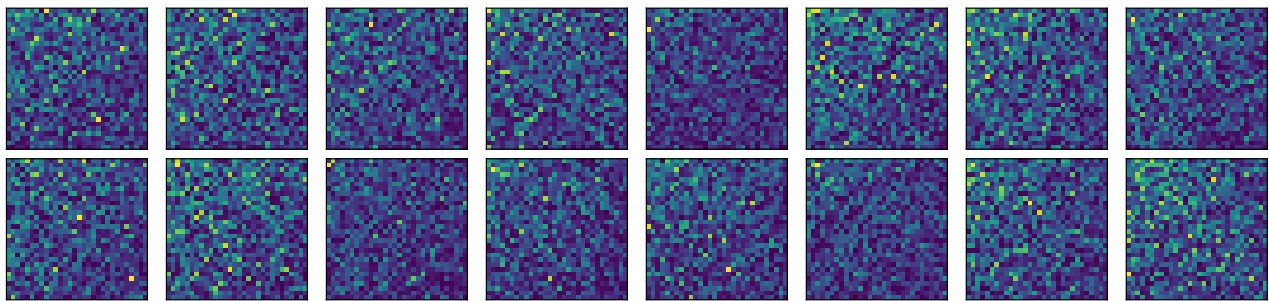

*Figure F.12.* **Samples of RandomGaussianMatrix.** We sample $30 \times 30$ matrices and show the absolute values of their entries. We permute their indices by sorting the absolute values of the top left- and right-singular vectors of their absolute values, respectively.

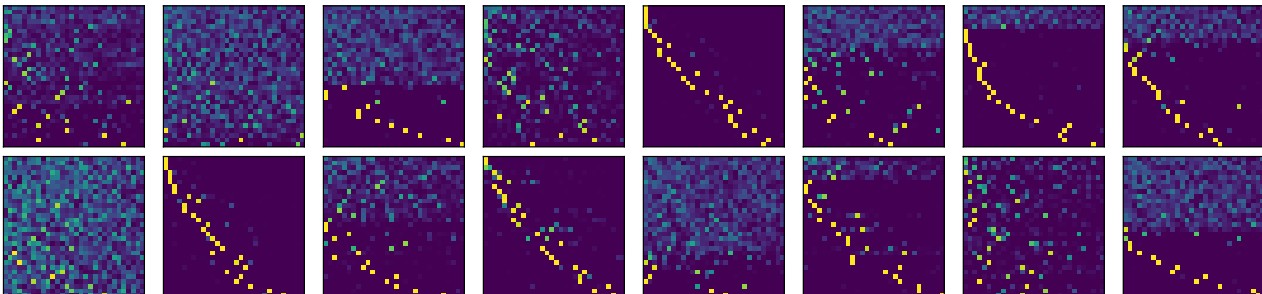

*Figure F.13.* **Samples of RandomWeightsMatrix.** We sample $30 \times 30$ matrices and show the absolute values of their entries. We permute their indices by sorting the absolute values of the top left- and right-singular vectors of their absolute values, respectively.

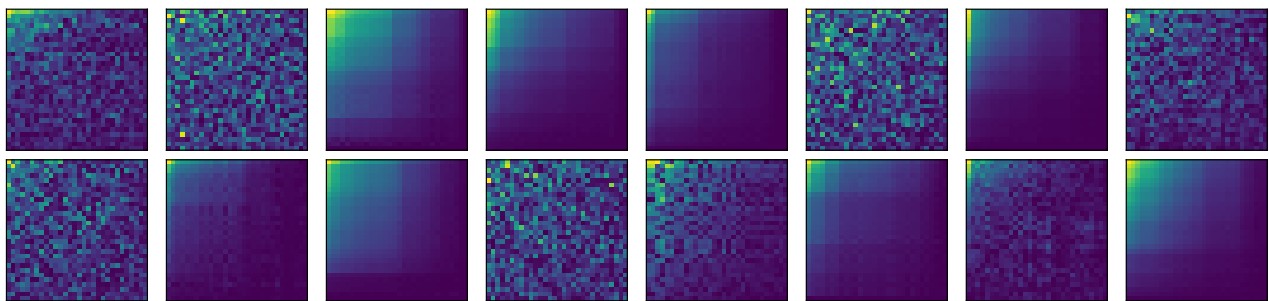

*Figure F.14.* **Samples of RandomSingularValuesMatrix.** We sample $30 \times 30$ matrices and show the absolute values of their entries. We permute their indices by sorting the absolute values of the top left- and right-singular vectors of their absolute values, respectively.

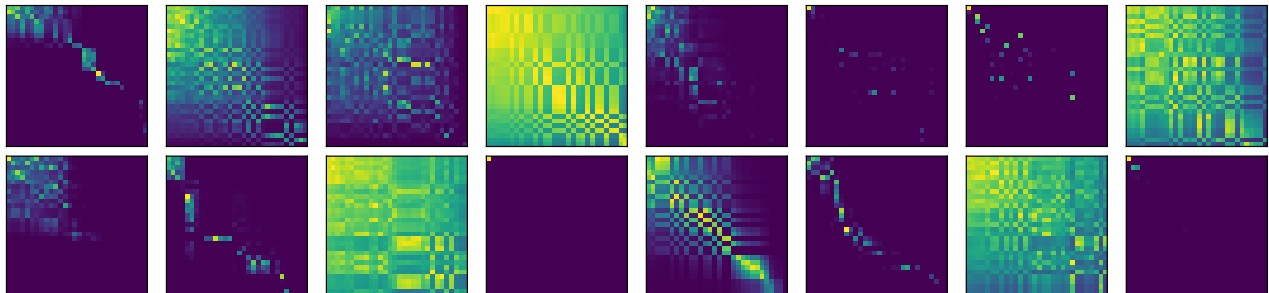

*Figure F.15.* **Samples of RandomKernelMatrix.** We sample $30 \times 30$ matrices and show the absolute values of their entries. We permute their indices by sorting the absolute values of the top left- and right-singular vectors of their absolute values, respectively.

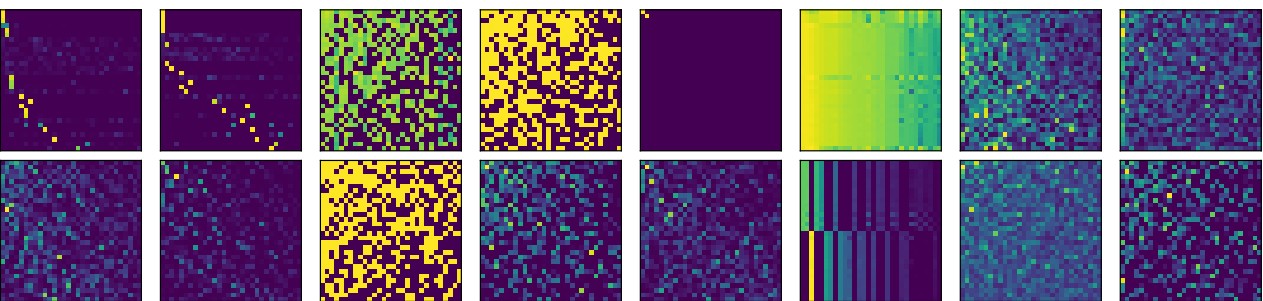

*Figure F.16.* **Samples of RandomActivationMatrix.** We sample $30 \times 30$ matrices and show the absolute values of their entries. We permute their indices by sorting the absolute values of the top left- and right-singular vectors of their absolute values, respectively.

# G. Path Smoothness for Gaussian Processes

In the following, we prove a result for the smoothness of functions sampled from Gaussian processes. We quantify the smoothness in terms of which Sobolev spaces $H^s(B)$ the functions $f$ belong to, for some domain $B \subseteq \mathbb{R}^d$ and smoothness $s \geq 0$. There are different ways to define Sobolev spaces, and we refer the curious reader to the literature, but essentially functions in $H^s(B)$ have an $s$-th derivative that is square-integrable. For non-integer $s$, the Sobolev-Slobodeckij norm treats the fractional part $s - \lfloor s \rfloor$ using a Hölder-like criterion on the $\lfloor s \rfloor$-th derivative.

**Notation.** We write $\mathrm{GP}(0, k)$ for the distribution of Gaussian processes with mean zero and covariance kernel $k$. We assume that such Gaussian processes are defined on a complete probability space. We say that two stochastic processes $(X_t)_{t \in T}, (Y_t)_{t \in T}$ on the same probability space $(\Omega, \mathcal{F}, P)$ are a modification (or version) of each other if $P(X_t = Y_t) = 1$ for all $t$. Moreover, for integrable $g : \mathbb{R}^d \to \mathbb{R}_{\geq 0}$, we denote its inverse Fourier transform as $\check{g}(x) = \int e^{i \langle \omega, x \rangle} g(\omega) \, d\omega$.

In the following theorem, it might be possible to relax the criterion $q > 2d$ to $q > d$ by using other results from the literature. However, in this case the result would not guarantee anymore that the paths are continuous.

**Theorem G.1** (Smoothness of GP sample paths). *Let $g : \mathbb{R}^d \to \mathbb{R}_{\geq 0}$ be integrable and even such that there exist constants $c, C, r_0 > 0$ and $q > 2d$ with*

$$c \|\omega\|^{-q} \leq g(\omega) \leq C \|\omega\|^{-q} \quad \text{for all } \omega \in \mathbb{R}^d \text{ with } \|\omega\| \geq r_0. \tag{G.1}$$

*Then, $k : \mathbb{R}^d \times \mathbb{R}^d \to \mathbb{R}, k(x, x') := \check{g}(x - x')$ is a kernel.*

*Let $X \sim \mathrm{GP}(0, k)$, let $B \subseteq \mathbb{R}^d$ be an arbitrary ball, and set $s_* := (q - d)/2$. Then,*

- *For every $s < s_*$, there exists a modification $Y$ of $X$ such that $P(Y|_B \in H^s(B)) = 1$.*

- *For every $s \geq s_*$ and every modification $Y$ of $X$, $P(Y|_B \in H^s(B)) = 0$.*

*Proof.* **Step 1: "Repair" the spectral density in the center.** It follows from Lemma G.2 that $k$ is a kernel. To apply the remainder of Lemma G.2, we need a spectral density $g_*$ satisfying

$$c_* (1 + \|\omega\|^2)^{-q/2} \leq g_*(\omega) \leq C_* (1 + \|\omega\|^2)^{-q/2} \tag{G.2}$$

for suitable constants $c_*, C^* > 0$ and all $\omega \in \mathbb{R}^d$, not just those with large enough radius. Our function $g$ may not satisfy this since Eq. (G.1) only needs to hold for large enough $\|\omega\|$. Using the ball $B_{r_0} := \{\omega \in \mathbb{R}^d \mid \|\omega\| \leq r_0\}$, define

$$g_{\mathrm{hi}} := g \cdot \mathbb{1}_{\mathbb{R}^d \setminus B_{r_0}}$$
$$g_{\mathrm{lo}} := g \cdot \mathbb{1}_{B_{r_0}}$$
$$g_0(\omega) := (1 + \|\omega\|^2)^{-q/2} \mathbb{1}_{B_{r_0}}(\omega)$$

Then, $g = g_{\mathrm{hi}} + g_{\mathrm{lo}}$, and we can set $g_* = g_{\mathrm{hi}} + g_0$. Since all of these functions are non-negative, integrable, and even, we can also consider their associated kernels. If $X_{\mathrm{hi}} \sim \mathrm{GP}(0, k_{\mathrm{hi}}), X_{\mathrm{lo}} \sim \mathrm{GP}(0, k_{\mathrm{lo}}), X_0 \sim \mathrm{GP}(0, k_0)$ are independent, then $X := X_{\mathrm{hi}} + X_{\mathrm{lo}} \sim \mathrm{GP}(0, k)$ and $X_* := X_{\mathrm{hi}} + X_0 \sim \mathrm{GP}(0, k_*)$.

**Step 2: Equivalence to repaired version.** The inverse Fourier transforms $\check{g}_{\mathrm{lo}}, \check{g}_0$ are infinitely differentiable: for example, since $g_{\mathrm{lo}}$ is supported on a ball of radius $r_0$, we have

$$\check{g}'_{\mathrm{lo}}(x) = \frac{\partial}{\partial x} \int e^{i \langle x, \omega \rangle} g_{\mathrm{lo}}(\omega) \, d\omega = \int \omega e^{i \langle x, \omega \rangle} g_{\mathrm{lo}}(\omega) \, d\omega \,,$$

where the integration and differentiation can be exchanged because

$$\int \|\omega e^{i \langle x, \omega \rangle} g_{\mathrm{lo}}(\omega)\| \, d\omega \leq r_0 \int_{\|\omega\| \leq r_0} |g_{\mathrm{lo}}(\omega)| \, d\omega < \infty \,.$$

Hence, the associated kernels $k_{\mathrm{lo}}, k_0$ are infinitely differentiable. For a given smoothness $s \in \mathbb{R}$, using Theorem 4.1 of Da Costa et al. (2023) and choosing a suitable modification of $X_{\mathrm{lo}}, X_0$, we almost surely have $X_{\mathrm{lo}}|B, X_0|B \in H^s(B)$. Hence, if there exists a modification $\tilde{X}$ of $X$ with $P(\tilde{X}|_B \in H^s(B)) = p$ for some $p \in [0, 1]$, then $\tilde{X}_* := \tilde{X} - X_{\mathrm{lo}} + X_0$

is a modification of $X_*$ with $P(\tilde{X}_*|_B \in H^s(B)) = p$. The same holds with switched roles of $X$ and $X_*$. Hence, it suffices to prove the desired smoothness results for $X_*$ instead of $X$.

**Step 3: Smoothness of sample paths for $X_*$.** By Lemma G.2 and Eq. (G.2), the RKHS $\mathcal{H}_*$ of $k_*$ is equivalent to the Sobolev space $H^{q/2}(\mathbb{R}^d)$, and therefore $\mathcal{H}_*|_B$ is equivalent to $H^{q/2}(\mathbb{R}^d)|_B$, which is equivalent to $H^{q/2}(B)$. Hence, the inclusion operator $I : \mathcal{H}_*|_B \to H^{q/2}(B)$ is bounded and so is its inverse. For Hilbert spaces $A \subseteq B$, Steinwart (2024) defines in Definition 1.1 that $A \ll B$ if the associated inclusion operator from $A \to B$ is Hilbert-Schmidt. From Lemma 6.9 of Steinwart (2024), if $s > d/2$, then $H^{q/2}(B) \ll H^s(B)$ if and only if $s < (q - d)/2 = s_*$. By Theorem 15 of Bell (2016), since $I, I^{-1}$ are bounded, we obtain $\mathcal{H}_*|_B \ll H^s(B)$ if and only if $s < (q - d)/2$. By Theorem 1.2 in Steinwart (2024) (which is a reformulation of a result of Lukić & Beder 2001), the smoothness criterion for $X_*$ is equivalent to $\mathcal{H}_*|_B \ll H^s(B)$, which completes the proof for the cases $s > d/2$. But since the spaces for $s \le d/2$ are supersets of those for $s > d/2$, the statement extends to them as well. $\square$

**Lemma G.2** (Fourier characterization of Sobolev kernels)**.** *Let $g : \mathbb{R}^d \to \mathbb{R}_{\ge 0}$ be even and integrable. Then, $k : \mathbb{R}^d \times \mathbb{R}^d \to \mathbb{R}$, $k(x, y) = \check{g}(x - y) \in \mathbb{R}$ is a kernel. Moreover, if there exist constants $C > 0$ and $s > d/2$ with*

$$C^{-1}(1 + \|\omega\|^2)^{-s} \le g(\omega) \le C(1 + \|\omega\|^2)^{-s} \tag{G.3}$$

*for all $\omega \in \mathbb{R}^d$, then the reproducing kernel Hilbert space (RKHS) $\mathcal{H}_k$ associated with $k$ is equivalent to the Sobolev space $H^s(\mathbb{R}^d)$, meaning that they are equal as sets and their norms are equivalent.*

*Proof.* **Step 1: Constructing the RKHS via a feature space.** Define $g_0 : \mathbb{R}^d \to \mathbb{R}_{\ge 0}, \omega \mapsto (1 + \|\omega\|^2)^{-s}$ and $g_1 := g$. Since $s > d/2$, $g_0$ is integrable, and $g_1$ is integrable by assumption. We will define feature maps into $H := L^2(\mathbb{R}^d, \mathbb{C})$, the Hilbert space of complex-valued square-integrable functions on $\mathbb{R}^d$. For $i \in \{0, 1\}$, define

$$\phi_i : \mathbb{R}^d \to L^2(\mathbb{R}^d, \mathbb{C}), \quad \phi_i(x)(\omega) := e^{-i\langle x, \omega \rangle} \sqrt{g_i(\omega)} ,$$

which is valid since $g \in L^1$ implies $\sqrt{g} \in L^2$. This feature map is associated with the kernel

$$k_i(x, y) = \langle \phi_i(x), \phi_i(y) \rangle_H = \int \overline{\phi_i(x)(\omega)} \phi_i(y)(\omega) \, d\omega = \int e^{i\langle x-y, \omega \rangle} g_i(\omega) \, d\omega = \check{g}_i(x - y) , \tag{G.4}$$

which is real-valued since $g_i$ is even. Especially, $k = k_1$ is a real-valued kernel. Now, by Theorem 4.21 in Steinwart & Christmann (2008), the associated RKHS is

$$\mathcal{H}_i := \mathcal{H}_{g_i} = \{x \mapsto \langle h, \phi_i(x) \rangle_H \mid h \in H\}$$

with norm

$$\|f\|_{\mathcal{H}_i} = \inf\{\|h\|_H \mid h \in H, f = \langle h, \phi_i(\cdot) \rangle_H\} .$$

**Step 2: Equivalence of the RKHSs.** Now, by Eq. (G.3), we have $C^{-1}g_0 \le g_1 \le Cg_0$. If $f \in \mathcal{H}_0$, then there exists $h \in H$ with $f = \langle h, \phi_0(\cdot) \rangle_H$, and therefore $f = \langle h\sqrt{g_0/g_1}, \phi_1(\cdot) \rangle_H \in \mathcal{H}_1$, since $\|h\sqrt{g_0/g_1}\|_H \le \sqrt{C}\|h\|_H$. Together with the norm characterization above it therefore follows that $\mathcal{H}_0 \subseteq \mathcal{H}_1$ with $\|f\|_{\mathcal{H}_1} \le \sqrt{C}\|f\|_{\mathcal{H}_0}$. By switching the roles of $g_0$ and $g_1$, we therefore conclude that $\mathcal{H}_0$ and $\mathcal{H}_1$ are equivalent.

**Step 3: $\mathcal{H}_0$ is the desired Sobolev space.** Eq. (G.4) yields $k_0(x, y) = \int e^{i\langle x-y, \omega \rangle}(1 + \|\omega\|^2)^{-s} \, d\omega$, which is up to a constant factor exactly the kernel of (an equivalent version of) the Sobolev space $H^s(\mathbb{R}^d)$ (see e.g. De Vito et al., 2021, section 7.1). By step 2, this means that the RKHS of $g$ is equivalent to $H^s(\mathbb{R}^d)$. $\square$

# H. Inference Optimization for TabICLv2

## H.1. Efficient attention computation via selective query-key-value projections

We implement two attention optimizations that reduce redundant computation by selectively computing query, key, and value projections based on the specific attention patterns required in each stage of TabICLv2.

**Row-wise inter-feature interaction.** During the row-wise interaction, we prepend $c$ learnable [CLS] tokens (we use $c = 4$ as TabICL) to the feature embeddings and use only their final outputs, which are concatenated into a single vector as the row representation for the subsequent in-context learning. Since only the [CLS] token outputs are required, we optimize the final block of $\text{TF}_{\text{row}}$ by restricting the query computation to these $c$ positions while allowing them to attend to the full sequence ([CLS] tokens and all features) as keys and values. Concretely, given a sequence of length $c + m$, the final block computes attention outputs only for the first $c$ query positions:

$$\text{Attention}(\mathbf{Q}_{1:c}, \mathbf{K}_{1:c+m}, \mathbf{V}_{1:c+m})$$

where $\mathbf{Q}_{1:c} \in \mathbb{R}^{c \times d}$ represents queries from [CLS] tokens only, while $\mathbf{K}, \mathbf{V} \in \mathbb{R}^{(c+m) \times d}$ span the full sequence. This reduces the query projection cost from $\mathcal{O}((c + m) \cdot d^2)$ to $\mathcal{O}(c \cdot d^2)$ and the attention computation from $\mathcal{O}((c + m)^2 \cdot d)$ to $\mathcal{O}(c \cdot (c + m) \cdot d)$ in the final block.

**Dataset-wise in-context learning.** During the final in-context learning, test samples learn from training samples via cross-attention, where test queries attend only to training keys and values. Since test samples never serve as context for other samples, computing their key and value projections is unnecessary. We therefore compute key and value projections only for the $n_{\text{train}}$ training samples, while computing queries for all $n_{\text{train}} + n_{\text{test}}$ samples:

$$\text{Attention}(\mathbf{Q}_{1:n_{\text{train}}+n_{\text{test}}}, \mathbf{K}_{1:n_{\text{train}}}, \mathbf{V}_{1:n_{\text{train}}})$$

This reduces the key and value projection costs from $\mathcal{O}((n_{\text{train}} + n_{\text{test}}) \cdot d^2)$ to $\mathcal{O}(n_{\text{train}} \cdot d^2)$ each.

**Layer normalization reuse.** We adopted the pre-norm setting in the transformer block. To avoid redundant computation, we first apply layer normalization to the full input sequence, then perform slicing to extract the required subset for query, key, or value computation. This ensures that normalization statistics are computed once over the complete sequence, and the sliced representations remain properly normalized without requiring separate normalization passes for different subsets.

## H.2. Offloading technique

**Batch size estimation.** To dynamically adjust the batch size of the three transformers based on available memory and avoid out-of-memory errors, following TabICL, TabICLv2 employs polynomial regression to estimate the inference peak GPU memory consumption:

$$\text{MEM} = \alpha_1 \times \text{batch\_size} + \alpha_2 \times \text{seq\_len} + \alpha_3 \times \text{batch\_size} \times \text{seq\_len} + \alpha_4$$

Compared to TabICL, we introduce query-aware scalable softmax in $\text{TF}_{\text{col}}$ and $\text{TF}_{\text{icl}}$, which requires re-evaluating the memory coefficients for these components. The updated coefficients are as follows ($\text{TF}_{\text{row}}$ remains unchanged from TabICL):

$$\text{MEM}_{\text{col}} = 0.146 \times \text{batch\_size} + 1.94 \times 10^{-5} \times \text{seq\_len} + 0.00488 \times \text{batch\_size} \times \text{seq\_len} + 142.91$$

$$\text{MEM}_{\text{row}} = -2.07 \times 10^{-5} \times \text{batch\_size} + 2.27 \times 10^{-4} \times \text{seq\_len} + 0.00537 \times \text{batch\_size} \times \text{seq\_len} + 138.54$$

$$\text{MEM}_{\text{icl}} = -0.04 \times \text{batch\_size} + 5.43 \times 10^{-7} \times \text{seq\_len} + 0.0195 \times \text{batch\_size} \times \text{seq\_len} + 142.84$$

where the estimated memory is measured in megabytes (MB).

TabICLv2 demonstrates substantially improved capability for handling large tables compared to TabICL. To provide an affordable setup for processing tables with millions of samples, we implement a hierarchical offloading strategy that extends beyond the CPU offloading of TabICL to include disk-based offloading.

**Memory bottleneck analysis.** Given an input $X \in \mathbb{R}^{b \times n \times m}$, where $b$, $n$, and $m$ represent the number of datasets, the number of samples, and the number of features respectively, $X$ is first reshaped to $\mathbb{R}^{(b \times m) \times n}$ and processed by $\text{TF}_{\text{col}}$ to produce feature embeddings $E \in \mathbb{R}^{(b \times m) \times n \times d}$. The memory bottleneck lies in storing this intermediate tensor $E$. For a table with one million samples and 500 features, $E$ requires approximately 250 GB of memory (with $d = 128$ and float32 precision). TabICL addresses this by offloading $E$ to CPU memory during column-wise embedding. However, 250 GB of CPU memory remains prohibitive for many users. We therefore implement disk offloading to further alleviate memory constraints.

**Disk offloading via memory-mapped files.** Our disk offloading implementation leverages memory-mapped files (memmap) through NumPy, which allows the operating system to handle paging between disk and memory transparently. The key components are:

1. **Pre-allocation:** Before processing begins, we pre-allocate a memory-mapped file on disk with the exact size required for the output tensor. This reserves contiguous disk space and avoids fragmentation during incremental writes.

2. **Incremental writing:** During column-wise embedding, each batch's output is written directly to the memory-mapped file at the corresponding indices. This streaming approach ensures that GPU memory only holds the current batch, while completed results are persisted to disk.

3. **Periodic flushing:** To balance I/O efficiency with memory usage, we periodically flush the memory-mapped file to disk after accumulating a configurable amount of data (default is 8 GB). This prevents the operating system's page cache from consuming excessive memory.

4. **Automatic cleanup:** We register a weak reference finalizer for each memory-mapped file, ensuring automatic deletion when the associated tensor is garbage collected.

**Asynchronous data transfer.** To overlap GPU computation with data movement, we employ a dedicated CUDA stream for device-to-host (D2H) transfers. The workflow operates as follows: (1) GPU tensor is asynchronously copied to a pinned CPU buffer on the copy stream; (2) a CUDA event is recorded to track completion; (3) upon event completion, data is written to the final target (CPU tensor or disk); (4) the pinned buffer is returned to a buffer pool for reuse. This pipelining hides transfer latency and improves throughput. We maintain a configurable maximum number of pending asynchronous copies (default is 4) before blocking, balancing memory usage against throughput.

**Automatic mode selection.** We provide an inference manager that supports four offloading modes: `GPU` (keep everything on GPU), `CPU` (offload to CPU memory), `DISK` (offload to memory-mapped files), and `AUTO` (automatically choose based on available resources). In `AUTO` mode, the manager estimates the output tensor size and compares it against available GPU memory, CPU memory, and disk space with configurable safety factors to select the most appropriate storage backend.

Figures H.1 and H.2 illustrate the resource utilization profiles for CPU and disk offloading respectively, processing a table with 1 million samples and 500 features (80% training, 20% test) on an H100 GPU with FlashAttention-3 and automatic mixed precision enabled. CPU offloading achieves faster execution (115s vs. 450s) but requires 250 GB of RAM. Disk offloading trades speed for accessibility, requiring only 24 GB of CPU memory and 50 GB of GPU memory while using 250 GB of disk space, which is a configuration available on most modern workstations.

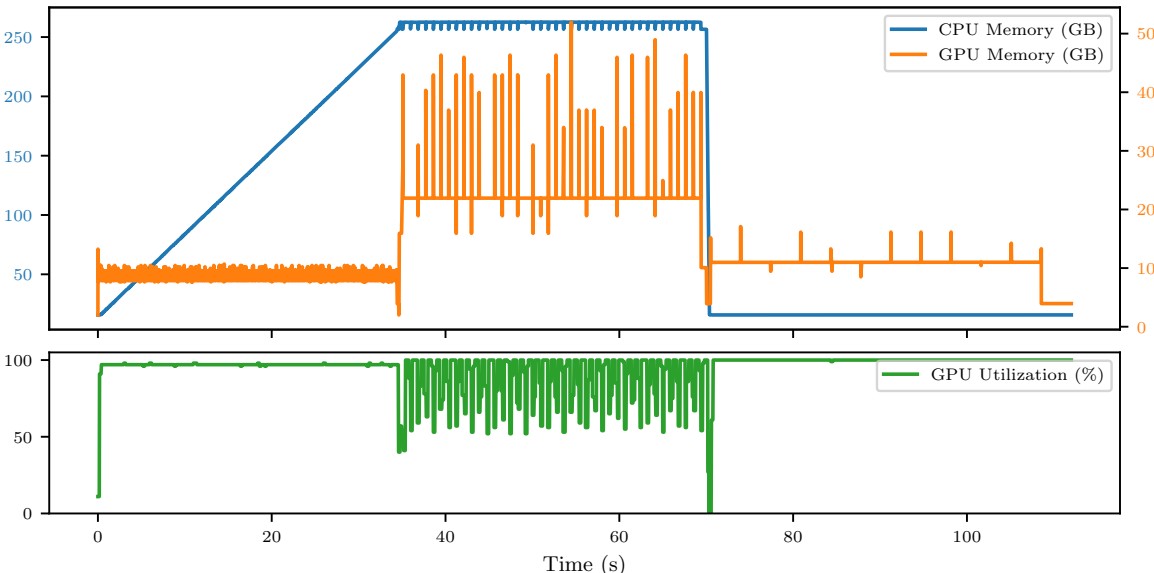

*Figure H.1.* **CPU offloading for a table with 1M samples and 500 features.** The intermediate feature embeddings tensor $E$ requires approximately 250 GB of storage. During column-wise embedding (0–35s), $E$ is progressively offloaded to CPU memory, causing CPU memory usage to increase linearly. During row-wise interaction (35–70s), batches are loaded from CPU to GPU for computation, resulting in fluctuating GPU utilization due to CPU-GPU communication overhead. The forward pass completes in 115 seconds with peak GPU memory of 50 GB. However, the 250 GB CPU memory requirement remains prohibitive for most systems.

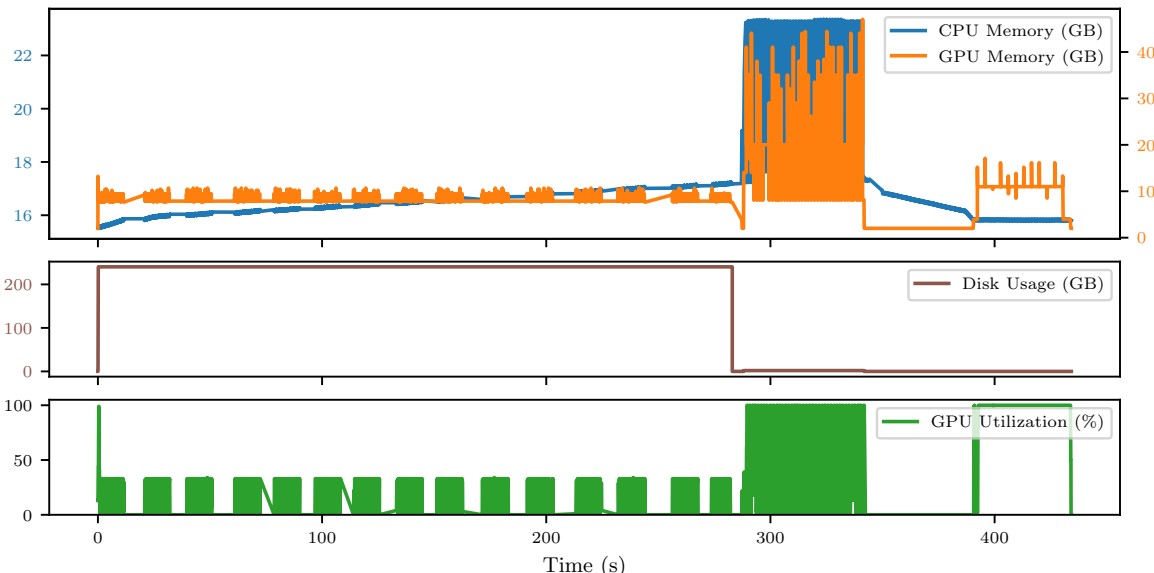

*Figure H.2.* **Disk offloading for a table with 1M samples and 500 features.** We first pre-allocate a 250 GB memory-mapped file on disk. During column-wise embedding (0–280s), feature embeddings are streamed directly to disk. The periodic drops in GPU utilization correspond to synchronization points where the asynchronous copy manager drains pending transfers and flushes data to disk. During row-wise interaction (280–350s), data is loaded from disk in batches. The total forward pass takes 450 seconds—approximately 4× slower than CPU offloading due to disk I/O latency, but with dramatically reduced memory requirements (CPU memory under 24 GB and GPU memory under 50 GB) that make million-scale inference accessible on commodity hardware.

# I. Quantile Distribution

## I.1. Problem setup

For regression tasks, given an input feature vector $x$, TabICLv2 predicts a set of conditional quantiles:

$$\hat{Q}(\alpha|x) \quad \text{for } \alpha \in \{\alpha_1, \alpha_2, \ldots, \alpha_K\}$$

where $\alpha_k$ represents probability levels. In this work, we set $K = 999$ and choose uniformly spaced probability levels between $\alpha_L = 0.001$ and $\alpha_R = 0.999$, that is, $\boldsymbol{\alpha} = \{0.001, 0.002, \ldots, 0.999\}$. However, these raw predicted quantiles present several challenges (Chernozhukov et al., 2010; Bondell et al., 2010):

1. **Quantile crossing**: TabICLv2 may predict non-monotonic quantiles, i.e., $\hat{Q}(\alpha_i) > \hat{Q}(\alpha_j)$ for $\alpha_i < \alpha_j$, which violates the fundamental property of quantile functions.

2. **Incomplete support**: Predictions are only available for $\alpha \in [\alpha_L, \alpha_R]$, leaving the extreme tails undefined.

3. **Lack of analytical functions**: Raw quantiles do not directly provide probability density function (PDF), cumulative distribution function (CDF), or analytical moments required by many downstream applications.

Therefore, we propose an approach to construct a probabilistic distribution from predicted quantiles, addressing these challenges by:

1. **Monotonicity enforcement**: Correcting quantile crossing and enforcing monotonic quantiles for a valid quantile function.

2. **Tail extrapolation**: Extending the distribution beyond $[\alpha_L, \alpha_R]$ using parametric exponential tail models with data-inferred parameters.

3. **Analytical statistics**: Providing closed-form expressions for PDF, CDF, and analytical moments.

## I.2. Quantile function

The complete quantile function is defined over three regions:

$$Q(\alpha) = \begin{cases} Q_{\text{left}}(\alpha) & \text{if } \alpha < \alpha_L \text{ (left tail)} \\ Q_{\text{spline}}(\alpha) & \text{if } \alpha_L \leq \alpha \leq \alpha_R \text{ (interior)} \\ Q_{\text{right}}(\alpha) & \text{if } \alpha > \alpha_R \text{ (right tail)} \end{cases}$$

In the interior region $[\alpha_L, \alpha_R]$, the quantile function is modeled as a piecewise linear spline connecting the predicted quantile knots. For $\alpha \in [\alpha_i, \alpha_{i+1}]$ where $i \in \{1, 2, \ldots, K-1\}$:

$$Q_{\text{spline}}(\alpha) = q_i + \frac{q_{i+1} - q_i}{\alpha_{i+1} - \alpha_i}(\alpha - \alpha_i) = q_i + m_i(\alpha - \alpha_i)$$

where the slope of segment $i$ is:

$$m_i = \frac{q_{i+1} - q_i}{\alpha_{i+1} - \alpha_i} = \frac{\Delta q_i}{\Delta \alpha_i}$$

The slopes $m_i$ must be non-negative for a valid quantile function, which is ensured by the monotonicity correction described later. To extrapolate the distribution beyond the observed quantile range $[\alpha_L, \alpha_R]$, we employ parametric exponential tail models suitable for sub-exponential distributions (e.g., Gaussian) (Beirlant et al., 2004).

For the left tail ($\alpha < \alpha_L$):

$$Q_{\text{left}}(\alpha) = \beta_L \ln(\alpha) + c_L$$

where $\beta_L > 0$ is the scale parameter and $c_L$ is the intercept determined by continuity at the boundary.

Requiring $Q_{\text{left}}(\alpha_L) = q_L$ gives:

$$c_L = q_L - \beta_L \ln(\alpha_L)$$

Thus, the complete left tail formula is:

$$Q_{\text{left}}(\alpha) = q_L + \beta_L \ln\left(\frac{\alpha}{\alpha_L}\right)$$

For the right tail ($\alpha > \alpha_R$):

$$Q_{\text{right}}(\alpha) = -\beta_R \ln(1 - \alpha) + c_R$$

where $\beta_R > 0$ is the scale parameter.

Requiring $Q_{\text{right}}(\alpha_R) = q_R$ gives:

$$c_R = q_R + \beta_R \ln(1 - \alpha_R)$$

Thus, the complete right tail formula is:

$$Q_{\text{right}}(\alpha) = q_R - \beta_R \ln\left(\frac{1 - \alpha}{1 - \alpha_R}\right)$$

The derivative $dQ/d\alpha$ is crucial for PDF computation:

$$\frac{dQ}{d\alpha}(\alpha) = \begin{cases} \frac{dQ_{\text{left}}}{d\alpha} = \frac{\beta_L}{\alpha}, & 0 < \alpha < \alpha_L \\ \frac{dQ_{\text{spline}}}{d\alpha} = \frac{q_{i+1} - q_i}{\alpha_{i+1} - \alpha_i}, & \alpha \in [\alpha_i, \alpha_{i+1}) \\ \frac{dQ_{\text{right}}}{d\alpha} = \frac{\beta_R}{1 - \alpha}, & \alpha_R < \alpha < 1 \end{cases}$$

### I.3. Quantile crossing correction

A quantile crossing violation occurs when for some $i < j$:

$$\hat{Q}(\alpha_i) > \hat{Q}(\alpha_j) \quad \text{despite } \alpha_i < \alpha_j$$

This violates the fundamental monotonicity requirement of quantile functions. We provide three ways to handle this issue:

**(1) No correction.** The simplest approach is to ignore crossing violations. This may be acceptable when crossings are rare. However, this can lead to invalid probability densities in regions where crossings occur.

**(2) Sorting.** A straightforward correction is to sort the predicted quantiles:

$$\mathbf{q}^{\text{sorted}} = \text{sort}(\hat{q}_1, \hat{q}_2, \ldots, \hat{q}_K)$$

This guarantees monotonicity with $O(K \log K)$ complexity. However, sorting may destroy the correspondence between quantile values and their original probability levels, which can distort the distribution shape.

**(3) Isotonic regression.** The optimal correction in the $L^2$ sense is given by isotonic regression (Barlow & Brunk, 1972), which solves:

$$\mathbf{q}^* = \underset{\mathbf{q}: q_1 \leq q_2 \leq \cdots \leq q_K}{\arg\min} \sum_{k=1}^{K} w_k (\hat{q}_k - q_k)^2$$

where $w_k > 0$ are optional weights (by default, $w_k = 1$). This problem admits a unique solution that can be computed in $O(K)$ time via the Pool Adjacent Violators Algorithm (PAVA) (Best & Chakravarti, 1990). PAVA iteratively merges adjacent blocks that violate monotonicity and replaces each merged block with its weighted average. Despite its linear per-sample complexity, PAVA is inherently a 1D, data-dependent procedure in which the sequence of merges depends on local violations and therefore introduces sequential dependencies along the quantile index. As a result, it is not straightforward to implement

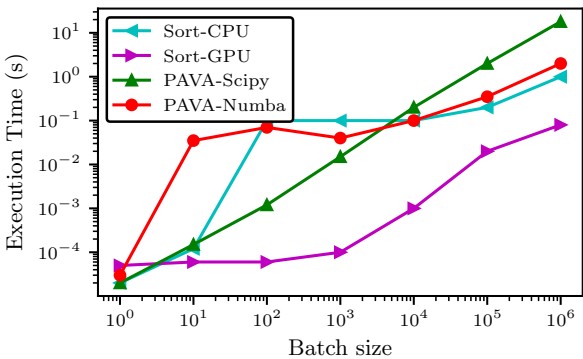

*Figure I.1.* Execution time comparison of different monotonicity enforcement methods across batch sizes ($K = 999$ quantiles per sample). **Sort-GPU** (`torch.sort` on CUDA) achieves the best performance. **Numba** (our parallel PAVA implementation) and **Sort-CPU** show competitive scaling on CPU. **Scipy** (`scipy.optimize.isotonic_regression`) processes samples sequentially, becoming slower than Numba at large batch sizes.

PAVA as a single fully vectorized operation, which limits its practical throughput because we often need to process large batches of predictions (e.g., the entire test set).

We leverage the just-in-time compilation of Numba (Lam et al., 2015) to optimize the PAVA implementation, achieving effective CPU parallelism. As shown in Figure I.1, the Numba-based PAVA implementation achieves competitive performance with `torch.sort` on CPU for large batch sizes, while being faster than the PAVA implementation of Scipy (Virtanen et al., 2020).

However, the Numba-based implementation cannot compete with `torch.sort` on GPU, which relies on highly optimized parallel sorting primitives implemented in dedicated CUDA kernels. Given this performance gap and the empirical observation that quantile crossing is rare in the predictions of TabICLv2, we use sorting by default for monotonicity enforcement.

### I.4. Tail parameter estimation

The exponential tail scale parameters $\beta_L$ and $\beta_R$ are estimated from the boundary quantiles using log-space linear regression. For the left tail, the model is $Q(\alpha) = \beta_L \ln(\alpha) + c_L$. Given $K_{\text{tail}}$ quantiles in the left tail region (default is 20), we estimate $\beta_L$ using ordinary least squares:

$$\hat{\beta}_L = \frac{\text{Cov}(Q, \ln \alpha)}{\text{Var}(\ln \alpha)}\bigg|_{\alpha \in \{\alpha_1, \ldots, \alpha_{K_{\text{tail}}}\}}$$

Specifically:

$$\hat{\beta}_L = \frac{\sum_{k=1}^{K_{\text{tail}}} (q_k - \bar{q})(\ln \alpha_k - \overline{\ln \alpha})}{\sum_{k=1}^{K_{\text{tail}}} (\ln \alpha_k - \overline{\ln \alpha})^2}$$

For the right tail, the model is $Q(\alpha) = -\beta_R \ln(1 - \alpha) + c_R$. The estimation is analogous:

$$\hat{\beta}_R = -\frac{\text{Cov}(Q, \ln(1 - \alpha))}{\text{Var}(\ln(1 - \alpha))}\bigg|_{\alpha \in \{\alpha_{K - K_{\text{tail}}+1}, \ldots, \alpha_K\}}$$

The estimated parameters are clamped to ensure numerical stability: $\beta \in [\beta_{\min}, \beta_{\max}] = [0.01, 100]$.

### I.5. Cumulative distribution function (CDF)

The CDF $F(z) = P(Z \leq z)$ is the inverse of the quantile function: $F(z) = Q^{-1}(z)$.

**Spline region** ($z \in [q_i, q_{i+1})$):

$$F_{\text{spline}}(z) = \alpha_i + \frac{z - q_i}{q_{i+1} - q_i}(\alpha_{i+1} - \alpha_i)$$

**Left tail** ($z < q_L$): From $z = q_L + \beta_L \ln(\alpha/\alpha_L)$, we solve for $\alpha$:

$$F_{\text{left}}(z) = \alpha_L \exp\left(\frac{z - q_L}{\beta_L}\right)$$

**Right tail** ($z > q_R$):

$$F_{\text{right}}(z) = 1 - (1 - \alpha_R) \exp\left(-\frac{z - q_R}{\beta_R}\right)$$

### I.6. Probability density function (PDF)

The PDF is related to the quantile function derivative by the inverse function theorem:

$$f(z) = \frac{1}{Q'(F(z))} = \frac{1}{\frac{dQ}{d\alpha}\big|_{\alpha=F(z)}}$$

The PDF computation procedure is:

1. Compute $\alpha = F(z)$ using the appropriate CDF formula

2. Compute $\frac{dQ}{d\alpha}$ at $\alpha$ using the appropriate derivative formula

3. Return $f(z) = \left(\frac{dQ}{d\alpha}\right)^{-1}$

**Spline region** ($z \in [q_i, q_{i+1})$):

$$f_{\text{spline}}(z) = \frac{1}{m_i} = \frac{\alpha_{i+1} - \alpha_i}{q_{i+1} - q_i}$$

**Left tail** ($z < q_L$):

$$f_{\text{left}}(z) = \frac{F(z)}{\beta_L} = \frac{\alpha_L}{\beta_L} \exp\left(\frac{z - q_L}{\beta_L}\right)$$

**Right tail** ($z > q_R$):

$$f_{\text{right}}(z) = \frac{1 - F(z)}{\beta_R} = \frac{1 - \alpha_R}{\beta_R} \exp\left(-\frac{z - q_R}{\beta_R}\right)$$

The log probability density is computed directly to avoid numerical issues with very small densities:

$$\ln f(z) = -\ln\left(\frac{dQ}{d\alpha}\bigg|_{\alpha=F(z)}\right)$$

### I.7. Continuous ranked probability score (CRPS)

The CRPS for a distribution with CDF $F$ and observation $z$ is:

$$\text{CRPS}(F, z) = \int_{-\infty}^{\infty} (F(y) - \mathbf{1}_{y \geq z})^2 \, dy$$

where $\mathbf{1}_{y \geq z}$ is the indicator function. This can be equivalently expressed in quantile space:

$$\text{CRPS}(F, z) = \int_0^1 2\rho_\alpha(z - Q(\alpha))d\alpha$$

where $\rho_\alpha(u) = u(\alpha - \mathbf{1}_{u < 0})$ is the pinball loss. The CRPS decomposes as:

$$\text{CRPS}(F, z) = \int_0^{F(z)} 2\alpha(z - Q(\alpha))d\alpha + \int_{F(z)}^1 2(1 - \alpha)(Q(\alpha) - z)d\alpha$$

We compute CRPS analytically by integrating over each region.

### I.7.1. CRPS CONTRIBUTION FROM SPLINE REGION

For segment $i$ with $\alpha \in [\alpha_i, \alpha_{i+1}]$ and $Q(\alpha) = q_i + m_i(\alpha - \alpha_i)$, we let $r = \min(\max(F(z), \alpha_i), \alpha_{i+1})$ be the clamped CDF value. The contribution to CRPS from segment $i$ is:

$$\mathrm{CRPS}_i = I_1^{(i)} + I_2^{(i)}$$

where:

$$I_1^{(i)} = (z - q_i)(r^2 - \alpha_i^2) - 2m_i \left( \frac{r^3}{3} - \frac{\alpha_i r^2}{2} + \frac{\alpha_i^3}{6} \right)$$

$$I_2^{(i)} = 2 \int_r^{\alpha_{i+1}} (1 - \alpha)(Q(\alpha) - z) \, d\alpha$$

For the first integral ($\alpha \leq F(z)$):

$$I_1^{(i)} = \int_{\alpha_i}^r 2\alpha(z - Q(\alpha)) \, d\alpha$$

$$= \int_{\alpha_i}^r 2\alpha(z - q_i - m_i(\alpha - \alpha_i)) \, d\alpha$$

$$= 2(z - q_i) \int_{\alpha_i}^r \alpha \, d\alpha - 2m_i \int_{\alpha_i}^r \alpha(\alpha - \alpha_i) \, d\alpha$$

Computing each integral:

$$\int_{\alpha_i}^r \alpha \, d\alpha = \frac{r^2 - \alpha_i^2}{2}$$

$$\int_{\alpha_i}^r \alpha(\alpha - \alpha_i) \, d\alpha = \int_{\alpha_i}^r (\alpha^2 - \alpha_i \alpha) \, d\alpha = \frac{r^3 - \alpha_i^3}{3} - \alpha_i \frac{r^2 - \alpha_i^2}{2}$$

Substituting yields the formula for $I_1^{(i)}$. The second integral $I_2^{(i)}$ is computed analogously.

The total spline CRPS is:

$$\mathrm{CRPS}_{\mathrm{spline}} = \sum_{i=1}^{K-1} \mathrm{CRPS}_i$$

### I.7.2. CRPS CONTRIBUTION FROM EXPONENTIAL LEFT TAIL

For the left exponential tail with $Q(\alpha) = q_L + \beta_L \ln(\alpha/\alpha_L)$, let $\tilde{\alpha} = \min(F(z), \alpha_L)$ be the clamped CDF value and $b_L = q_L - \beta_L \ln \alpha_L$. We have:

$$\mathrm{CRPS}_{\mathrm{left}} = (z - b_L)(\alpha_L^2 - 2\alpha_L + 2\tilde{\alpha}) + \alpha_L^2 \beta_L \left( -\ln \alpha_L + \frac{1}{2} \right) + T_{\mathrm{left}}$$

where:

$$T_{\mathrm{left}} = \begin{cases} 2\alpha_L \beta_L (\ln \alpha_L - 1) + 2\tilde{\alpha}(-z + b_L + \beta_L) & \text{if } z < q_L \\ 0 & \text{otherwise} \end{cases}$$

### I.7.3. CRPS CONTRIBUTION FROM EXPONENTIAL RIGHT TAIL

For the right exponential tail with $Q(\alpha) = q_R - \beta_R \ln((1 - \alpha)/(1 - \alpha_R))$, let $\tilde{\alpha} = \max(F(z), \alpha_R)$ be the clamped CDF value. We have:

$$\mathrm{CRPS}_{\mathrm{right}} = (z - b_R)(-1 - \alpha_R^2 + 2\tilde{\alpha}) + a_R \left( -\frac{(1 + \alpha_R)^2}{2} + (\alpha_R^2 - 1) \ln(1 - \alpha_R) + 2\tilde{\alpha} \right) + T_{\mathrm{right}}$$

where $a_R = -\beta_R$, $b_R = q_R + \beta_R \ln(1 - \alpha_R)$, and:

$$T_{\mathrm{right}} = \begin{cases} 2(1 - \tilde{\alpha})(z - b_R) & \text{if } z > q_R \\ 2a_R(1 - \alpha_R) \ln(1 - \alpha_R) & \text{otherwise} \end{cases}$$

## I.8. Moment calculations

### I.8.1. MEAN

The mean of a distribution can be computed as:

$$\mathbb{E}[Z] = \int_0^1 Q(\alpha)d\alpha$$

**Spline contribution**:

$$\int_{\alpha_L}^{\alpha_R} Q_{\text{spline}}(\alpha)d\alpha = \sum_{i=1}^{K-1} \frac{(q_i + q_{i+1})\Delta\alpha_i}{2}$$

**Left tail contribution:**

$$
\begin{aligned}
\int_0^{\alpha_L} Q_{\text{left}}(\alpha)\,\mathrm{d}\alpha &= \int_0^{\alpha_L} (q_L + \beta_L \ln(\alpha/\alpha_L))\,\mathrm{d}\alpha \\
&= q_L\alpha_L + \beta_L \int_0^{\alpha_L} \ln(\alpha/\alpha_L)\,\mathrm{d}\alpha \\
&= q_L\alpha_L + \beta_L \left[\alpha \ln(\alpha/\alpha_L) - \alpha\right]_0^{\alpha_L} \\
&= q_L\alpha_L - \beta_L\alpha_L = \alpha_L(q_L - \beta_L)
\end{aligned}
$$

**Right tail contribution:**

$$
\begin{aligned}
\int_{\alpha_R}^1 Q_{\text{right}}(\alpha)\,\mathrm{d}\alpha &= \int_{\alpha_R}^1 (q_R - \beta_R \ln((1-\alpha)/(1-\alpha_R)))\,\mathrm{d}\alpha \\
&= (1-\alpha_R)(q_R + \beta_R)
\end{aligned}
$$

**Total mean:**

$$\mathbb{E}[Z] = \alpha_L(q_L - \beta_L) + \sum_{i=1}^{K-1} \frac{(q_i + q_{i+1})\Delta\alpha_i}{2} + (1-\alpha_R)(q_R + \beta_R)$$

### I.8.2. VARIANCE

The variance is computed as:

$$\text{Var}[Z] = \mathbb{E}[Z^2] - (\mathbb{E}[Z])^2$$

where:

$$\mathbb{E}[Z^2] = \int_0^1 Q(\alpha)^2 d\alpha$$

**Spline contribution**: For a linear segment:

$$
\begin{aligned}
\int_{\alpha_i}^{\alpha_{i+1}} Q(\alpha)^2\,\mathrm{d}\alpha &= \int_{\alpha_i}^{\alpha_{i+1}} (q_i + m_i(\alpha - \alpha_i))^2\,\mathrm{d}\alpha \\
&= \frac{\Delta\alpha_i}{3}(q_i^2 + q_iq_{i+1} + q_{i+1}^2)
\end{aligned}
$$

Thus:

$$\mathbb{E}[Z^2]_{\text{spline}} = \sum_{i=1}^{K-1} \frac{\Delta\alpha_i(q_i^2 + q_iq_{i+1} + q_{i+1}^2)}{3}$$

**Left tail contribution:**

$$\mathbb{E}[Z^2]_{\text{left}} = \alpha_L(q_L^2 - 2\beta_Lq_L + 2\beta_L^2)$$

**Right tail contribution:**

$$\mathbb{E}[Z^2]_{\text{right}} = (1-\alpha_R)(q_R^2 + 2\beta_Rq_R + 2\beta_R^2)$$

### I.9. Empirical validation on synthetic regression tasks

To validate that `QuantileDistribution` correctly constructs probability distributions from the predicted quantiles of TabICLv2, we design four synthetic regression datasets with known ground-truth distributions. This allows direct comparison between predicted and true distributional quantities, PDF, and CDF.

#### I.9.1. SYNTHETIC REGRESSION DATASETS

**Dataset 1: Quadratic with homoscedastic gaussian noise.** This serves as a simple baseline with constant variance:

$$y = 0.15x^2 - 0.5 + \epsilon, \quad \epsilon \sim \mathcal{N}(0, 0.25^2) \tag{I.1}$$

The true conditional distribution is $p(y|x) = \mathcal{N}(0.15x^2 - 0.5, 0.25^2)$. The predictive distribution should exhibit uniform spread across all $x$ values, with symmetric Gaussian PDFs centered on the quadratic mean function.

**Dataset 2: Sinusoidal with heteroscedastic noise.** This dataset tests the ability of TabICLv2 to capture input-dependent uncertainty:

$$y = \sin(2x) + 0.2x + \epsilon, \quad \epsilon \sim \mathcal{N}(0, \sigma(x)^2), \quad \sigma(x) = 0.12 + 0.1|x| \tag{I.2}$$

The true conditional distribution is $p(y|x) = \mathcal{N}(\sin(2x) + 0.2x, \sigma(x)^2)$. The noise variance increases with $|x|$, requiring the model to predict wider quantile intervals at the boundaries than at the center.

**Dataset 3: Step function with noise.** This dataset tests behavior at discontinuities:

$$y = \begin{cases} -1 + \epsilon & \text{if } x < 0 \\ +1 + \epsilon & \text{if } x \geq 0 \end{cases}, \quad \epsilon \sim \mathcal{N}(0, 0.3^2) \tag{I.3}$$

The true conditional distribution is $p(y|x) = \mathcal{N}(\text{sign}(x), 0.3^2)$. At $x = 0$, the model must handle the abrupt transition between two distinct modes.

**Dataset 4: Linear with heavy-tailed noise.** This dataset introduces heavy-tailed behavior via a Gaussian mixture to test tail extrapolation:

$$y = 0.3x + \epsilon, \quad \epsilon \sim (1 - w)\mathcal{N}(0, \sigma_1^2) + w\mathcal{N}(0, \sigma_2^2) \tag{I.4}$$

where $w = 0.1$ (10% outlier weight), $\sigma_1 = 0.2$ (inlier scale), and $\sigma_2 = 0.8$ (outlier scale). The true conditional distribution is a Gaussian mixture $p(y|x) = 0.9 \cdot \mathcal{N}(0.3x, 0.2^2) + 0.1 \cdot \mathcal{N}(0.3x, 0.8^2)$. This creates heavier tails than a pure Gaussian, testing whether the exponential tail model adequately captures extreme quantiles.

#### I.9.2. VISUALIZATION AND ANALYSIS

Figure I.2 presents a comprehensive 6-row × 4-column visualization comparing predicted distributions (solid lines) against ground-truth distributions (dashed lines). Each column corresponds to one dataset:

1. **Row 1 (quantile lines)**: Training data overlaid with predicted quantile curves. The median tracks the distribution center, while extreme quantiles delineate tail behavior. For the heteroscedastic dataset (column 2), the quantile band visibly widens toward the boundaries $|x| = 3$, correctly capturing input-dependent variance.

2. **Row 2 (quantile functions)**: The quantile function for three representative inputs $x \in \{-2, 0, 2\}$. Predicted curves (solid) closely match the true quantile functions (dashed) across all datasets. The crossing correction is evident, and the exponential tail extrapolation smoothly extends the curves beyond $[\alpha_L, \alpha_R] = [0.001, 0.999]$.

3. **Row 3 (PDF)**: Predicted PDFs (solid) align well with true PDFs (dashed), capturing both the location and spread of the conditional distributions. The heteroscedastic dataset shows narrower peaks at $x = 0$ and broader peaks at $x = \pm 2$, matching the ground truth.

4. **Row 4 (CDF)**: The smooth S-curves of predicted CDFs (solid) closely follow the true CDFs (dashed).

5. **Row 5 (Density heatmaps)**: A 2D visualization of the conditional density $f(y|x)$ across the input domain (log scale). The heteroscedastic dataset shows a funnel-shaped density widening with $|x|$, while the step function exhibits two distinct horizontal bands separated at $x = 0$.

6. **Row 6 (resampled data)**: Synthetic samples drawn from the learned distribution via inverse transform sampling ($y = Q(U)$ where $U \sim \text{Uniform}(0, 1)$). The resampled points (blue) closely match the original training data (gray).

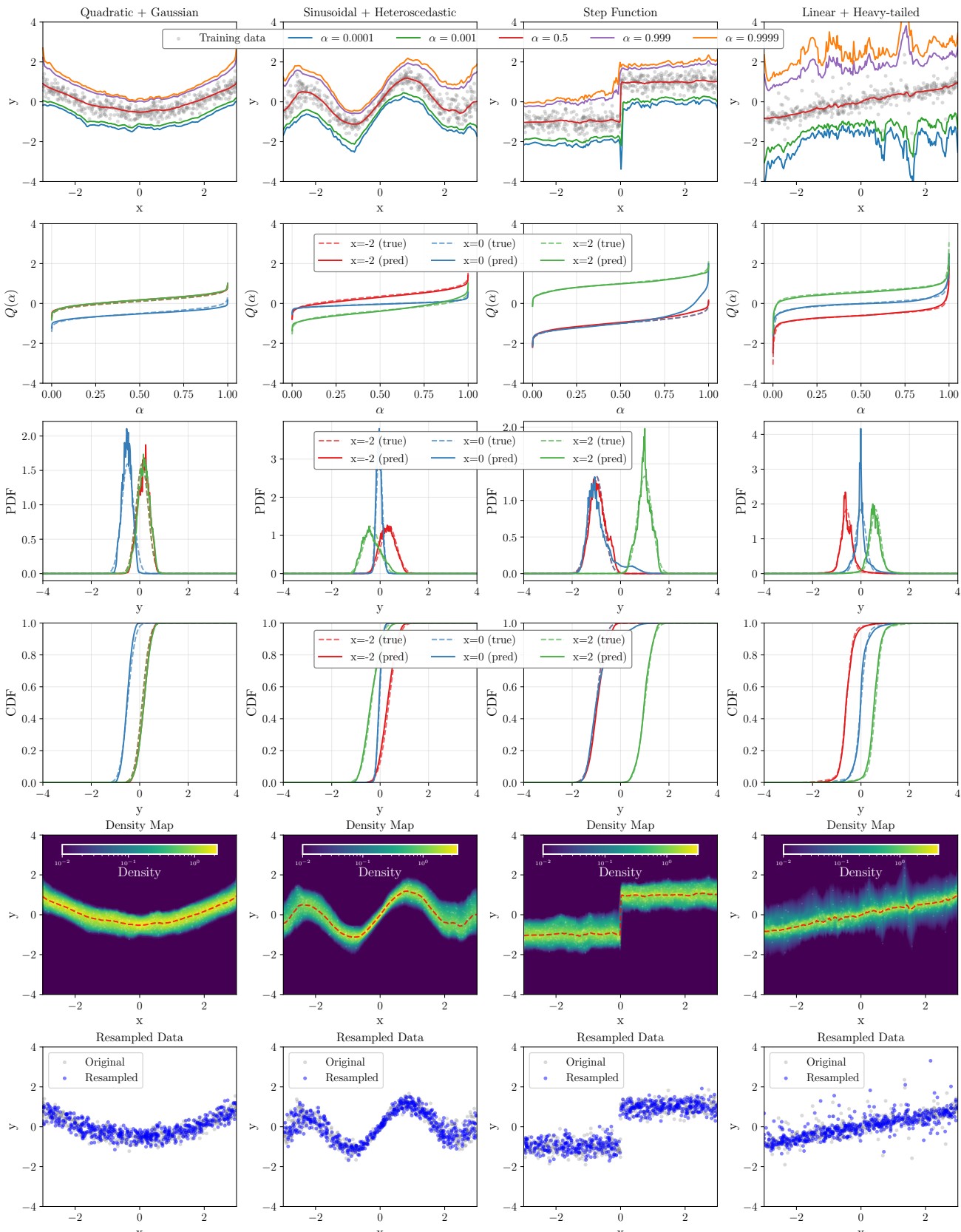

*Figure I.2.* **Validation of `QuantileDistribution` on four synthetic regression tasks with known ground-truth distributions.**

# J. Detailed Results on the TabArena Benchmark

## J.1. Aggregation metrics

TabArena (Erickson et al., 2025) evaluates models using task-specific error metrics and aggregates results across datasets using several complementary metrics. We briefly summarize these metrics below.

**Per-dataset error metrics.** For each dataset $i$, TabArena computes an error metric $\text{err}_i$ by averaging over all outer cross-validation folds:

- **Binary classification**: $1 - \text{ROC AUC}$
- **Multiclass classification**: Log-Loss
- **Regression**: RMSE

**Elo rating.** Elo is a pairwise comparison-based rating system where each model's rating predicts its expected win probability against others. A 400-point Elo gap corresponds to a 10:1 (91%) expected win rate. For two models $A$ and $B$ with ratings $R_A$ and $R_B$, the expected win probability of $A$ is:

$$\mathbb{E}[A \text{ wins}] = \frac{1}{1 + 10^{(R_B - R_A)/400}}$$

Elo is based solely on wins, ties, or losses and ignores the magnitude of performance differences. This ensures each dataset contributes equally to the final ranking, avoiding bias toward certain domains or dataset properties. TabArena calibrates 1000 Elo to the performance of default RandomForest and uses 200 rounds of bootstrapping for 95% confidence intervals.

**Improvability.** Improvability measures the relative error gap between a method and the best-performing method on each dataset, then averages across datasets. For a model $m$ on dataset $i$:

$$\text{Improvability}_i(m) = \frac{\text{err}_i(m) - \text{err}_i^*}{\text{err}_i(m)} \times 100\%$$

where $\text{err}_i^* = \min_{m'} \text{err}_i(m')$ is the error of the best method on dataset $i$. Improvability is always between 0% (optimal) and 100%. Unlike Elo, improvability is sensitive to the magnitude of performance differences, making it more informative for practitioners who care about how much a method lags behind the best.

**Average rank.** For each dataset, models are ranked by their error (rank 1 is best). The average rank is simply the mean rank across all datasets:

$$\text{AvgRank}(m) = \frac{1}{|\mathcal{D}|} \sum_{i \in \mathcal{D}} \text{rank}_i(m)$$

Lower average rank indicates better overall performance.

**Discussion.** Each aggregation metric has its own strengths and limitations. Elo treats all datasets equally regardless of performance gaps, improvability captures the magnitude of differences, and rank-based metrics are robust to outliers. We primarily report improvability in the main paper for its interpretability, but provide Elo and rankings in this appendix for completeness. Across all metrics, TabICLv2 consistently achieves state-of-the-art performance.

## J.2. Results on all datasets

Figures in the section present the complete results on all 51 TabArena datasets.

**Ranking and Elo.** TabICLv2 (default) achieves an average rank of 4.82, outperforming AutoGluon 1.4 (extreme, 4h) at 5.24 and RealTabPFN-2.5 (T+E) at 5.88. Here, AutoGluon 1.4 (extreme, 4h) refers to AutoGluon 1.4 (Erickson et al., 2020) with the `extreme_quality` preset and a 4-hour training budget, which ensembles multiple model families with extensive hyperparameter tuning. Despite a single forward-pass without any tuning, TabICLv2 achieves better performance than this heavily optimized ensemble system.

**Pairwise win rates.**    While TabICLv2 ranks below AutoGluon 1.5 (extreme, 4h), an automated machine learning system that tunes and ensembles over multiple tabular models, in average rank (4.82 vs. 3.88), the pairwise win rate matrix (Figure J.2) reveals a more nuanced picture: TabICLv2 achieves a 57% win rate against AutoGluon 1.5 and a 59% win rate against RealTabPFN-2.5 (T+E). This indicates that TabICLv2 wins on the majority of datasets in head-to-head comparisons.

The discrepancy between win rate and average rank arises from how these metrics handle the magnitude of performance differences. Win rate only counts wins and losses, treating all victories equally. In contrast, average rank penalizes methods that perform poorly on certain datasets, even if they win on most others. This suggests that while TabICLv2 wins more often, AutoGluon 1.5 may achieve more consistent rankings across datasets, avoiding the occasional poor performance that can inflate average rank. This suggests that TabICLv2 may struggle on specific datasets that fall outside its pretraining distribution.

**Pareto efficiency.**    As shown in Figures J.3 and J.4, TabICLv2 dominates the Pareto front of both improvability vs. runtime and Elo vs. runtime among all tabular foundation models. TabICLv2 achieves the best trade-off between predictive performance and computational cost, being both faster and more accurate than competing TFMs including RealTabPFN-2.5, TabPFN-2.5, TabICL, LimiX, and Mitra.

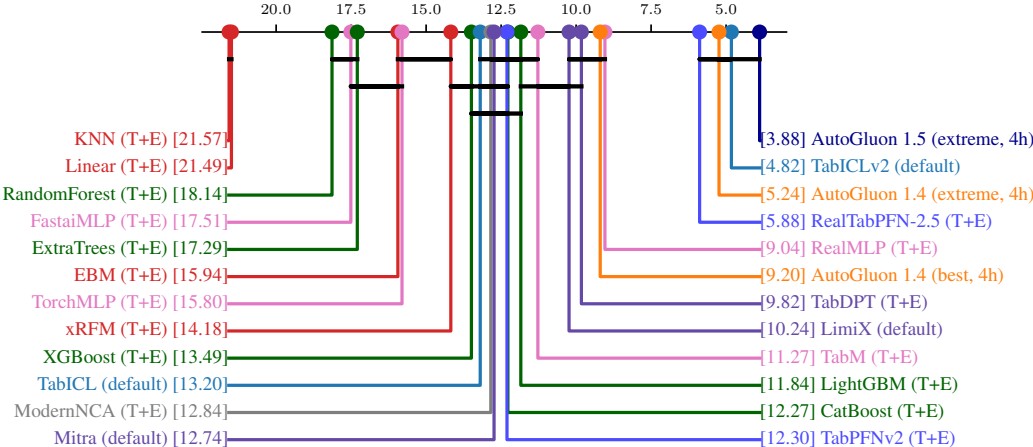

*Figure J.1.* **Critical difference diagram on the TabArena benchmark.**

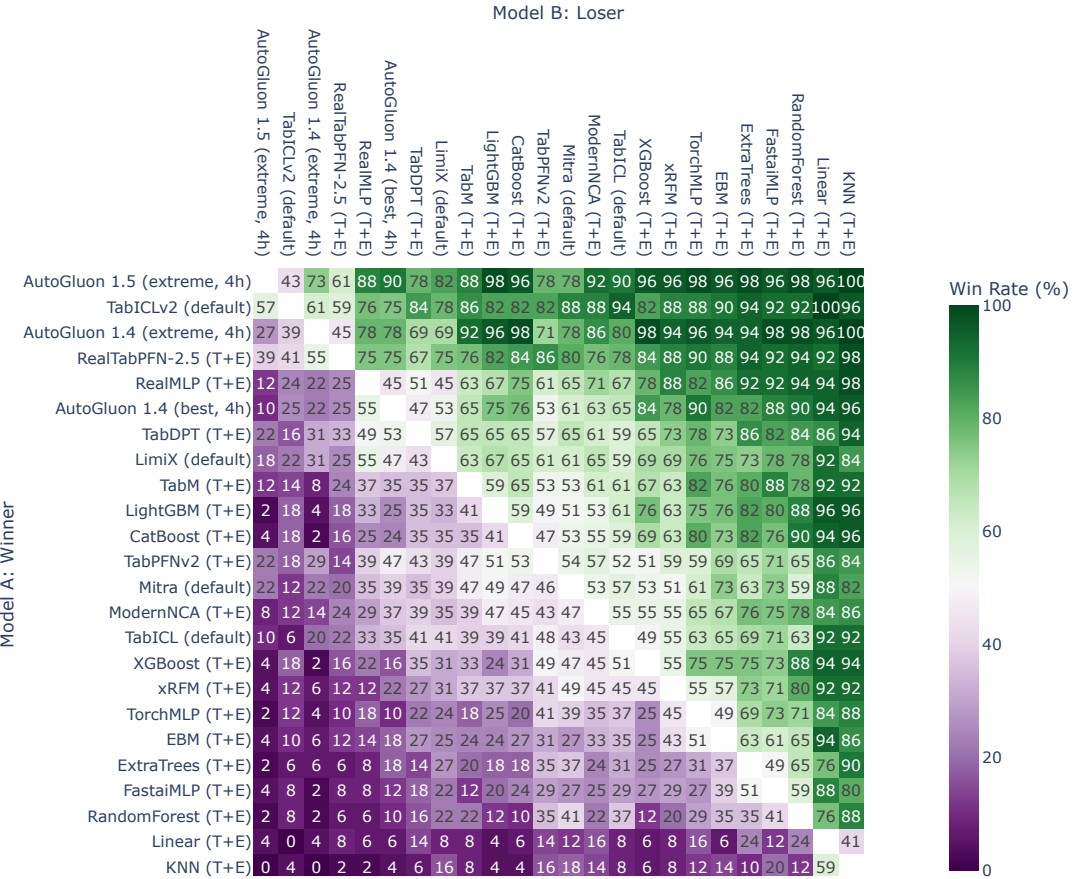

*Figure J.2.* **Win-rate matrix on the TabArena benchmark.**

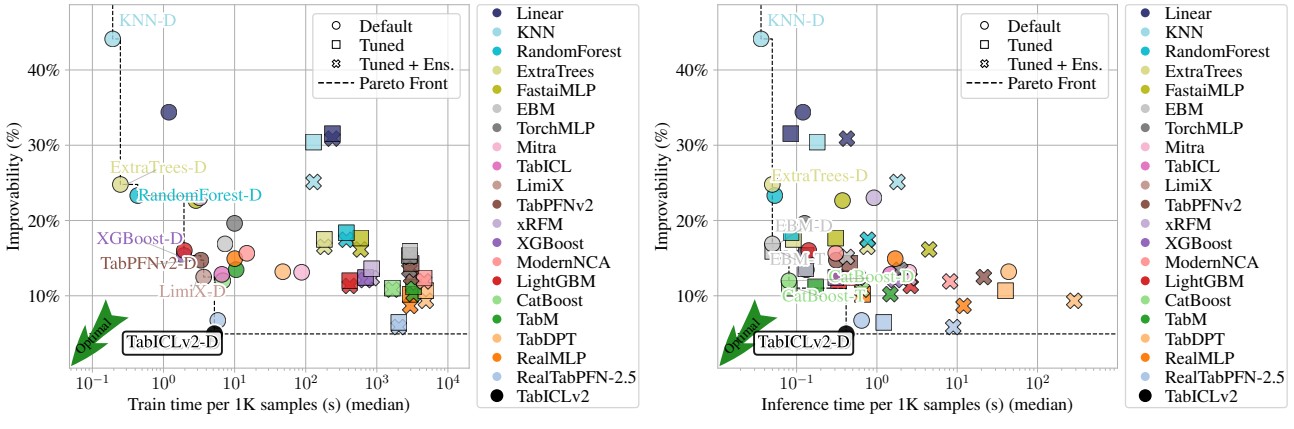

*(a)* Pareto front of improvability and train time

*(b)* Pareto front of improvability and inference time

*Figure J.3.* **Pareto front of improvability and train/inference time on the TabArena benchmark.**

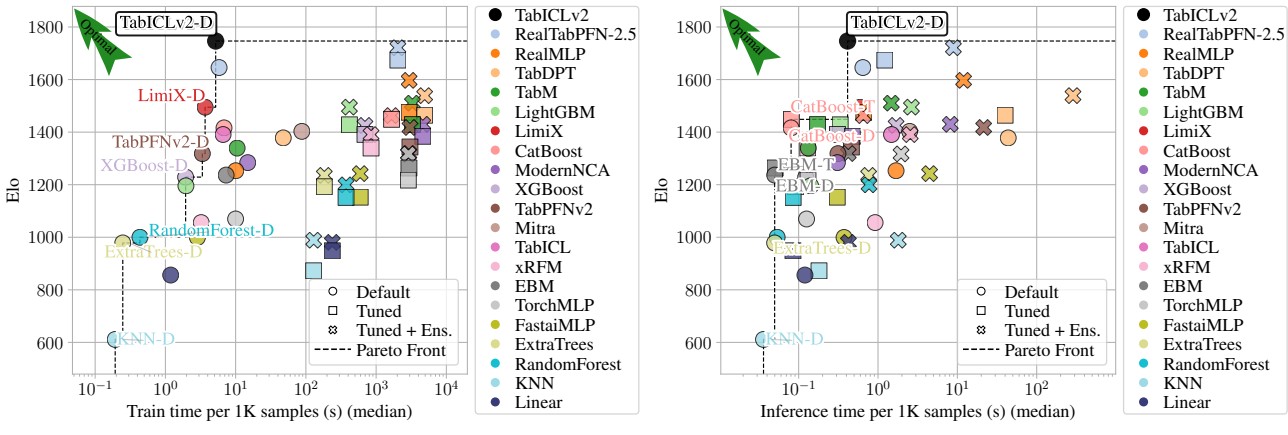

*(a)* Pareto front of Elo and train time        *(b)* Pareto front of Elo and inference time

*Figure J.4.* **Pareto front of Elo and train/inference time on the TabArena benchmark.**

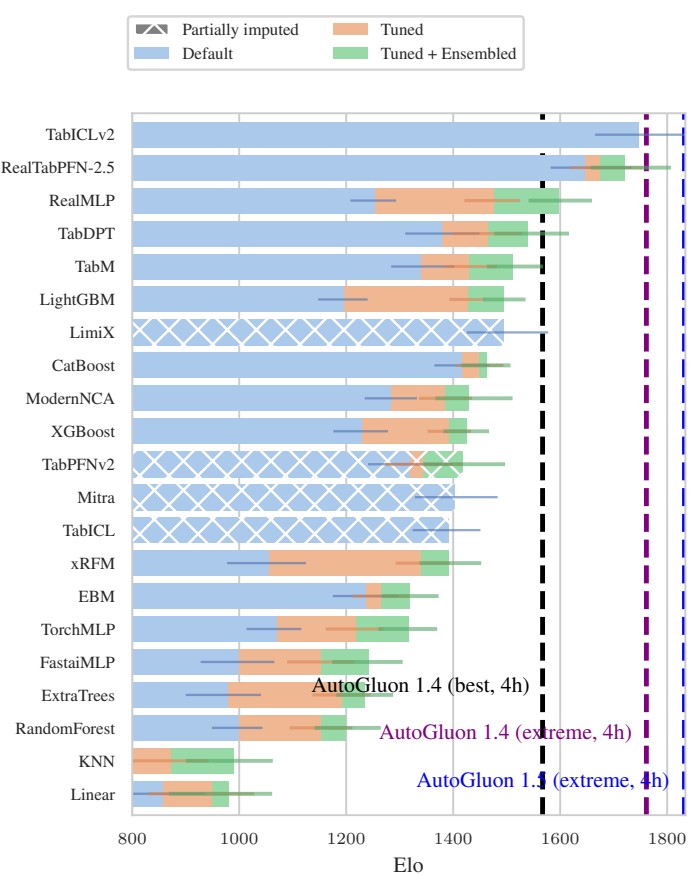

*Figure J.5.* **TabArena Elo.**

## J.3. Results on binary classification datasets

Figures in this section present results on 24 binary classification datasets in TabArena. TabICLv2 achieves an average rank of 4.43, placing second behind AutoGluon 1.5 (extreme, 4h) at 3.83. Notably, TabICLv2 substantially outperforms RealTabPFN-2.5 (T+E) at 6.90. On the Pareto fronts, TabICLv2 remains the strongest among all tabular foundation models for binary classification.

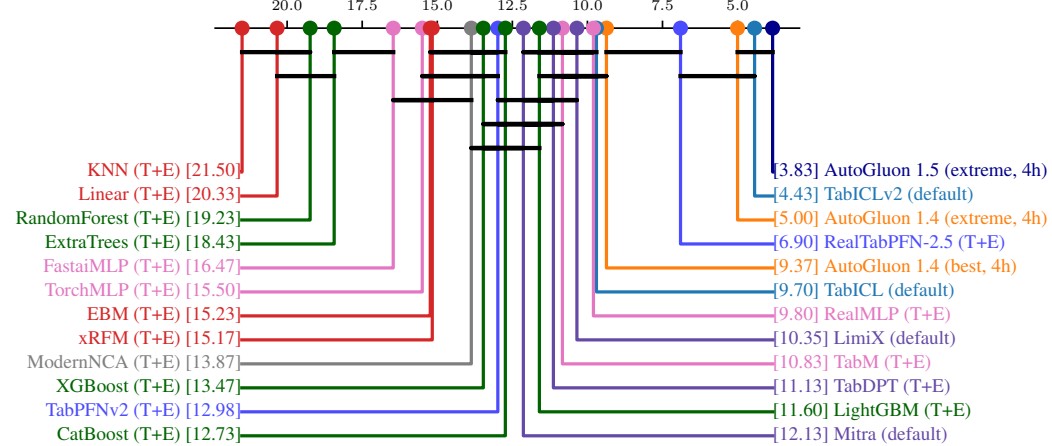

*Figure J.6.* **Critical difference diagram on binary classification datasets of the TabArena benchmark.**

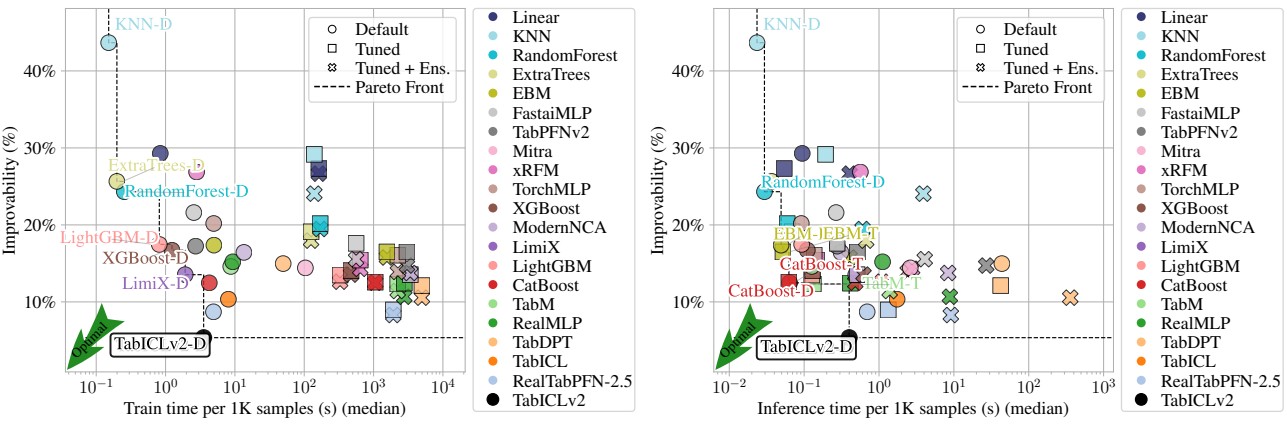

*(a)* Pareto front of improvability and train time

*(b)* Pareto front of improvability and inference time

*Figure J.7.* **Pareto front of improvability and train/inference time on binary classification datasets of the TabArena benchmark.**

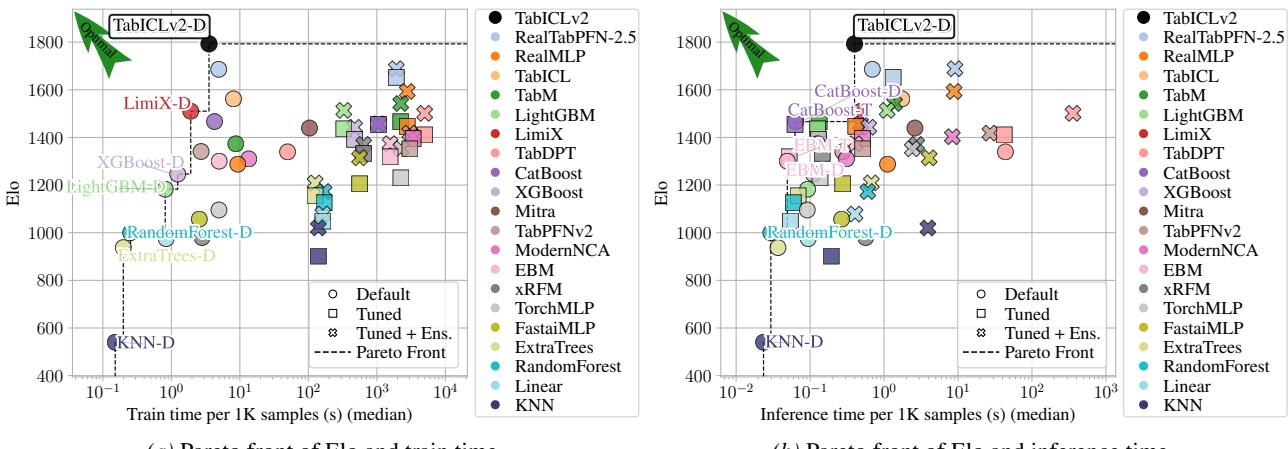

*(a)* Pareto front of Elo and train time         *(b)* Pareto front of Elo and inference time

*Figure J.8.* **Pareto front of Elo and train/inference time on binary classification datasets of the TabArena benchmark.**

## J.4. Results on multiclass classification datasets

Figures in the section present results on 14 multiclass classification datasets in TabArena. On multiclass classification, TabICLv2 (default) achieves an average rank of 6.75, behind AutoGluon 1.5 (extreme, 4h) at 4.00, RealTabPFN-2.5 (T+E) at 4.25, and AutoGluon 1.4 (extreme, 4h) at 4.50. While TabICLv2 does not surpass RealTabPFN-2.5 (T+E) on this subset, it is important to note that RealTabPFN-2.5 employs tuning and ensembling whereas TabICLv2 does not perform any tuning. In addition, TabICLv2 substantially outperforms other TFMs, such as TabPFNv2 (T+E) at 8.62 and LimiX (default) at 9.31.

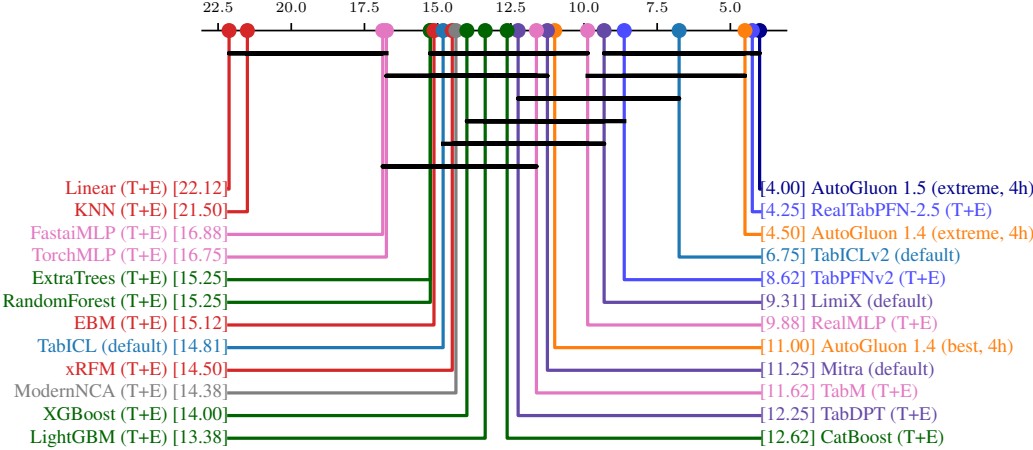

*Figure J.9.* **Critical difference diagram on multiclass classification datasets of the TabArena benchmark.**

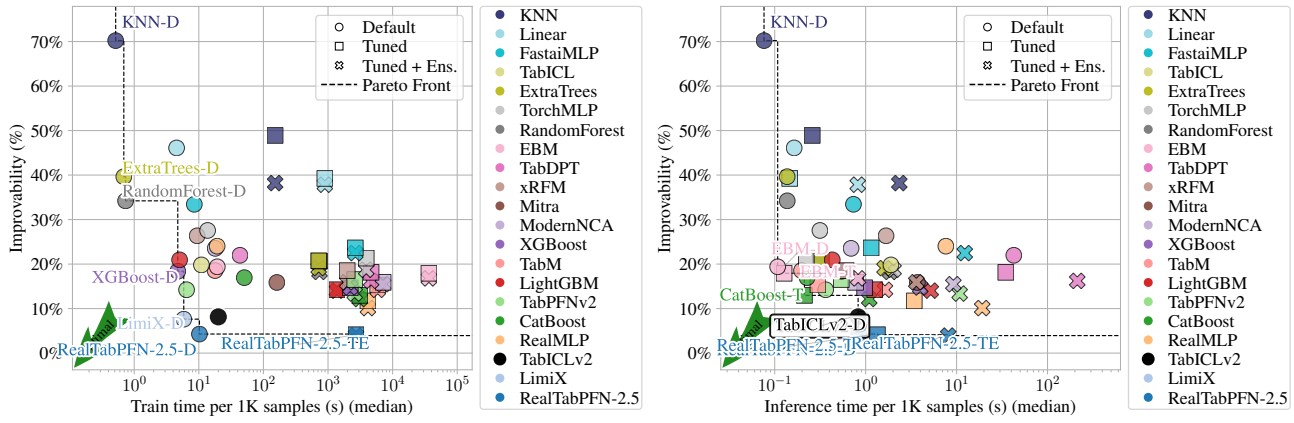

*(a)* Pareto front of improvability and train time

*(b)* Pareto front of improvability and inference time

*Figure J.10.* **Pareto front of improvability and train/inference time on multiclass classification datasets of the TabArena benchmark.**

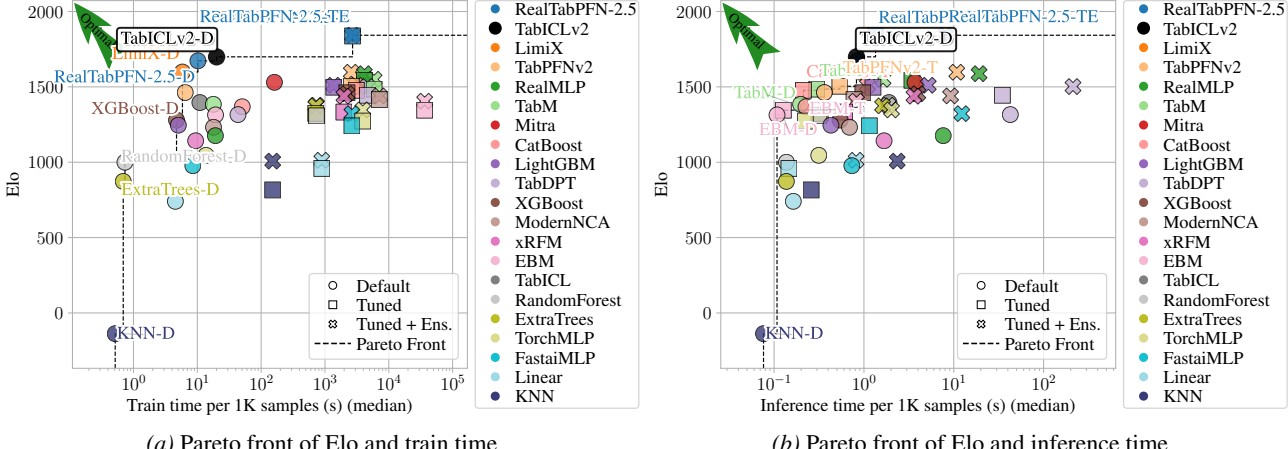

*(a)* Pareto front of Elo and train time

*(b)* Pareto front of Elo and inference time

*Figure J.11.* **Pareto front of Elo and train/inference time on multiclass classification datasets of the TabArena benchmark.**

## J.5. Results on regression datasets

Figures in this section present results on 13 regression datasets in TabArena. On regression tasks, TabICLv2 (default) achieves an average rank of 4.54, tying with RealTabPFN-2.5 (T+E) and trailing only AutoGluon 1.5 (extreme, 4h) at 3.92. Interestingly, TabDPT (T+E) achieves a competitive rank of 5.31 on regression, substantially better than its performance on classification tasks where it ranks much lower. TabDPT is pretrained on real-world datasets rather than synthetic data. However, its strong regression performance raises questions about potential data leakage between the training corpus of TabDPT and the regression datasets of TabArena.

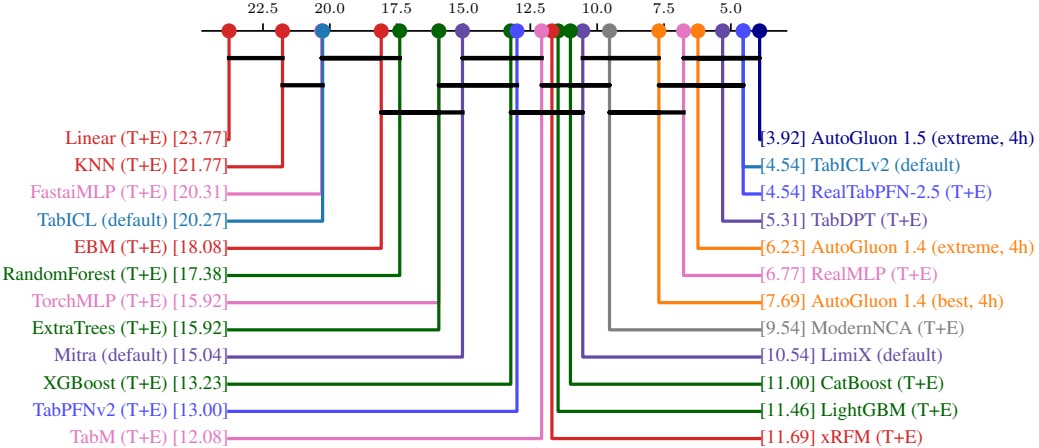

*Figure J.12.* **Critical difference diagram on regression datasets of the TabArena benchmark.**

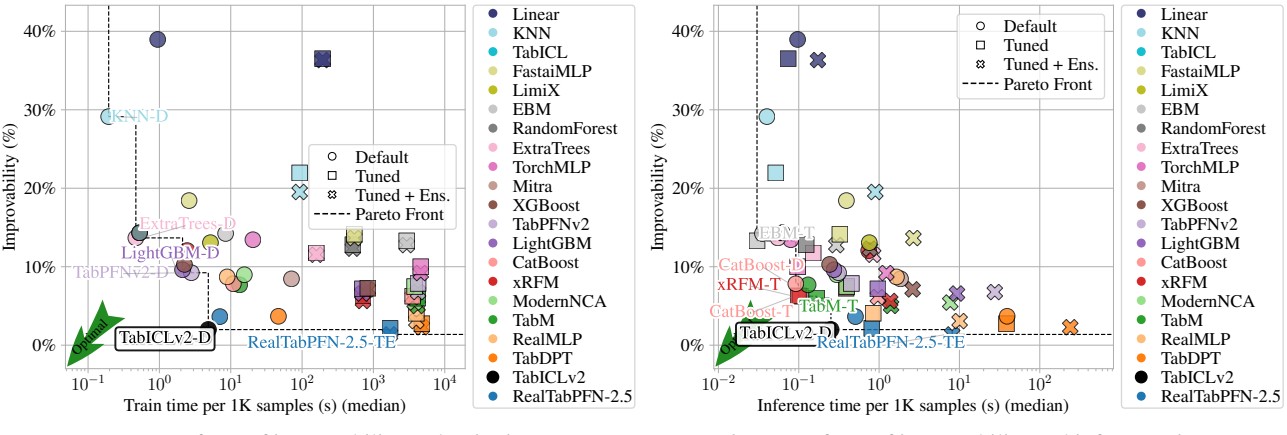

*(a)* Pareto front of improvability and train time        *(b)* Pareto front of improvability and inference time

*Figure J.13.* **Pareto front of improvability and train/inference time on regression datasets of the TabArena benchmark.**

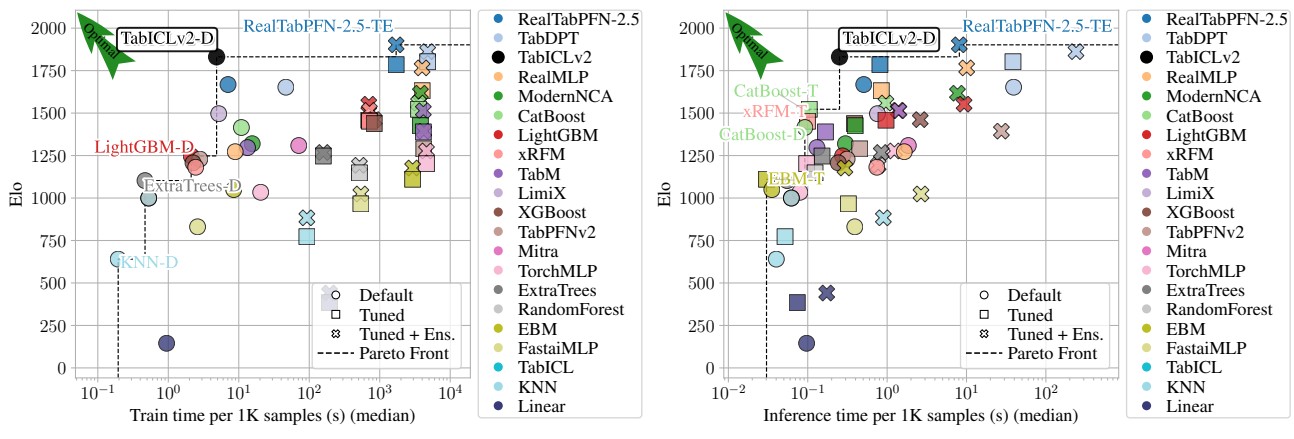

(a) Pareto front of Elo and train time      (b) Pareto front of Elo and inference time

*Figure J.14.* **Pareto front of Elo and train/inference time on regression datasets of the TabArena benchmark.**

# K. Detailed Results on the TALENT Benchmark

## K.1. Benchmark overview

TALENT (Ye et al., 2024) comprises 300 datasets spanning three task types:

- **Binary classification**: 120 datasets
- **Multiclass classification**: 80 datasets
- **Regression**: 100 datasets

Each dataset is split into 64%/16%/20% for training, validation, and test sets, respectively. Hyperparameters are selected on the validation set based on accuracy, and final performance is reported on the held-out test set.

**Evaluation metrics.** Following the TALENT protocol, we use accuracy as the primary metric for classification tasks and RMSE for regression tasks. For aggregating results across datasets, we compute:

- **Improvability**: The relative error gap to the best method, using $1 -$ accuracy (classification) and RMSE (regression).
- **Elo**: Pairwise comparison-based rating using accuracy (classification) and negative RMSE (regression).
- **Average rank**: Mean rank across datasets based on accuracy (classification) or RMSE (regression).

TALENT also provides supplementary metrics including log-loss and AUC for classification, and MAE and $R^2$ for regression. In the following subsections, we present detailed results stratified by task type and dataset characteristics.

Note that, for a fair comparison, we exclude the development datasets from the main paper used for the development of TabICLv2.

## K.2. Results on all datasets

TabICLv2 achieves the best average rank of 4.66, outperforming RealTabPFN-2.5 (5.11) and TabPFN-2.5 (5.45). The pairwise win rates further confirm the dominance of TabICLv2: 62% against RealTabPFN-2.5 and 65% against TabPFN-2.5.

On TALENT, we evaluate both RealTabPFN-2.5 (fine-tuned on real data) and TabPFN-2.5 (not fine-tuned). RealTabPFN-2.5 outperforms TabPFN-2.5 (5.11 vs. 5.45), demonstrating that fine-tuning on real-world data provides measurable benefits. Note that TabArena does not report results for TabPFN-2.5, making TALENT a valuable benchmark for isolating the effect of fine-tuning on TabPFN-2.5.

TabICLv2 substantially outperforms other tabular foundation models. LimiX and TabPFNv2 achieve average ranks of 8.34 and 8.82, respectively, nearly twice that of TabICLv2.

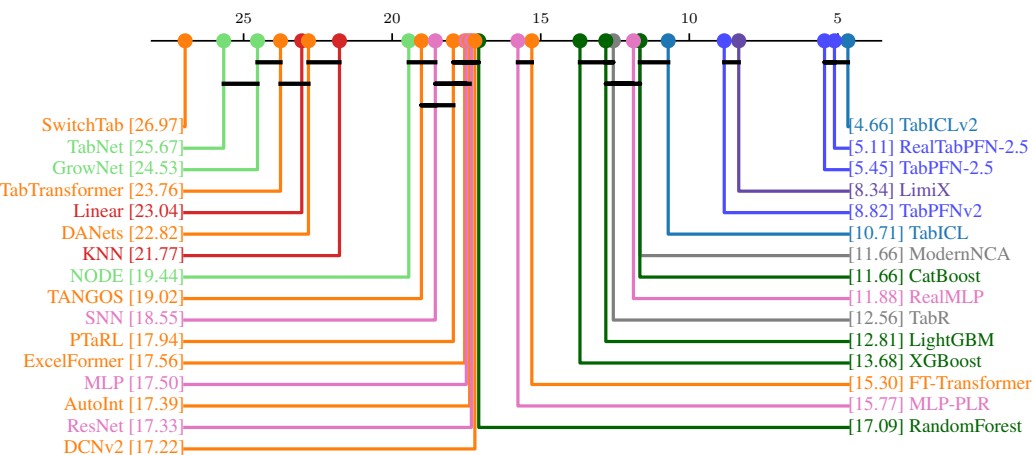

*Figure K.1.* **Critical difference diagram on the TALENT benchmark.**

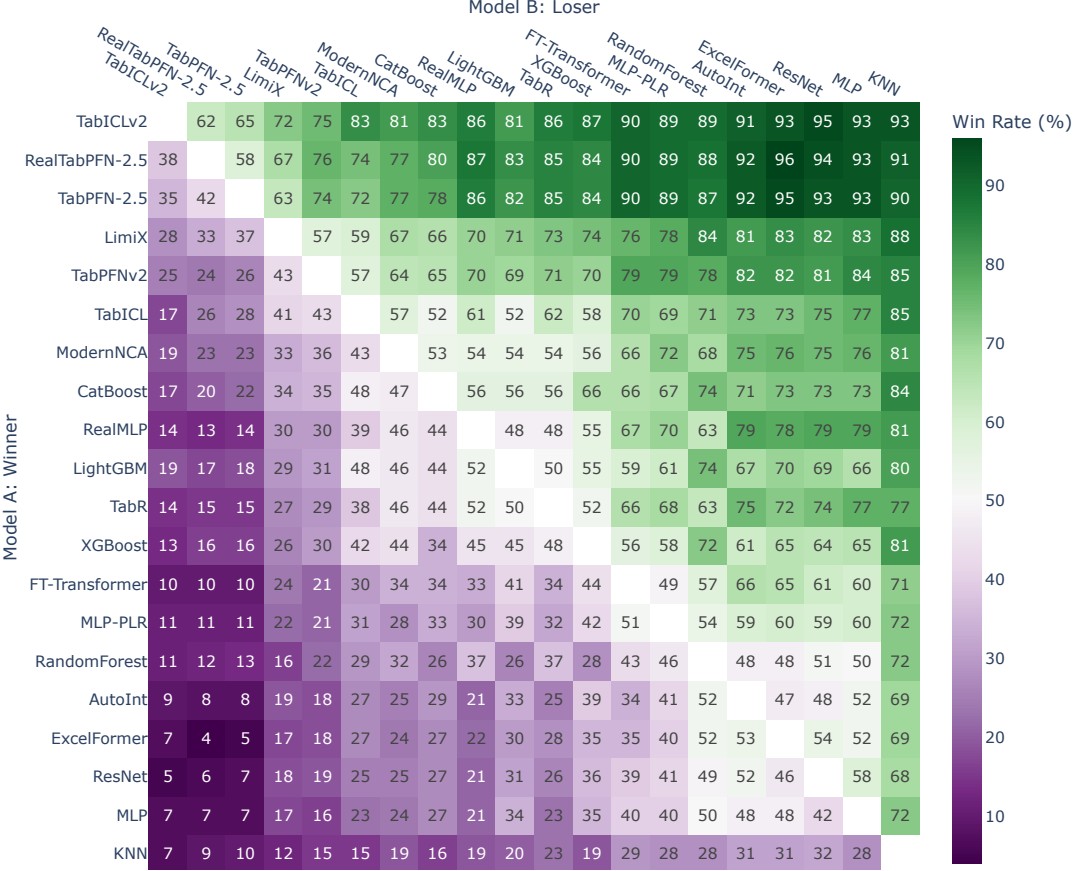

*Figure K.2.* **Win-rate matrix on the TALENT benchmark.**

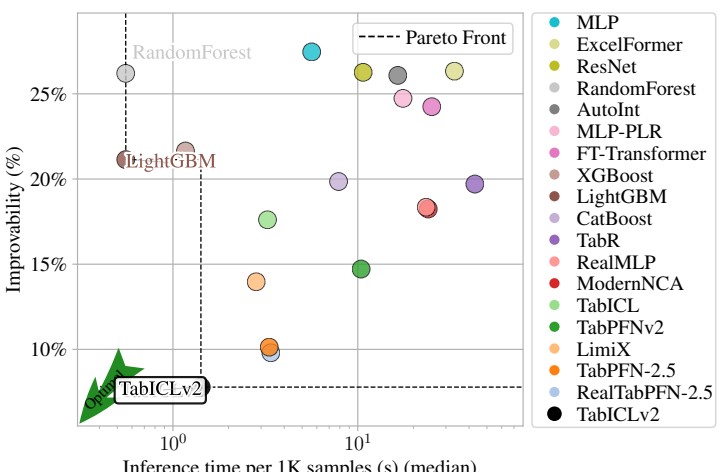

*Figure K.3.* **Pareto front of improvability and inference time on the TALENT benchmark.**

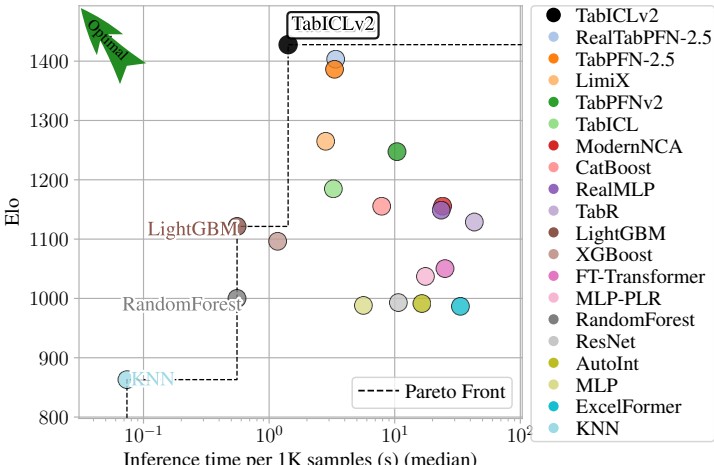

*Figure K.4.* **Pareto front of Elo and inference time on the TALENT benchmark.**

### K.3. Results on binary classification datasets

For binary classification datasets, TabICLv2 (4.98) is essentially tied with RealTabPFN-2.5 (4.82) on accuracy, both outperforming TabPFN-2.5 (5.28), TabICL (8.48), and LimiX (8.61). However, on AUC and log-loss, TabICLv2 achieves a clear lead:

- **AUC**: TabICLv2 (3.31) vs. RealTabPFN-2.5 (4.62) vs. TabPFN-2.5 (5.45)
- **log-loss**: TabICLv2 (2.83) vs. RealTabPFN-2.5 (3.78) vs. TabPFN-2.5 (4.31)

The gap between accuracy and probabilistic metrics (AUC, log-loss) reveals important differences in model behavior. Accuracy only evaluates predictions at a fixed decision threshold, whereas AUC measures ranking quality across all thresholds and log-loss evaluates probability calibration. The fact that TabICLv2 outperforms RealTabPFN-2.5 on AUC and log-loss while matching on accuracy suggests that TabICLv2 produces better probability estimates. This is valuable in practice, where better probabilities enable reliable uncertainty quantification and informed decision-making beyond simple class predictions.

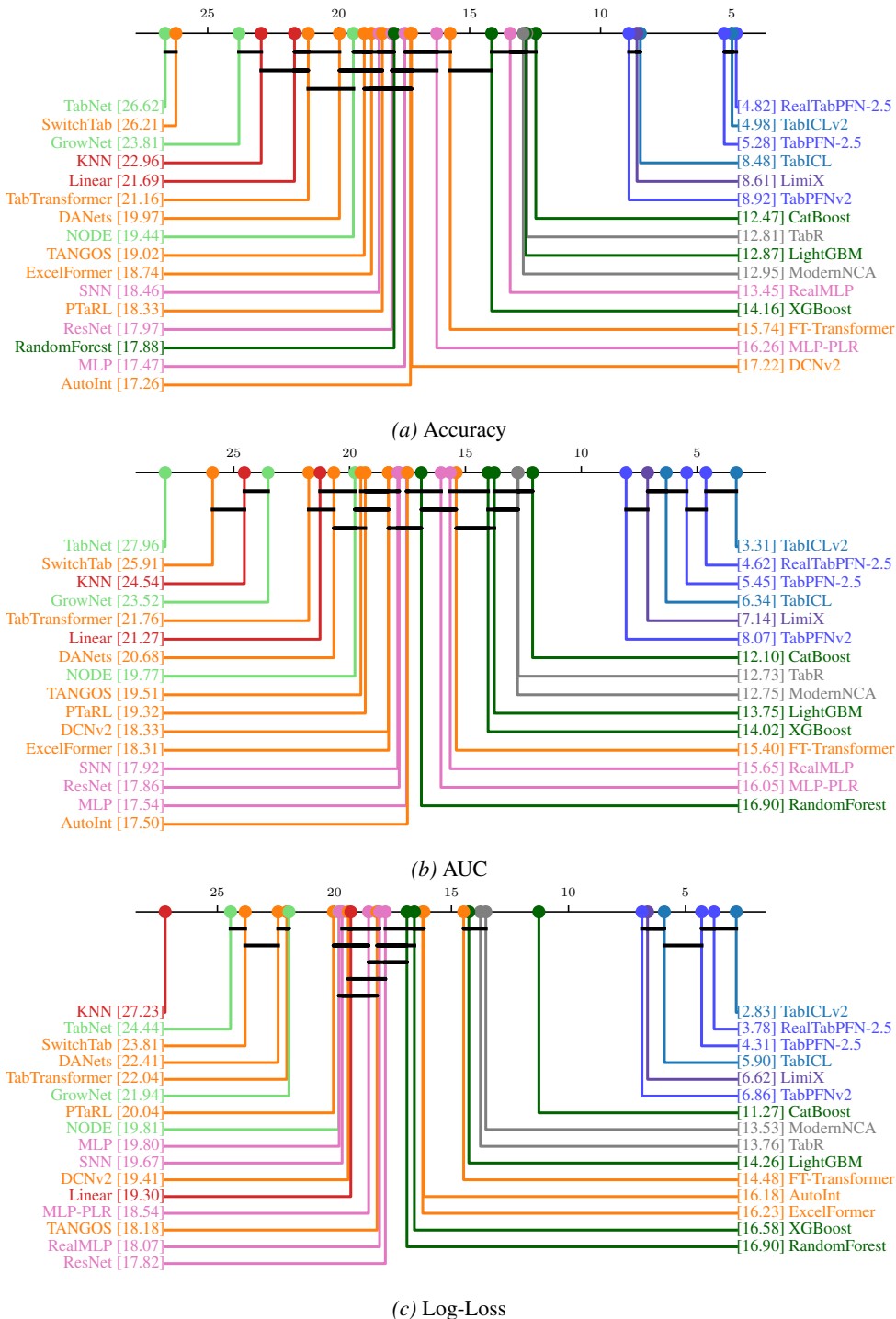

*(a)* Accuracy

*(b)* AUC

*(c)* Log-Loss

*Figure K.5.* **Critical difference diagram on binary classification datasets of the TALENT benchmark.**

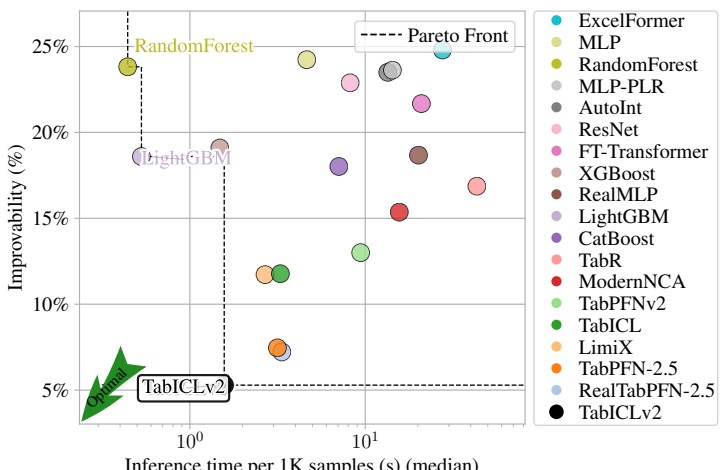

*Figure K.6.* **Pareto front of improvability and inference time on binary classification datasets of the TALENT benchmark.**

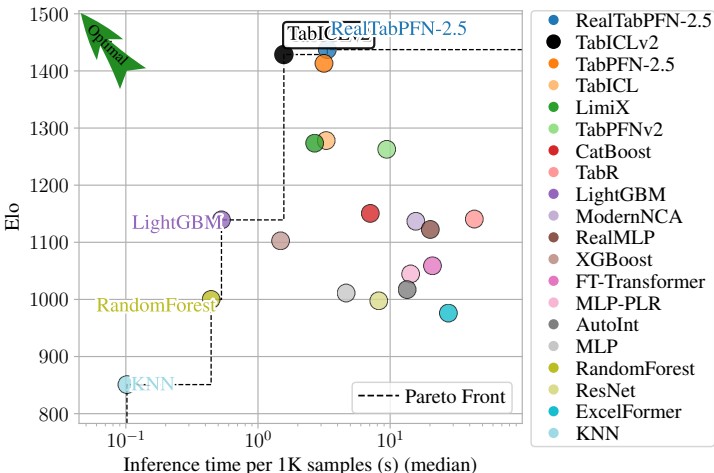

*Figure K.7.* **Pareto front of Elo and inference time on binary classification datasets of the TALENT benchmark.**

**K.4. Results on multiclass classification datasets ($\leq$ 10 classes)**

Unlike binary classification where TabICLv2 and RealTabPFN-2.5 are comparable on accuracy, TabICLv2 achieves clear superiority on multiclass tasks across all evaluation metrics:

- **Accuracy**: TabICLv2 (4.58) vs. RealTabPFN-2.5 (5.64)
- **AUC**: TabICLv2 (3.38) vs. RealTabPFN-2.5 (5.20)
- **Log-loss**: TabICLv2 (2.67) vs. RealTabPFN-2.5 (4.48)

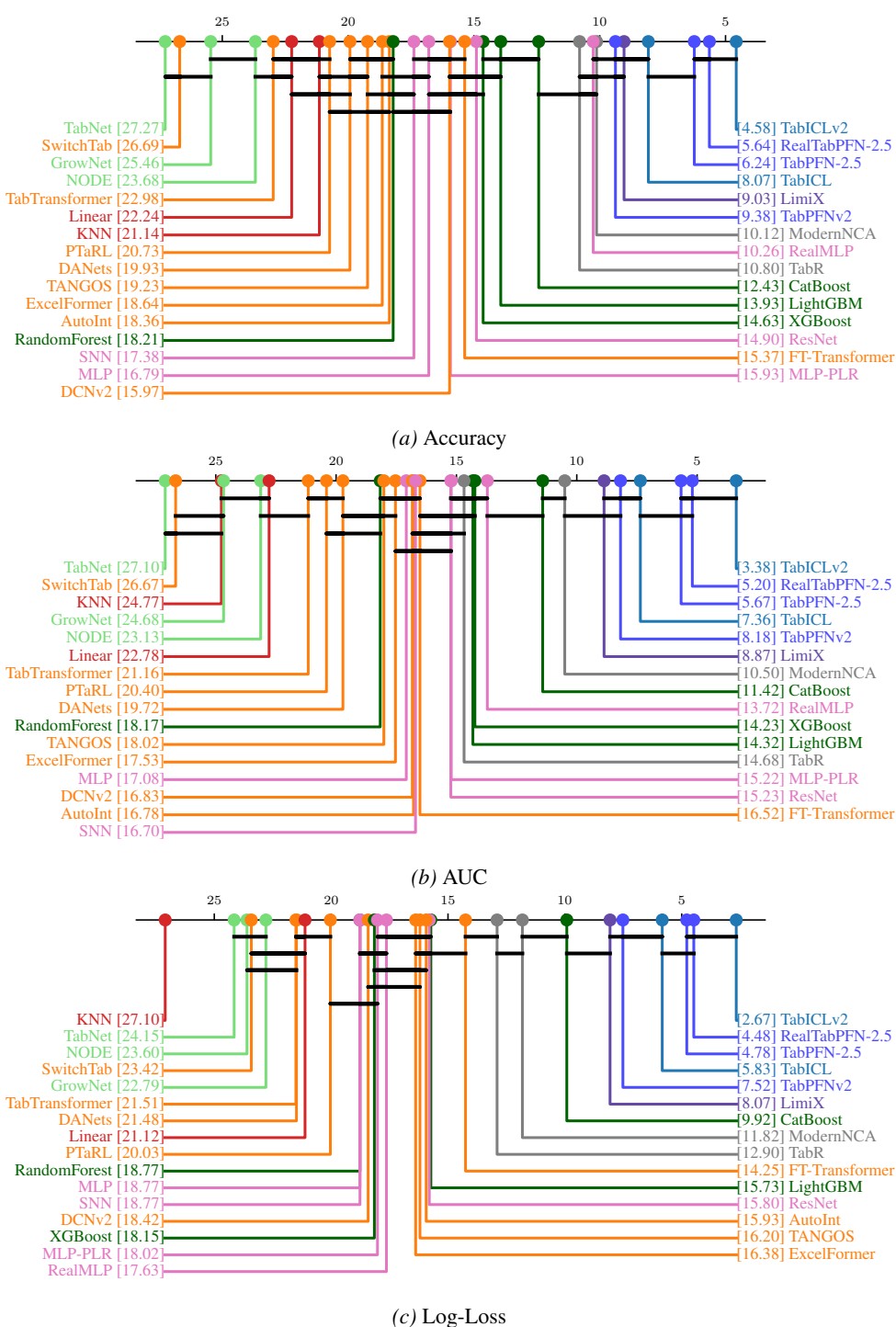

*(a)* Accuracy

*(b)* AUC

*(c)* Log-Loss

*Figure K.8.* **Critical difference diagram on multiclass classification datasets ($\leq$ 10 classes) of the TALENT benchmark.**

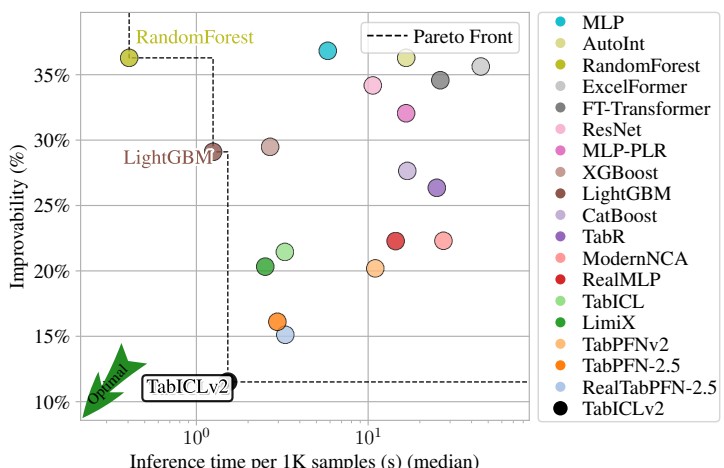

*Figure K.9.* **Pareto front of improvability and inference time on multiclass classification datasets of the TALENT benchmark.**

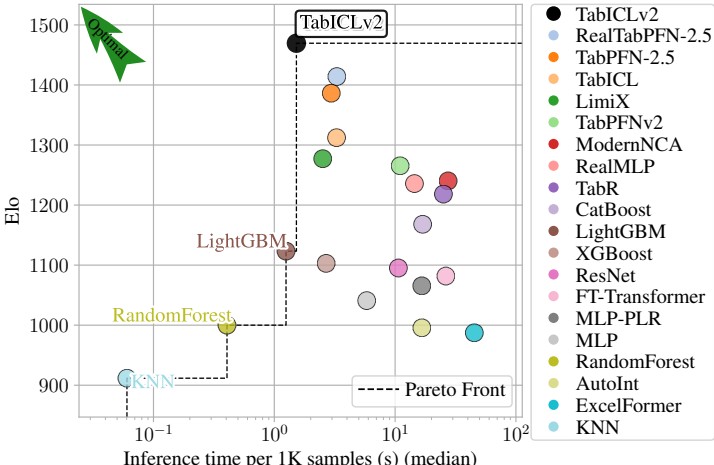

*Figure K.10.* **Pareto front of Elo and inference time on multiclass classification datasets ($\leq$ 10 classes) of the TALENT benchmark.**

## K.5. Results on multiclass classification datasets ($>$ 10 classes)

This section presents results on 12 multiclass classification datasets with more than 10 classes in TALENT. TabICLv2 clearly outperforms RealTabPFN-2.5 on many-class classification. Here, ECOC refers to the error-correcting output codes wrapper from TabPFNv2 (Hollmann et al., 2025). TabICLv2 achieves strong performance both with the ECOC wrapper and with its native many-class handling via mixed-radix ensembling. Both variants outperform RealTabPFN-2.5-ECOC.

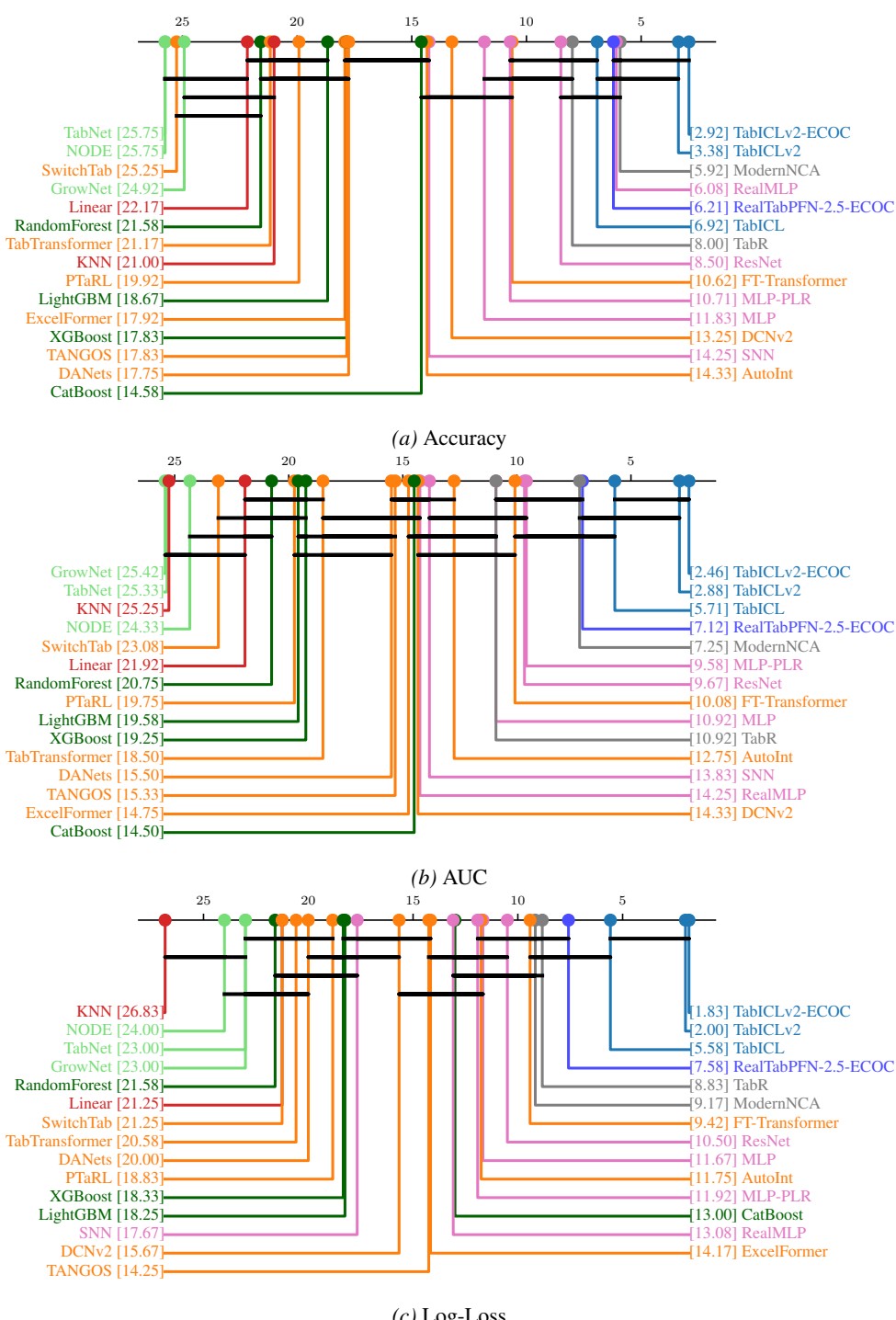

*(a)* Accuracy

*(b)* AUC

*(c)* Log-Loss

*Figure K.11.* **Critical difference diagram on multiclass classification datasets ($>$ 10 classes) of the TALENT benchmark.**

### K.6. Results on regression datasets

TabICLv2 outperforms TabPFN-2.5 on RMSE and $R^2$, while TabPFN-2.5 has a slight edge on MAE. RMSE penalizes large errors, making it sensitive to outliers and tail behavior, while MAE treats all errors equally. The quantile regression of TabICLv2, trained with pinball loss across 999 quantiles, is designed to capture the full conditional distribution rather than optimizing for a single point estimate. This distributional focus leads to better performance on RMSE and $R^2$, which reward accurate modeling of variance and extreme values. The slight disadvantage on MAE suggests that the bin-based approach of TabPFN-2.5 may be marginally better optimized for median prediction, though the difference is small (4.43 vs. 4.63).

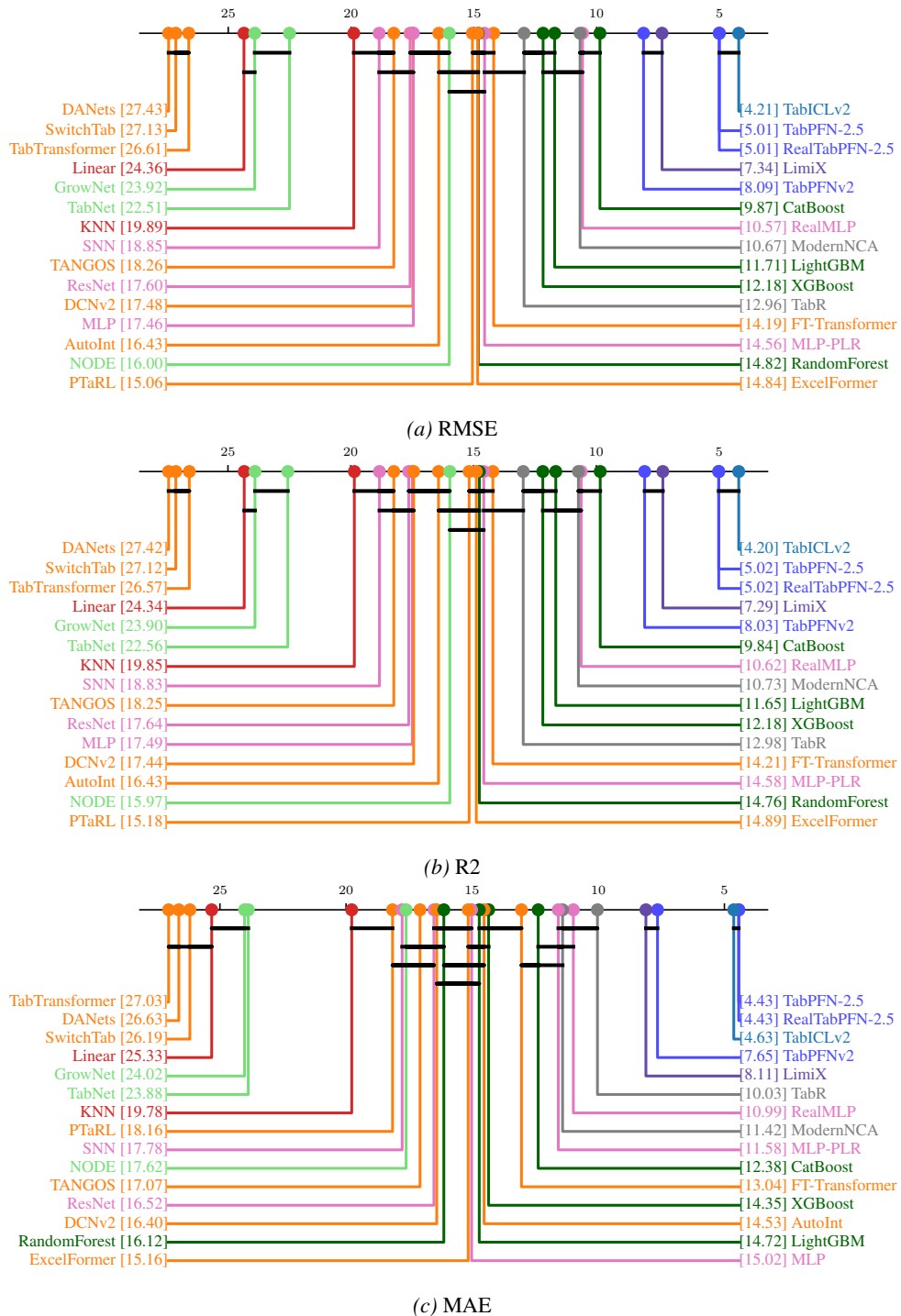

Figure K.12. **Critical difference diagram on regression datasets of the TALENT benchmark.**

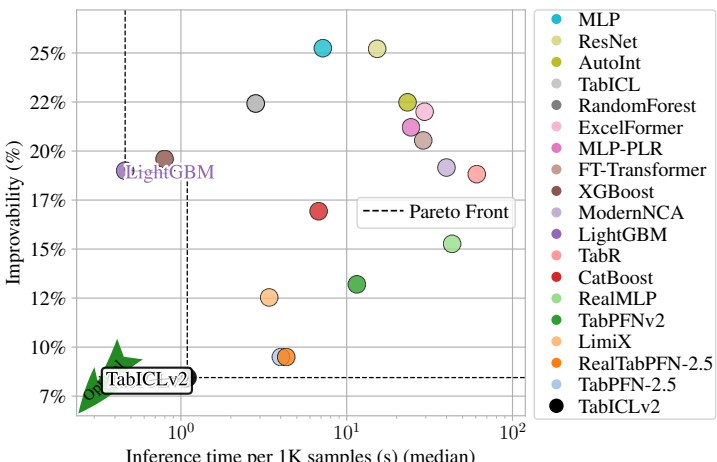

*Figure K.13.* **Pareto front of improvability and inference time on regression datasets of the TALENT benchmark.**

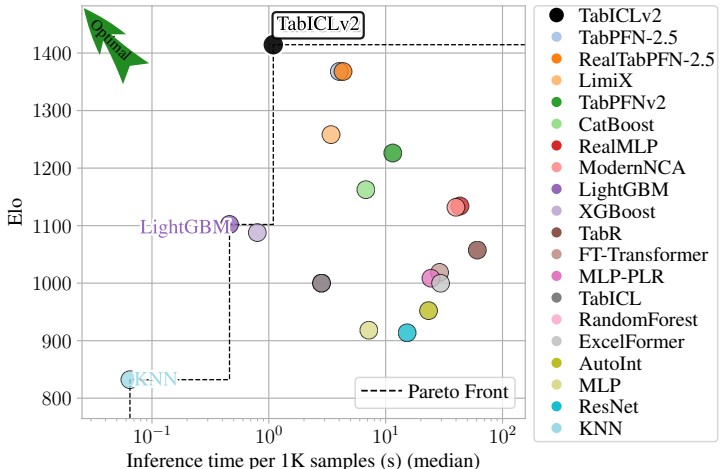

*Figure K.14.* **Pareto front of Elo and inference time on regression datasets of the TALENT benchmark.**

## K.7. Results on small datasets with less than 10K samples

TabICLv2 and RealTabPFN-2.5 achieve comparable performance on small datasets, the regime where tabular foundation models are originally designed to excel.

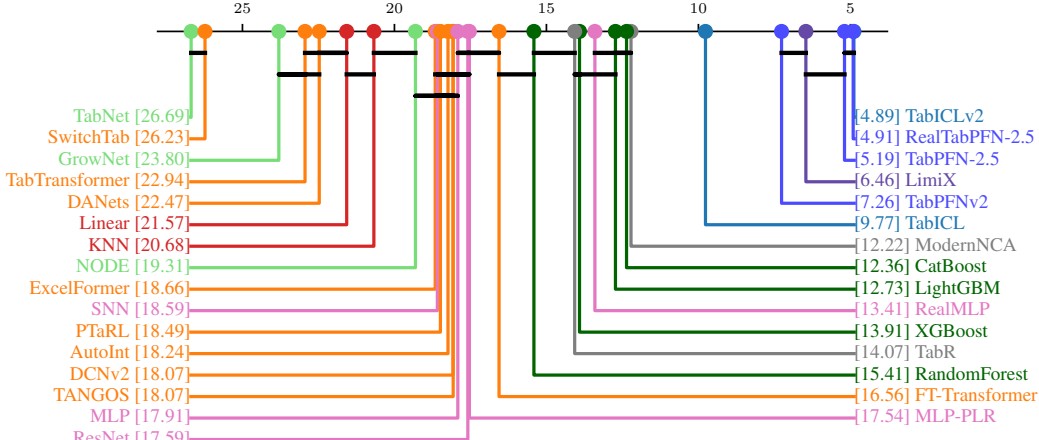

*Figure K.15.* **Critical difference diagram on small datasets of the TALENT benchmark.**

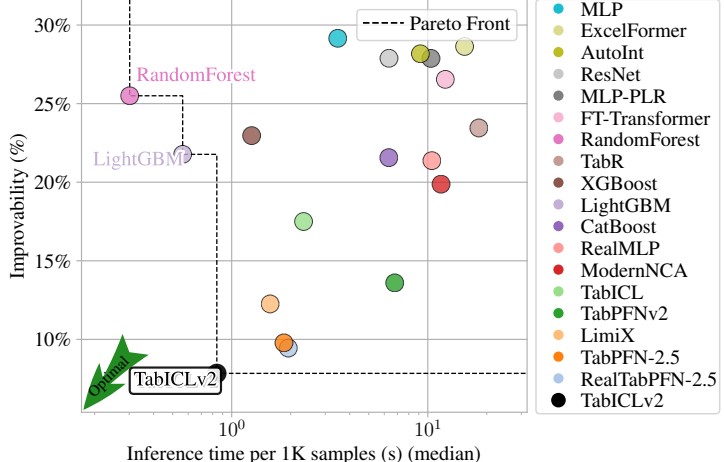

*Figure K.16.* **Pareto front of improvability and inference time on small datasets of the TALENT benchmark.**

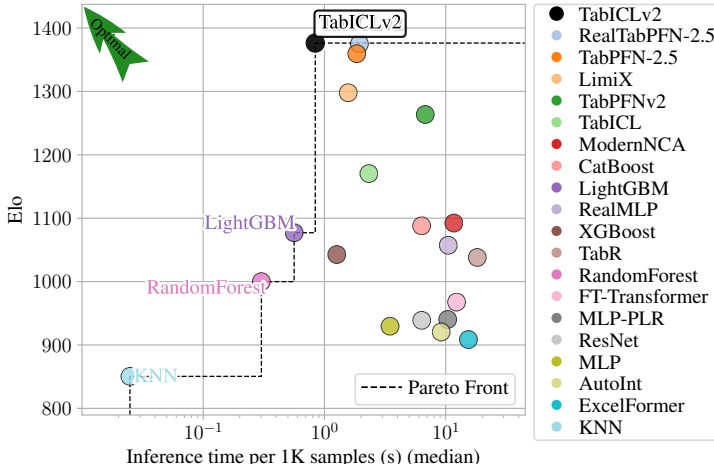

*Figure K.17.* **Pareto front of Elo and inference time on small datasets of the TALENT benchmark.**

## K.8. Results on large datasets with more than 10K samples

On large datasets, TabICLv2 demonstrates a clear advantage over both RealTabPFN-2.5 and TabPFN-2.5.

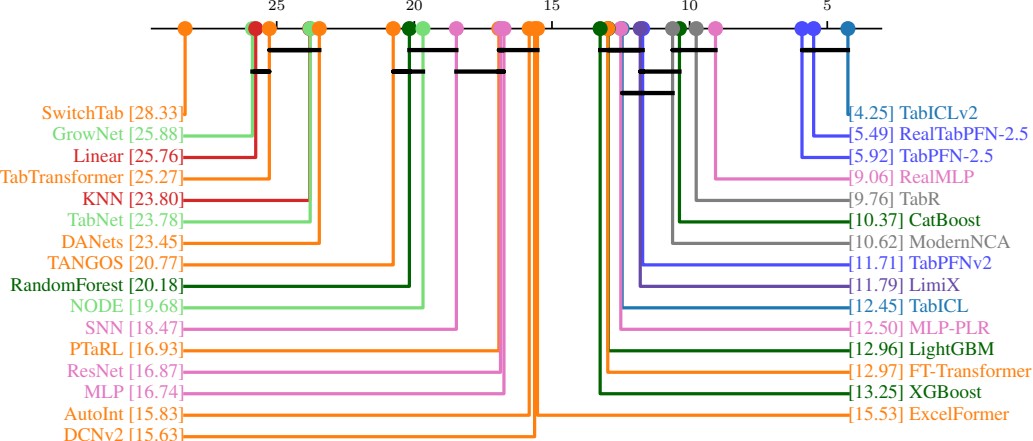

*Figure K.18.* **Critical difference diagram on large datasets of the TALENT benchmark.**

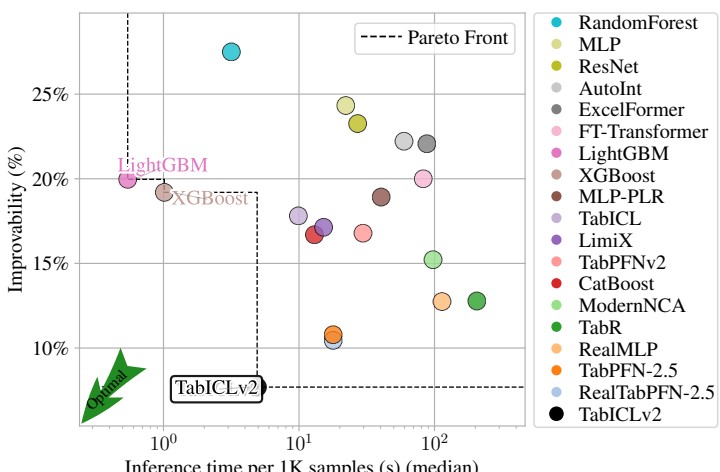

*Figure K.19.* **Pareto front of improvability and inference time on large datasets of the TALENT benchmark.**

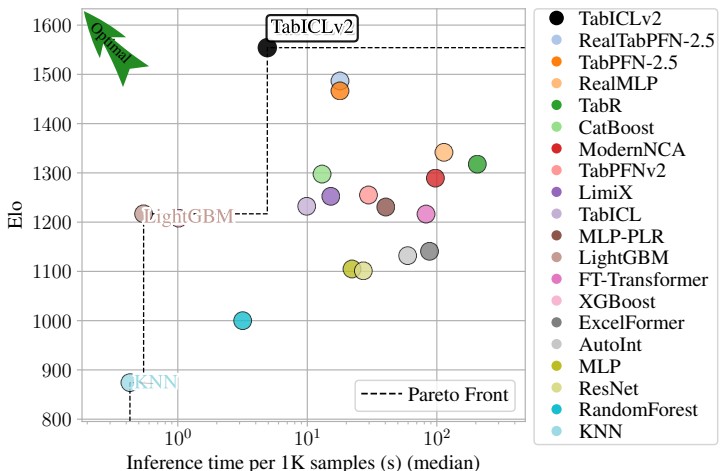

*Figure K.20.* **Pareto front of Elo and inference time on large datasets of the TALENT benchmark.**

## K.9. Model rankings with respect to meta-features

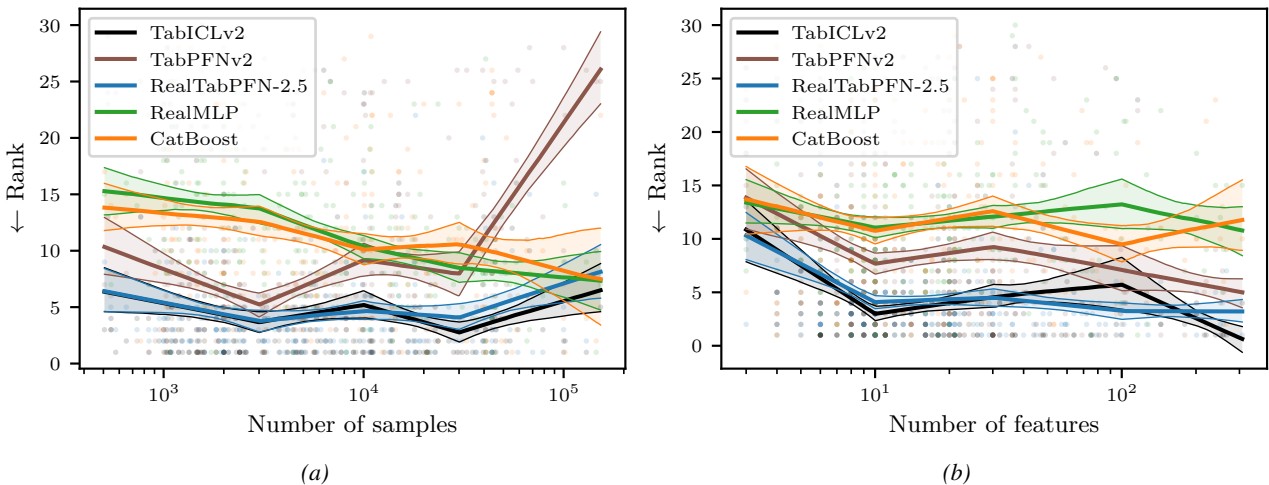

*(a)*                                                                  *(b)*

*Figure K.21.* **Model rankings with respect to meta-features across all datasets of the TALENT benchmark.**

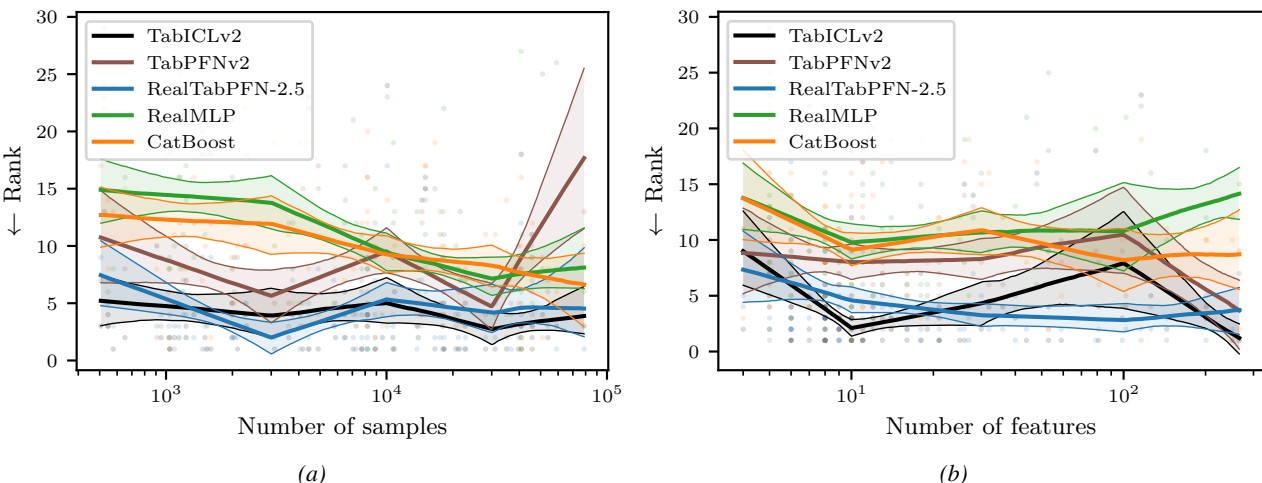

*Figure K.22.* **Model rankings with respect to meta-features across classification datasets of the TALENT benchmark.**

*Figure K.23.* **Model rankings with respect to meta-features across regression datasets of the TALENT benchmark.**

# L. Other Experiments

## L.1. Effect of ensemble size

We study the effect of ensemble size $E \in \{1, 2, 4, 8, 16, 32\}$ on the TALENT benchmark. Gains are reported relative to $E = 1$ averaged over all datasets in Table L.1. Performance improves with ensemble size but with diminishing returns; most gains are captured by $E = 8$.

## L.2. Direct statistical comparison to RealTabPFN-2.5

Here, we complement the critical difference diagrams with a direct statistical comparison between TabICLv2 and RealTabPFN-2.5. While the CD diagrams do not show a statistically significant difference between the two methods, they may lack statistical power, for instance due to corrections for multiple comparisons across many methods. We therefore performed a direct binomial test on TALENT to assess whether TabICLv2 achieves a win rate greater than 50% against RealTabPFN-2.5 / TabPFN-2.5. The results, reported in Table L.2, are statistically significant.

Alternatively, a Wilcoxon signed-rank test yields similar results.

## L.3. Few-shot benchmark

To evaluate TabICLv2 on tiny datasets, we run a simple few-shot benchmark with only 16 training+validation samples and up to 1984 test samples on TALENT through pytabkit (119 regression, 101 binary classification datasets; no multi-class to avoid too many missing classes in the small train+val set). We used 2 outer train/test splits and 4-fold inner cross-validation, ensembling the four fitted models (which worked better than refitting a single model on the full dataset). Gradient-boosted trees and MLPs use early stopping on the validation set based on log-loss (classification) or RMSE (regression). We only use default parameters to avoid overtuning on such small datasets. Note that TabICLv2 has never seen datasets below 300 training samples during pretraining.

Results are shown in Table L.3. TabICLv2 ranks second only to RealTabPFN-2.5 and outperforms all traditional methods, despite never being pretrained on datasets this small. Performance could likely improve further by including smaller datasets during pretraining.

## L.4. Few-shot comparison to LLMs

We evaluate LLMs on TALENT using the MachineLearningLM codebase (Dong et al., 2025), which provides a complete pipeline: tabular serialization (token-efficient Concat and TabLLM formats), prompt assembly, JSON parsing, and evaluation.

We test Qwen3.5 (4B, 27B, 112B) and Claude Opus 4.6. Due to compute constraints and the cost of proprietary APIs (ca. \$14 in Claude tokens for these experiments), we focus on 9 TALENT classification datasets with $\leq 2{,}048$ samples, using train sizes $\{8, 16, 64\}$ with random sampling.

Results (average accuracy across 9 datasets) are shown in Table L.4. Stronger LLMs yield better tabular accuracy. However, LLMs are only competitive in the very-few-shot regime (8 shots); TabICLv2 dominates from 16 shots onward while being orders of magnitude more efficient.

We also study sample selection and serialization with Qwen3.5-27B in Table L.5. K-center greedy (maximizing feature-space coverage) helps at 16 shots but hurts at 64 shots. The TabLLM format shows no benefit.

## L.5. Benchmarking TabPFN's Random Forest extension

Here, we explore the random forest (RF) extension proposed by Hollmann et al. (2025), which routes test samples through an RF ensemble and fits the TFM on training subsets at each leaf. This approach showed improvements for TabPFNv2 and TabICL on large datasets (Qu et al., 2025, Appendix G). We apply it to TabICLv2 on TALENT datasets with $>$10K samples. The results are shown in Table L.6.

Despite its positive effect on TabPFNv2 and TabICL, the RF extension degrades TabICLv2's performance. While it reduces peak GPU memory, TabICLv2 already handles large datasets well via our QASSMax. Partitioning data through RF routing hurts by fragmenting useful context. The RF extension is also slow. However, training example selection remains a complementary direction worthy of further investigation.

*Table L.1.* **Effect of ensemble size (number of estimators) on the TALENT benchmark relative to size 1.**

| Ensemble size | Classification accuracy gain | Regression RMSE gain | Inference time |
|---|---|---|---|
| 1 | — | — | 0.2s |
| 2 | +0.03% | +0.09% | 0.5s |
| 4 | +0.11% | +0.77% | 0.7s |
| 8 | +0.20% | +1.22% | 0.9s |
| 16 | +0.23% | +1.38% | 1.5s |
| 32 | +0.26% | +1.46% | 2.6s |

*Table L.2.* **Results of a binomial comparison test.** We test whether the win-rate of TabICLv2 is >50% on the TALENT benchmark (classification or regression) and report the p-value.

| Comparison | Metric | p-value |
|---|---|---|
| TabICLv2 vs. RealTabPFN-2.5 | Accuracy | 0.0018 |
| TabICLv2 vs. TabPFN-2.5 | Accuracy | 4.4e-05 |
| TabICLv2 vs. RealTabPFN-2.5 | AUC | 6.4e-09 |
| TabICLv2 vs. TabPFN-2.5 | AUC | 2.5e-09 |
| TabICLv2 vs. RealTabPFN-2.5 | Log-loss | 1.4e-08 |
| TabICLv2 vs. TabPFN-2.5 | Log-loss | 1.0e-10 |
| TabICLv2 vs. RealTabPFN-2.5 | $R^2$ | 0.0028 |
| TabICLv2 vs. TabPFN-2.5 | $R^2$ | 0.0028 |

## L.6. Noise resilience

TabArena and TALENT comprise real-world datasets spanning diverse domains with inherent noise. TabICLv2's strong performance on benchmarks already shows its noise resilience. To further probe this, we selected six low-noise binary classification datasets from TabArena (judged by the best AUC achieved in the TabArena paper) and progressively replaced 0%–50% of training labels with random ones. As shown in Figure L.1, the performance of TabICLv2 tends to degrade less than that of tree-based methods.

*Table L.3.* **Few-shot benchmark results on TALENT datasets subsampled to 16 training samples.** Here, TD refers to tuned defaults from pytabkit, Huertas refers to defaults optimized for few-shot learning from Huertas (2024), and RealMLP-TD (TabArena) refers to the version from TabArena without label smoothing and with 8 ensemble members per fold.

| Method | Classification rank (AUC) | Regression rank (RMSE) |
|---|---|---|
| RealTabPFN-2.5 | 4.00 | 4.18 |
| TabICLv2 | 4.11 | 5.02 |
| ExtraTrees | 4.49 | 5.52 |
| RandomForest | 4.91 | 6.22 |
| RealMLP-TD (TabArena) | 6.00 | 5.21 |
| LightGBM-Huertas | 6.07 | 7.89 |
| XGBoost-TD | 6.57 | 7.45 |
| CatBoost-TD | 6.76 | 8.61 |
| TabM | 6.96 | 7.13 |

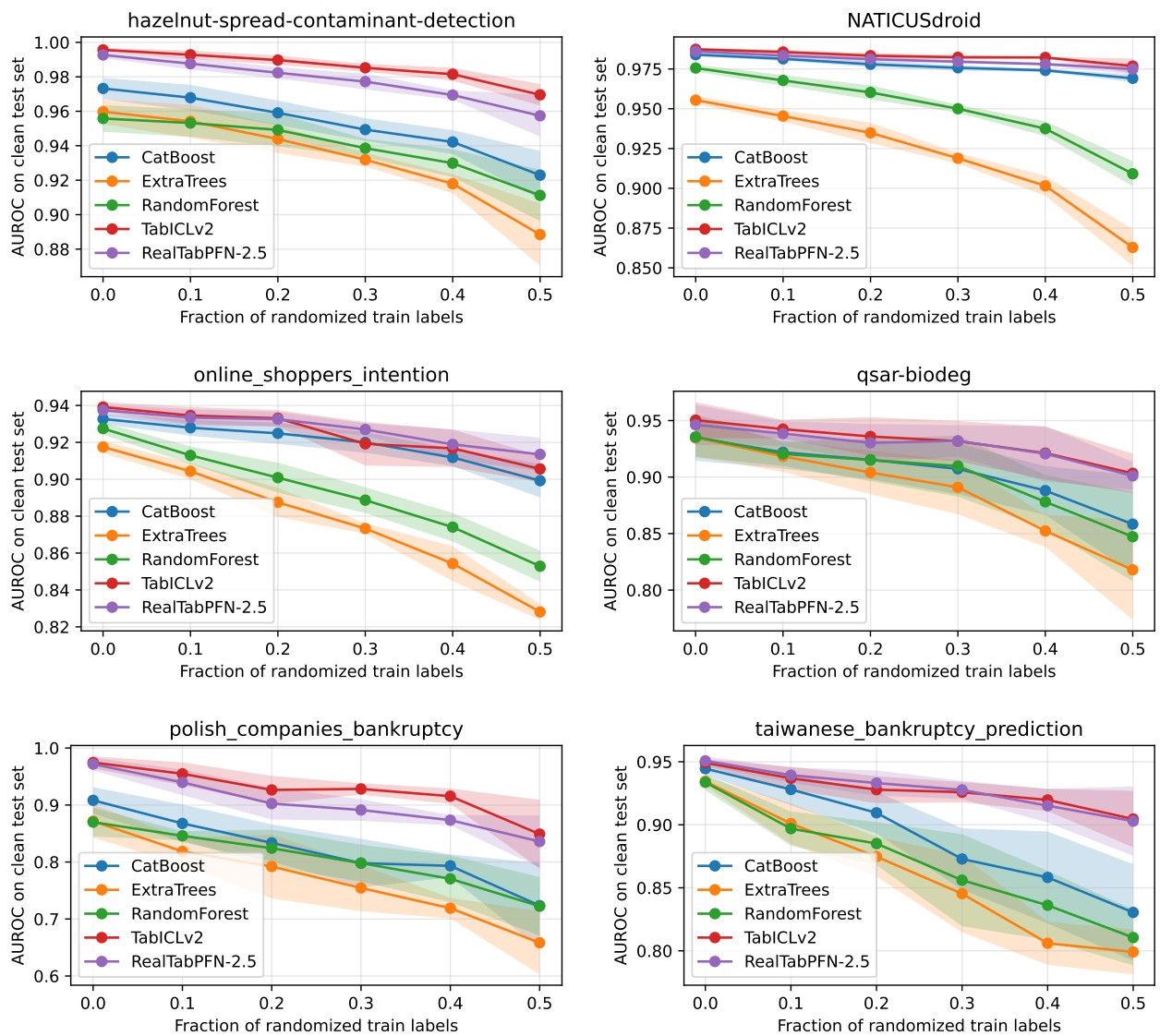

*Figure L.1.* **Robustness of different methods to label noise in binary classification.** We take the 6 binary datasets with the largest best AUROC from TabArena and replace different fractions of training points by random labels. TabICLv2 and RealTabPFN-2.5 show higher resistance to noisy labels than tree-based methods. CatBoost uses the tuned default parameters from pytabkit and an ensemble of models from 8-fold cross-validation, early-stopped for AUROC. Shaded areas show the standard deviations across 5 runs with random outer splits using 40% of the data for testing.

*Table L.4.* **Comparing TabICLv2 to LLMs on few-shot prediction accuracy averaged over 9 TALENT classification datasets.**

| Train size | Qwen3.5-4B | Qwen3.5-27B | Qwen3.5-112B | Claude Opus 4.6 | TabICLv2 |
|:---:|:---:|:---:|:---:|:---:|:---:|
| 8 | 0.482 | 0.505 | 0.511 | 0.536 | 0.515 |
| 16 | 0.492 | 0.526 | 0.531 | 0.541 | 0.586 |
| 64 | 0.522 | 0.569 | 0.574 | 0.606 | 0.659 |

*Table L.5.* **Effect of sample selection and serialization on Qwen3.5-27B.**

| Train size | Random | K-center greedy | TabLLM format |
|:---:|:---:|:---:|:---:|
| 16 | 0.526 | 0.556 | 0.521 |
| 64 | 0.569 | 0.530 | 0.520 |

*Table L.6.* **Evaluating the random-forest extension of Hollmann et al. (2025) on TabICLv2.** Quantities are averaged across TALENT datasets with >10K samples.

| | TabICLv2 | TabICLv2-RF |
|:---|:---:|:---:|
| Classification: Accuracy / AUC / Log-loss | 0.857 / 0.897 / 0.306 | 0.848 / 0.891 / 0.327 |
| Regression: R² / RMSE | 0.720 / 2293 | 0.687 / 2846 |

