# OpenReview forum: "TabICLv2: A Better, Faster, Scalable, and Open Tabular Foundation Model"
_ICML.cc/2026/Conference — ICML 2026 regular_

### Official Review · Reviewer_uQVH · 2026-03-09

**Soundness:** 4
**Presentation:** 4
**Significance:** 3
**Originality:** 3
**Overall Recommendation:** 5
**Confidence:** 4

**Summary:**

TabICooL introduces a state-of-the-art tabular foundation model built on three pillars: a novel synthetic data generation engine for high pretraining diversity, architectural innovations including Query-Aware Scalable Softmax (QASSMax) for better long-context generalization, and optimized pretraining protocols using the Muon optimizer. Overall, the manuscript explores how to improve scalability while achieving faster inference than existing methods. The model achieves state-of-the-art performance on TabArena and TALENT benchmarks without hyperparameter tuning.

**Compliance With Llm Reviewing Policy:**

Affirmed.

**Key Questions For Authors:**

- Can you provide more justification for choosing the Muon optimizer? Is this purely based on empirical evaluation, or are there theoretical reasons why Muon is particularly suited for tabular foundation models?
- How would recent LLMs (e.g., GPT-5, Claude Opus) perform on these benchmarks if given training data as in-context examples, potentially with random or diversity-based sample selection? This comparison would help position TabICooL against LLM-based approaches.
- Would training example selection help further improve performance and scalability?
- How resilient is the model to noise in the data, which is common in real-world settings?

**Limitations:**

yes

**Strengths And Weaknesses:**

Strengths
  - Achieves state-of-the-art performance on two established benchmarks (TabArena and TALENT) with a valid evaluation setting
  - Inference is more than 10x faster than competing foundation models while scaling to million-sample datasets under 50GB GPU memory
  - Strong commitment to open research: weights, synthetic data engine, and pretraining code will be released, addressing the closed-source trend in the field
  - Clear and well-organized paper presentation with detailed technical explanations and extensive ablation studies quantifying each contribution
  - Main claims (novel synthetic data generation, architectural innovations like QASSMax, Muon optimizer benefits) are well-supported by empirical evidence and ablations

Weaknesses
  - The choice of Muon optimizer over AdamW lacks sufficient justification beyond empirical performance

---

> ### Author Rebuttal · Authors · 2026-03-30
>
> We thank Reviewer uQVH for the thoughtful review. We address each question below.
>
> > 1. Can you provide more justification for choosing the Muon optimizer?
>
> The choice of Muon is primarily empirical but theoretically motivated. Muon [1] orthogonalizes gradient updates via Newton-Schulz iteration, corresponding to steepest descent under the spectral norm [2] — a geometry natural for Transformer linear layers, as it directly controls output sensitivity to weight updates. [3] proved that Muon with weight decay implicitly constrains the spectral norm of weight matrices, a regularization absent in AdamW. Since TabICooL shares the same Transformer backbone as LLMs, where Muon achieves ~2× efficiency over AdamW [4], similar benefits are expected. Our ablation confirms this: Muon improves over AdamW evidently. But we acknowledge that a theoretical analysis specific to the tabular ICL setting is an open question.
>
> [1] Muon: An optimizer for hidden layers in neural networks
> [2] Old Optimizer, New Norm: An Anthology
> [3] Muon Optimizes Under Spectral Norm Constraints
> [4] Muon is Scalable for LLM Training
>
> > 2. How would recent LLMs (e.g., GPT-5, Claude Opus) perform on these benchmarks...?
>
> We evaluated LLMs on TALENT using the [MachineLearningLM codebase](https://github.com/HaoAreYuDong/MachineLearningLM), which provides a complete pipeline: tabular serialization (token-efficient Concat and TabLLM formats), prompt assembly, JSON parsing, and evaluation.
>
> We tested Qwen3.5 (4B, 27B, 112B) and Claude Opus 4.6. Due to compute constraints and the cost of proprietary APIs (~$14 in Claude tokens for these experiments), we focused on 9 TALENT classification datasets with ≤2,048 samples, using train sizes {8, 16, 64} with random sampling.
>
> **Results (average accuracy across 9 datasets):**
>
> | Train size | Qwen3.5-4B | Qwen3.5-27B | Qwen3.5-112B | Claude Opus 4.6 | TabICooL |
> | --- | --- | --- | --- | --- | --- |
> | 8 | 0.482 | 0.505 | 0.511 | **0.536** | 0.515 |
> | 16 | 0.492 | 0.526 | 0.531 | 0.541 | **0.586** |
> | 64 | 0.522 | 0.569 | 0.574 | 0.606 | **0.659** |
>
> We also studied sample selection and serialization with Qwen3.5-27B. K-center greedy (maximizing feature-space coverage) helps at 16 shots but hurts at 64 shots. The TabLLM format shows no benefit.
>
> | Train size | Random | K-center greedy | TabLLM format |
> | --- | --- | --- | --- |
> | 16 | 0.526 | **0.556** | 0.521 |
> | 64 | **0.569** | 0.530 | 0.520 |
>
> Key insights: (1) Stronger LLMs yield better tabular accuracy. (2) LLMs are only competitive in the very-few-shot regime (8 shots); TabICooL dominates from 16 shots onward while being orders of magnitude more efficient.
>
> These findings align with TabuLa and MachineLearningLM. An extensive LLM evaluation is interesting but beyond the scope of this paper.
>
> TabuLa: Large Scale Transfer Learning for Tabular Data via Language Modeling
> MachineLearningLM: Scaling Many-shot In-context Learning via Continued Pretraining
>
> > 3. Would training example selection help further improve performance and scalability?
>
> Training example selection via nearest-neighbor sampling (e.g., LocalPFN) or training set compression (e.g., TuneTables) is actively explored for TFM scalability. However, the former prevents batch processing due to per-test-sample retrieval, while the latter requires costly per-dataset optimization.
>
> We instead explore the [random forest (RF) extension](https://github.com/PriorLabs/tabpfn-extensions) proposed by TabPFNv2 , which routes test samples through an RF ensemble and fits the TFM on training subsets at each leaf. This approach showed strong results for TabPFNv2 and TabICL on large datasets (TabICL, Appendix G). We applied it to TabICooL on TALENT datasets with >10K samples:
>
> |  | TabICooL | TabICooL-RF |
> | --- | --- | --- |
> | Classification: Accuracy / AUC / Log-loss | 0.857 / 0.897 / 0.306 | 0.848 / 0.891 / 0.327 |
> | Regression: R² / RMSE | 0.720 / 2293 | 0.687 / 2846 |
>
> Unlike its positive effect on TabPFNv2 and TabICL, the RF extension degrades TabICooL's performance. While it reduces peak GPU memory, TabICooL already handles large datasets well via our QASSMax. Partitioning data through RF routing hurts by fragmenting useful context. The RF extension is also slow, as shown in this figure (https://limewire.com/d/aY8XL#qVpA7K2DPh). However, training example selection remains a complementary direction worthy of further investigation.
>
> > 4. How resilient to noise in the data?
>
> TabArena and TALENT comprise real-world datasets spanning diverse domains with inherent noise. TabICooL's strong performance on benchmarks already shows its noise resilience. To further probe this, we selected six low-noise binary classification datasets from TabArena (judged by the best AUC achieved in the TabArena paper) and progressively replaced 0%–50% of training labels with random ones. As shown in this figure (https://limewire.com/d/vvGRM#Gmilc5UdnB), the performance of TabICooL tends to degrade less than tree-based methods.

---

> > ### Author Rebuttal · Reviewer_uQVH · 2026-04-03
> >
> > Thanks for the detailed response. This answers my questions.

---

### Official Review · Reviewer_GshV · 2026-03-10

**Soundness:** 3
**Presentation:** 4
**Significance:** 3
**Originality:** 2
**Overall Recommendation:** 5
**Confidence:** 3

**Summary:**

The paper introduces a novel tabular foundation model, TabICooL, with multiple architectural and training innovations in order to alleviate the limitations of existing models such as TabPFN and TabICL. Some of the major architectural improvements are incorporating repeated feature grouping, injecting label embeddings in an early stage of attention, and the use of query-aware softmax. The training process is improved mainly through the introduction of a novel synthetic prior. The new prior includes a different random graph sampling technique and additional function classes to model parent-child dependencies in the random causal graphs, compared to the TabPFN prior. Experiments on the performance of the model are based on two popular tabular benchmarks, TabArena and TALENT, on which the model shows significant improvements over existing foundation models. Multiple ablation experiments have been conducted to evaluate the contribution of each design choice.

**Compliance With Llm Reviewing Policy:**

Affirmed.

**Final Justification:**

In the light of the rebuttal, the results seem more significant and the authors have addressed all my concerns in detail.

**Key Questions For Authors:**

How does TabICooL perform in the few-shot regime (e.g., with <20 samples)?

**Limitations:**

Yes

**Strengths And Weaknesses:**

**Soundness:**

*Strengths:*
The paper is technically sound and builds upon the well-established literature on tabular foundation models and prior-data fitted networks. Authors have explained the heuristic insights behind most of the architectural design improvements over the existing methods. For example, the introduction of repeated feature grouping to avoid representation collapse and loss of fine-grained feature information is well-motivated and intuitive. The empirical study also establishes the contribution of each such modification, and is reflected in the final performance of TabICooL w.r.t. the existing models. Experiments are based on two widely used benchmarks, covering many binary and multi-class classification, and regression datasets.


*Weaknesses:*
However, there are many other modifications, particularly within the training framework, which do not have a proper rationale and justified only through the ablations. For example, target-aware embedding in the architecture, the correlated sampling of scalars in synthetic data generation, and the choice of the optimizer in the training setup. Moreover, based on the critical difference plots in the appendix, while TabCooL performs better than the existing models such as TabPFNv2 and TabICL, its performance doesn’t seem to be significantly better than TabPFN 2.5 and RealTabPFN.

**Presentation:**

The paper follows an easy-to-understand flow, and the content is well-organized into logical sections. All the modifications and the experiments are explained clearly, with additional details in the appendix.

**Significance:**

The problem of constructing tabular foundation models is an important area of research due to multiple reasons. Majority of potential machine learning applications in areas such as finance, healthcare and scientific discovery involves tabular data. In addition, the in-context learning performance of existing foundation models for language processing is not so good. Given these reasons, the focus of TabICooL towards expanding the horizons of tabular foundation models is timely and well-motivated. However, even though the model demonstrates significant performance gains over the existing models such as TabPFN and TabICL, the performance gains w.r.t. TabPFN 2.5 and RealTabPFN are not statistically distinguishable according to the critical difference plot.



**Originality:**
Overall, the work proposes a novel architecture and a training setup including modifications to the synthetic prior. However, considering each component independently, the improvements can be seen as either extensions of the existing works or combinations of them in contrast to a fundamentally new paradigm. Nevertheless, the total contribution can be considered original enough, given the overall performance gains exhibited in the empirical study.
The authors have sufficiently compared the proposed framework w.r.t. the existing methods within the dedicated section on related works as well as in the rest of the paper as relevant.

---

> ### Author Rebuttal · Authors · 2026-03-31
>
> We thank Reviewer GshV for the detailed and constructive review, and for recognizing the significance of the problem and the clarity of the presentation. We address each concern below.
>
> > 1. There are many other modifications which do not have a proper rationale and justified only through the ablations
>
> - **Target-aware embedding** increases expressiveness, as confirmed by our ablation (Figure 10): Unlike TabICL, which compresses features without knowing which are predictive, our early target injection enriches feature embeddings, allowing TabICooL to weight features by their relevance from the outset. It also mitigates representation collapse, as features with similar distributions can be distinguished by their differing target associations (Section 3, L151–154).
> - **Correlated sampling** reflects real-world data properties. For instance, the splice dataset in TabArena has 60 categorical features, 43 of which take on 5 distinct values. Under i.i.d. sampling, such homogeneity would be extremely unlikely during pretraining. Correlated sampling ensures the prior covers such realistic patterns.
> - **Muon optimizer** is primarily empirical but also theoretically motivated. We refer to our response to Reviewer uQVH, where we provide more reasons to motivate the choice of Muon.
>
> > 2. The performance gains w.r.t. RealTabPFN-2.5 are not statistically distinguishable according to the critical difference plot.
>
> The critical difference plots may lack statistical power due to corrections for multiple comparisons across many methods. We ran direct binomial tests on TALENT to test whether TabICooL achieves a >50% win rate against RealTabPFN-2.5 / TabPFN-2.5, and the results are clearly significant:
>
> | Comparison | Metric | p-value |
> | --- | --- | --- |
> | TabICooL vs. RealTabPFN-2.5 | Accuracy | 0.0018 |
> | TabICooL vs. TabPFN-2.5 | Accuracy | 4.4e-05 |
> | TabICooL vs. RealTabPFN-2.5 | AUC | 6.4e-09 |
> | TabICooL vs. TabPFN-2.5 | AUC | 2.5e-09 |
> | TabICooL vs. RealTabPFN-2.5 | Log-loss | 1.4e-08 |
> | TabICooL vs. TabPFN-2.5 | Log-loss | 1.0e-10 |
> | TabICooL vs. RealTabPFN-2.5 | R² | 0.0028 |
> | TabICooL vs. TabPFN-2.5 | R² | 0.0028 |
>
> Alternatively, a Wilcoxon signed-rank test yields similar results. We thank the reviewer for raising this point and will include these results in the revision.
>
> Beyond statistical significance, we emphasize that TabICooL achieves these results without real-data fine-tuning and is consistently faster than RealTabPFN-2.5 (Figure 6). Furthermore, RealTabPFN-2.5 pretraining and synthetic data generation are not publicly available, while we commit to fully open-sourcing TabICooL.
>
> > 3. How does TabICooL perform in the few-shot regime?
>
> We ran a dedicated few-shot benchmark with only 16 training+validation samples and up to 1984 test samples on TALENT (119 regression, 101 binary classification datasets; no multi-class to avoid too many missing classes in the small train+val set). We used 2 outer splits and 4-fold inner cross-validation, ensembling the four fitted models (which worked better than refitting a single model on the full dataset). We only use default parameters to avoid overtuning on such small datasets. Note that TabICooL has never seen datasets below 300 training samples during pretraining.
>
> Average ranks:
>
> | Method |  Classification rank (AUC) | Regression rank (RMSE) |
> | --- | --- | --- |
> | RealTabPFN-2.5 | **4.00** | **4.18** |
> | TabICooL | $\underline{4.11}$ | $\underline{5.02}$ |
> | ExtraTrees | 4.49 | 5.52 |
> | RandomForest | 4.91 | 6.22 |
> | RealMLP-TD (TabArena) | 6.00 | 5.21 |
> | LightGBM-Huertas | 6.07 | 7.89 |
> | XGBoost-TD | 6.57 | 7.45 |
> | CatBoost-TD | 6.76 | 8.61 |
> | TabM | 6.96 | 7.13 |
>
> (TD = tuned defaults from [pytabkit](https://github.com/dholzmueller/pytabkit); Huertas = defaults optimized for few-shot learning from https://arxiv.org/pdf/2411.04324)
>
> TabICooL ranks second only to RealTabPFN-2.5 and substantially outperforms all traditional methods, despite never being pretrained on datasets this small. Performance could likely improve further by including smaller datasets during pretraining.
>
> As a complementary perspective, we also compared TabICooL with LLMs in the few-shot regime in our response to Reviewer uQVH.
>
> > 4. The improvements can be seen as either extensions of the existing works or combinations … However, the total contribution can be considered original enough.
>
> We acknowledge that we are not introducing a new paradigm, but arguably no tabular foundation model paper has done so since TabPFNv1. We highlight two key contributions: (1) substantial extensions to the synthetic prior, including random GP functions with theoretical smoothness analysis and diverse random matrix types; and (2) QASSMax enabling long-context generalization. We discuss originality in more detail in our response to Reviewer R6UP.

---

> > ### Author Rebuttal · Reviewer_GshV · 2026-04-05
> >
> > Authors have addressed all my concerns in detail.

---

> > > ### Author Response · Authors · 2026-04-06
> > >
> > > Dear Reviewer GshV,
> > >
> > > Thank you for acknowledging that all concerns have been fully resolved. We truly appreciate the time and effort you invested in reviewing our work.
> > >
> > > We noticed that your acknowledgement selected option (a), which includes the suggestion to "please consider adjusting your score accordingly." Therefore, we want to kindly follow up on whether you might consider updating your overall recommendation in light of the resolved concerns. We believe this would help better reflect the current state of the review, which is very important to us.
> > >
> > > If there are any remaining considerations that may influence your score, we would be happy to address them promptly and incorporate the necessary changes in the final version.
> > >
> > > Thank you again for your constructive review, which has helped strengthen the paper.

---

### Official Review · Reviewer_RRau · 2026-03-12

**Soundness:** 4
**Presentation:** 4
**Significance:** 4
**Originality:** 3
**Overall Recommendation:** 5
**Confidence:** 3

**Summary:**

This paper presents TabICooL, a new tabular foundation model built on a stronger synthetic data generation engine together with additional architectural improvements. The authors evaluate the method on TabArena and TALENT, showing consistently strong results and a state-of-the-art overall accuracy/runtime tradeoff

**Compliance With Llm Reviewing Policy:**

Affirmed.

**Final Justification:**

The rebuttal reinforced my previous assessment that the submission should be accepted.

**Key Questions For Authors:**

How does performance change as the number of estimators increases? Can this be interpreted as a form of test-time compute?
Do the authors envision the synthetic data generation pipeline being useful beyond training tabular foundation models, for example as a more general synthetic benchmark or pretraining resource?

**Limitations:**

Yes

**Strengths And Weaknesses:**

Strengths:
The empirical evaluation is extensive and thorough, with many analyses that consistently support the strength of TabICooL.
The ablations, both in the main paper and in the appendix, are comprehensive and help clarify which components contribute most to the gains. In particular, the paper makes clear that both the synthetic data pipeline and the architectural changes matter.
I also appreciate the authors’ commitment to releasing the full codebase, including the synthetic data generation pipeline. This is especially valuable given the increasing closed-source trend in this area

Weaknesses
The runtime comparison is not perfectly apples-to-apples. TabICooL runtimes are measured on an H100 GPU, whereas the runtimes for competing methods are taken from the benchmark sources. As a result, it is difficult to fully disentangle improvements due to algorithmic advances from differences in evaluation setup.
At review time, the full codebase is not yet available, so reproducibility is still incomplete.

---

> ### Author Rebuttal · Authors · 2026-03-30
>
> We thank Reviewer RRau for the positive assessment and for highlighting the rigor of our evaluation, ablations, and open-source commitment. We address each point below.
>
> > 1. The runtime comparison is not perfectly apples-to-apples
>
> We acknowledge that a fully apples-to-apples runtime comparison would require rerunning all methods on identical hardware, which is prohibitively expensive, and some classical models do not support GPU. In TabArena, GPU-based methods (TabPFNv2, TabICL, TabDPT, TabM, ModernNCA) were benchmarked on NVIDIA L40S 48GB, as mentioned in the TabArena paper. For TALENT, hardware details are not specified in the paper or shared data. However, the most important comparison, that is, TabICooL vs. RealTabPFN-2.5, is fair: RealTabPFN-2.5 uses H200 GPU in TabArena (Figure 1) and H100 in TALENT (Figure 5), while TabICooL is measured on H100. Furthermore, Figure 6 provides a strictly controlled comparison between TabICooL and TabPFN-2.5 on identical hardware across three platforms (H100 GPU, AMD CPU, Apple CPU), where TabICooL is consistently faster than TabPFN-2.5.
>
> > 2. How does performance change as the number of estimators increases? Can this be interpreted as a form of test-time compute?
>
> We studied the effect of ensemble size $E \in \\{1, 2, 4, 8, 16, 32\\}$ on the TALENT benchmark. Gains are reported relative to $E=1$ averaged over all datasets:
>
> | Ensemble size $E$ | Classification accuracy gain | Regression RMSE gain | Inference time |
> | --- | --- | --- | --- |
> | 1 | — | — | 0.2s |
> | 2 | +0.03% | +0.09% | 0.5s |
> | 4 | +0.11% | +0.77% | 0.7s |
> | 8 | +0.20% | +1.22% | 0.9s |
> | 16 | +0.23% | +1.38% | 1.5s |
> | 32 | +0.26% | +1.46% | 2.6s |
>
> Performance improves with ensemble size but with diminishing returns; most gains are captured by $E=8$, with marginal improvement beyond $E=16$.
>
> Yes, this can be interpreted as a simple form of test-time compute. Each estimator runs with a different random column permutation, preprocessor, and class label shuffle, providing a complementary view of the data. Increasing $E$ trades inference time for improved robustness by aggregating these diverse predictions.
>
> > 3. At review time, the full codebase is not yet available, so reproducibility is still incomplete
>
> We fully commit to releasing all weights, synthetic data engine, and pretraining code upon acceptance. We believe this is particularly important given the increasing closed-source trend in tabular foundation models.
>
> > 4. Do the authors envision the synthetic data generation pipeline being useful beyond training tabular foundation models?
>
> Yes, the SCM-based synthetic prior paradigm is already being adopted across multiple domains, and we see broad potential:
> (1) **Causal ML**. SCM-generated datasets with known ground-truth causal structure are valuable for both benchmarking and amortized inference. CausalProfiler [1] uses SCM-sampled datasets with coverage guarantees to rigorously evaluate causal methods across all three levels of Pearl's hierarchy. CausalPFN [2] and Do-PFN [3] directly adopt the PFN synthetic pretraining paradigm for amortized causal effect estimation via in-context learning.
> (2) **Time series foundation models**. CauKer [4] combines GP kernel composition with SCMs to pretrain foundation models for time-series classification purely on synthetic data.
> (3) **Relational database foundation models**. High-quality relational databases are scarce and private, making synthetic generation critical. RDB-PFN [5] introduces a relational prior generator extending the PFN SCM framework to multi-table settings. PluRel [6] demonstrates power-law scaling for relational FMs pretrained on SCM-generated databases.
> (4) **Enhancing LLM capabilities**. MachineLearningLM [7] showed that continued pretraining of LLMs on SCM-generated tabular tasks significantly improves in-context tabular learning of LLMs.
>
> We commit to open-sourcing our synthetic data engine upon acceptance. We believe this will directly support and accelerate research across all these directions.
>
> References:
> [1] CausalProfiler: Generating Synthetic Benchmarks for Rigorous and Transparent Evaluation of Causal Machine Learning
> [2] CausalPFN: Amortized Causal Effect Estimation via In-Context Learning
> [3] Do-PFN: In-Context Learning for Causal Effect Estimation
> [4] CauKer: Classification Time Series Foundation Models Can Be Pretrained on Synthetic Data
> [5] RDB-PFN: Relational In-Context Learning via Synthetic Pre-training with Structural Prior
> [6] PluRel: Synthetic Data unlocks Scaling Laws for Relational Foundation Models
> [7] MachineLearningLM: Scaling Many-shot In-context Learning via Continued Pretraining

---

> > ### Author Rebuttal · Reviewer_RRau · 2026-04-01
> >
> > Thank you for your work, and looking forward for your code.

---

### Official Review · Reviewer_R6UP · 2026-03-16

**Soundness:** 3
**Presentation:** 3
**Significance:** 3
**Originality:** 1
**Overall Recommendation:** 2
**Confidence:** 5

**Summary:**

The paper introduces follow-up work presenting a series of improvements to the TabICL tabular foundation model.

In particular, the authors present the following improvements:

- A revised synthetic prior
- Changing the softmax layer of their architecture to handle attention fading in long contexts
- Repeated feature grouping
- Many-class classification through mix-radix ensembling

All proposed patches are either known techniques applied to an existing model or minor variations of the existing TabICL model.

As a result, the novelty of the paper relies mostly on the results, rather than on the method.

**Compliance With Llm Reviewing Policy:**

Affirmed.

**Final Justification:**

I recommend a direct rejection because the method is not reproducible, and the authors' explicitly expressed position of intentionally submitting a non-reproducible paper, such that "competitor methods do not benefit".

This rationale of intentional nonreproducibility continues the path of a dangerous precedent in tabular foundation model research and opposes any principle of peer review. Having SOTA results by using "special and non-disclosed pretraining data" does not advance science.

If authors do not want the community to profit from their research in a reproducible manner, then I recommend arXiv instead of peer-reviewed publishing.

**Key Questions For Authors:**

Can you better explain the performance of Ref with TabICL prior in the experiment Figure 10?

**Limitations:**

Why is the prior code not released? This severely limits the reproducibility of the paper.

As a result, I am giving a rejection recommendation.

**Strengths And Weaknesses:**

The paper is sound, and the presentation is clear. The paper addresses a niche domain in the Deep Learning community, which is nevertheless gaining momentum.

However, the originality is lacking as this reads as a follow-up work of TabICL, with patching together mostly known techniques/components in a novel combination.

---

> ### Author Rebuttal · Authors · 2026-03-30
>
> We thank the reviewer for their feedback and address the main points below.
>
> > 1. The paper addresses a niche domain in the Deep Learning community, which is nevertheless gaining momentum
>
> We respectfully disagree that tabular machine learning constitutes a “niche domain” at this stage. There have been many workshops at NeurIPS (TRL workshop) and ICML (FMSD workshop), and the TabPFN Nature paper has gained 872 citations within 14 months of its publication. Industry investment (e.g., the Fundamental Series A of 255 million $) further reflects this momentum.
>
>
> > 2. All proposed patches are either known techniques applied to an existing model or minor variations of the existing TabICL model.
>
> We believe this assessment does not accurately reflect the technical contributions. Some examples:
> - **Prior design**: Our prior involves many new components (as discussed in Appendix E.1). For example, we propose a novel mechanism for sampling from Gaussian processes with random kernels in a way that provably approximates functions of different smoothness (Appendix E.8 and G). We introduce several new families of random functions, new matrix-sampling schemes beyond Gaussian matrices used in prior work, soft feature–importance mechanisms via random weights, etc.
>
> - **QASSMax** is not “changing the softmax layer”. It is a separate module that rescales the queries based on themselves (self-gating) and the context size. It is crucial to achieving an attention block with a behavior optimal for a varying number of elements.
>
> - **Mixed-Radix ensembling** is a new technique. It is not obvious that averaging the embeddings of different class encodings should work.
>
> Under the standard implied by the review, many impactful recent papers would be judged as insufficiently novel (RealMLP, TabPFNv2, etc). Importantly, if achieving the reported performance required only “minor variations”, it is unclear why no prior model has reached similar results. The appendix documents many unsuccessful attempts: iterating through these design choices is part of the contribution.
>
> > 3. Why is the prior code not released?
>
> As stated in the paper, the full code—including the prior—will be released upon publication.
>
> There are two reasons: (1) the prior code was not refactored yet at the time of submission, and (2) The prior plays a key role to create state-of-the-art tabular foundation models. This is the reason why PriorLabs (creators of TabPFNv2) have released everything except their prior, thereby preventing reproducibility. Prematurely releasing our prior could disadvantage our submission if competing methods integrated it before acceptance. Nonetheless, we have described the prior in great detail in the appendix, which should confirm that our goal is not to keep things secret, unlike some competitors (TabPFNv2, LimiX, etc.).
>
> Two reviewers (RRau, uQVH) noted that committing to the full code release (including the prior) is a strong point of the paper “given the increasing closed-source trend in this area”.
>
> > 4. Can you better explain the performance of Ref with TabICL prior in the experiment Figure 10?
>
> To clarify the setup: "Ref" denotes the reference model, which is TabICooL without QASSMax (i.e., with all other innovations: target-aware embedding, repeated feature grouping, new prior, Muon optimizer). "Ref with TabICL prior" trains this same architecture but replaces our new prior with the TabICL prior.
>
> Under the setting of "Ref with TabICL prior" (gray line in Figure 10), performance is significantly worse. We attribute this to a capacity-diversity mismatch: TabICooL's architecture is more expressive than TabICL's, and requires a correspondingly diverse prior to learn general prediction strategies. The TabICL prior lacks sufficient diversity to fully leverage TabICooL's increased capacity, resulting in poor generalization to real-world validation datasets.
>
> Conversely, "Ref with TabICL architecture" (orange line) pretrains the TabICL architecture with our new prior. It only matches TabICL, indicating that the simpler architecture lacks the capacity to exploit the increased prior diversity.
>
> This architecture-prior interaction demonstrates that our contributions are not independent patches but form a synergistic whole because both the architecture and the prior are jointly necessary for TabICooL's performance.
>
> From a different perspective, perhaps TabICooL is better at approximating the true Bayesian posterior, but perhaps the true Bayesian posterior isn't that good for the TabICL prior (e.g., it assumes categoricals and classification labels are generated from discretizing 1D continuous data) while the lower flexibility of TabICL would allow to generalize better for its own prior. A similar effect was observed in the arXiv v1 of TabDPT (https://arxiv.org/pdf/2410.18164v1), where the performance on synthetic data degrades when scaling to large models.

---

> > ### Author Rebuttal · Reviewer_R6UP · 2026-03-31
> >
> > You are accepting that you are intentionally making the paper non-reproducible during the review period by not disclosing the prior code.
> >
> > The argument "we do not want competitors to use our datasets during the review" is not a justifiable scientific reason. If baselines are worse only because you have a better dataset, then the scientific community deserves to know this.
> >
> > Furthermore, vague promises that the experimental code for the dataset generation "will be released upon publication" define the purpose of peer review. Our duty as reviewers is to verify your empirical results by inspecting the code and by reproducing the experiments.
> >
> > As a result, my recommendation to reject the paper remains unchanged.

---

> > > ### Author Response · Authors · 2026-04-07
> > >
> > > We thank the reviewer for the follow-up.
> > >
> > > > If baselines are worse only because you have a better dataset, then the scientific community deserves to know this.
> > >
> > > Our ablation study (Figure 10) addresses this concern. Pretraining the TabICL architecture with our prior (orange line) only matches TabICL, showing that the prior alone does not explain TabICooL's gains. Performance requires the synergy of both architecture and prior, as we detailed in our first response to your question "Can you better explain the performance of Ref with TabICL prior in Figure 10?". We believe this ablation provides exactly the kind of transparency the reviewer is asking for.
> > >
> > > We also note that no competing tabular foundation model (eg, TabPFNv2, RealTabPFN-2.5, LimiX) has released their pretraining and prior code, making cross-prior comparisons currently impossible regardless. Moreover, their papers or technical reports do not provide sufficient details for reproducibility. In contrast, our paper provides over 15 pages of detailed prior description (Appendices E, F, G).

---

### Decision · Program_Chairs · 2026-04-30

**Decision:**

Accept (regular)

**Comment:**

All reviewers agreed that the paper makes significant contributions, and the results were further strengthened during the rebuttal. The authors addressed all concerns raised. While reproducibility is a valid and essential requirement—as one reviewer noted—I believe the authors provided sufficient evidence in the rebuttal and appendices to support their claims.